# Micro-Learning for Learning-Hard Problems

## Abstract

ML increasingly faces high complexity nonlinear data whose noise, imbalance, or small sample size thwart conventional models. We formalize this difficulty through the notion of Learning Hard Problems (LH-Ps), tasks that (i) defeat the vast majority of models, yet (ii) admit at least one high-quality solution if the relevant label-aware structural knowledge is appropriately incorporated during training. To address this, we introduce Micro-Learning (MiL), a principled framework that constructs traininglets: small, knowledge-fused subsets of the training data with demonstrably low complexity and infers a deterministic local model for each that collectively form a global predictor. We prove that the decision version of optimal traininglet selection is NP-complete, establishing a strong theoretical foundation for MiL. MiL dramatically reduces overfitting risk by eliminating irrelevant or noisy samples, while retaining interpretability and reproducibility through deterministic optimization in a RKHS space. Experiments in benchmark domains, from music information retrieval to medical proteomics, show that MiL solves LH-Ps and outperforms deep learning and classical baselines, especially on imbalanced or small-sample datasets, with negligible overfitting. Moreover, our work provides (i) the $1^{st}$ definition of LH-Ps, (ii) a Learning-Hard Index to quantify task difficulty pre-training, and (iii) theoretical guarantees on traininglet optimality and complexity, enriching learning theory and ethical AI.

## 1 Introduction

Modern AI increasingly confronts highly complex nonlinear data with noise, imbalance, or small-sample size challenge existing models. We argue that many such tasks belong to an under-studied class we call *Learning-Hard Problems (LH-Ps)*. An LH-P is characterized by two conditions:

1. *Near-universal failure*: almost all models in a broad hypothesis space perform poorly;
2. *Latent solvability*: there exists at least one model that can achieve satisfactory results once appropriate knowledge is fused into training, i.e., a good performance certificate exists.

**Definition 1** (**Learning-Hard Problem (LH-P)**). *Let $\mathcal{X}, \mathcal{Y}, \mathcal{P}, \mathbb{H}$ be the spaces of input data, label, and (unknown) data distribution respectively. For each $h \in \mathbb{H}$, let $L : \mathcal{Y} \times \mathcal{Y} \to \mathbb{R}_{\geq 0}$ be a loss, and define the generalization risk $R(h) = \mathbb{E}_{(x,y) \sim \mathcal{P}}\big[L\big(h(x), y\big)\big]$. Assume a family of knowledge-injection operators $\mathcal{K} = \{\varphi_\kappa : \mathcal{X} \to \mathcal{X}\}_\kappa$. A supervised task is an LH-P with respect to $(\mathbb{H}, \mathcal{K})$ if there exist constants $0 < \tau \ll \tau^\star$ satisfying:*

*(C1) Near-universal failure:* $\quad \min_{h \in \mathbb{H}} R(h) \geq \tau^\star$, *(C2) Latent solvability:* $\exists \kappa \in \mathcal{K}, \; h^\star \in \mathbb{H} \text{ s.t. } R\big(h^\star \circ \varphi_\kappa\big) \leq \tau.$

*Here $(h^\star \circ \varphi_\kappa)(x) = h^\star\big(\varphi_\kappa(x)\big)$; the operator $\varphi_\kappa$ is a fixed, and knowledge-fusion preprocessing map that can be a label-aware projection or a re-sampling operator that corrects distribution shift for training and test data.*

**LH-P Interpretation.** C1 states that *all* vanilla models in $\mathbb{H}$ incur a high failure risk, whereas C2 guaranties the *existence of a verifiable certificate of solvability*: some pair $(h^\star, \kappa)$ achieves a low risk once appropriate knowledge is fused. Importantly, Def. 1 is existential; it does *not* assert that standard training procedures can efficiently discover $(h^\star, \kappa)$, i.e., how to do knowledge fusion. In this study, we propose a Micro-Learning (MiL) approach to achieve this by conducting label-aware structural knowledge fusion through *traininglet* construction.

**Deep Learning Falters on LH-Ps.** LH-Ps are pervasive, appearing in domains from polyphonic music tagging and speech emotion recognition (SER) to imbalanced omics classification and COVID-19 diagnosis, etc. (Fuhrmann & Herrera, 2010), where deep learning (DL) models consistently stall at mediocre or even poor performance. For example, consider the IRMAS music-tagging

benchmark visualized in Fig.1.(b,c). Even SOTA architectures struggle: the carefully engineered convolutional network of Han et al. (2017) reaches only 60.2% F1, while the more sophisticated multitask CNN with onset-group auxiliary classification proposed by Yu et al. (2020) climbs to 68.5 %, still far from acceptable in practice.

The DL failure also is rooted in its built-in black-box nature (Li et al., 2019), vulnerability to overfitting, and poor reproducibility. Given an L-layer network with layer transformation function $g_i$, DL yields a highly nested decision function:

$$G(x) = f_{\text{softmax}}\Big((g_L \circ \cdots \circ g_1)(x)\Big) \tag{1}$$

This structure, analogous to an extremely high-order polynomial, is highly sensitive, meaning small input perturbations can cause large output swings, not to mention the non-determinism from model itself, parallel GPU speedup, and data preprocessing.

This architectural failure also lies in DL's lack of a knowledge-fusion mechanism to probe for latent label and structural information during training. This flaw is apparent with datasets that have hard-to-extract features, small-sample sizes, imbalanced distributions, or a significant distribution shift between training and test data. Although techniques such as sharpness-aware minimization (Foret et al., 2020) help, they neither expand the hypothesis space nor fuse knowledge in training.

**Inspire MiL.** We argue the key to solve LH-Ps is fuse knowledge during training. Figure 1 illustrates the core challenge of LH-Ps using the IRMAS dataset, a polyphonic music-tagging benchmark with 11 instrument classes (Yu et al., 2020). While a solution path may exist conceptually (a), the raw data of an LH-P can appear as a tangled swirl in t-SNE visualization (b), making this path hard to find. Feature selection offers little improvement (c), demonstrating that simple dimensionality reduction is insufficient. However, when label information is fused into the embedding process (d), the classes become clearly distinct. This reveals the key: the problem is not a lack of signal, but the failure of standard methods to leverage label-aware or structural knowledge, motivating our MiL.

**Micro-Learning (MiL).** We introduce *Micro-Learning* (MiL) for LH-Ps. For each query point, MiL extracts an instance-specific *traininglet*: a tailored, micro-sized training subset to classify that query with minimal learning cost. Traininglet construction is a *label-aware structural knowledge fusion* process. It selects the most discriminative samples while preserving their geometric proximity to the query's neighborhood. On this *traininglet*, we fit a deterministic, interpretable *learninglet* (e.g., a regularized SVM or a variant) in a Reproducing Kernel Hilbert Space (RKHS) to get a prediction function customized to that query.

*MiL learns locally.* Rather than training a single global model and relying on it to generalize to every query, MiL learns locally and adaptively for each query. This approach is inherently more resistant to overfitting and, through the use of an SVM-based *learninglet*, yields interpretable and reproducible predictions. By fusing label-aware structural knowledge before model induction and operating online per-query, MiL differs fundamentally from local-SVM ensembles, meta-learning kernels, and curriculum learning (Aha, 1997; Tappen et al., 2001; Snell et al., 2017; Bengio, 2009).

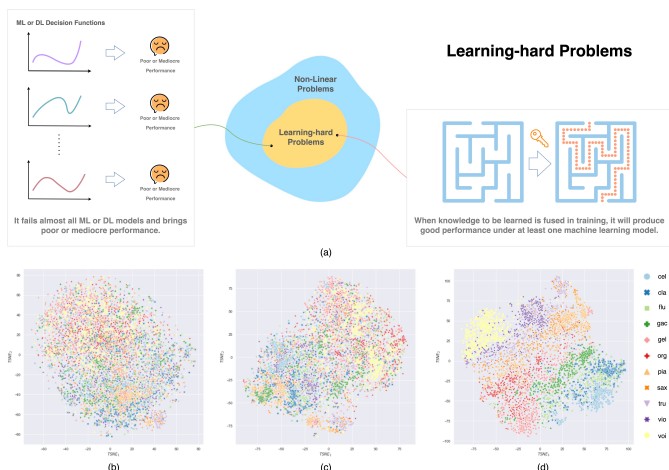

Figure 1: (a) Conceptual diagram of an LH-P where a viable solution path (gold) exists but is hard for standard learners to find. (b–d) t-SNE visualizations of IRMAS: (b) raw data lacks clear class structure; (c) feature selection yields only slight separation; (d) label-aware t-SNE reveals clear class clusters, demonstrating latent solvability

**Contributions.** (1) Formalize LH-Ps, introduce the Learning-Hard Index (LHI), and present MiL, an overfitting-resistant, explainable, and reproducible model for LH-P solving. (2) MiL successfully

solves LH-Ps that defeat various baselines on benchmarks. (3) Prove that the decision problem for optimal traininglet selection is NP-complete, and provide theoretical guarantees that MiL contracts train–test total-variation distance while reducing local Rademacher complexity. (Koltchinskii, 2006).

## 2 RELATED WORKS

*Kernel methods* fail on LHPs due to poor scalability ($\mathcal{O}(n^2)$ storage) and their tendency to amplify noise in high-variance data (Yu et al., 2020). *Deep networks* excel at automatic feature extraction but their nested nonlinearities are sensitive to perturbations, causing over-fitting, weak reproducibility and limited interpretability. Techniques such as sharpness-aware minimization (SAM) (Foret et al., 2020) and refined capacity measures (Zhang et al., 2021) reduce, and long-tail vision strategies like GLMC (Du et al., 2023) and PaCo (Cui et al., 2021) reduce, but do not eliminate these issues, leaving LH-Ps unsolved.

*Local and test-time learners.* Gradient-based meta-learning such as MAML (Finn, 2017), and local model-builders like LIME (Marco, 2016), MAPLE (Gregory, 2018), and T3A (Iwasawa et al., 2021) rely only on observed features, leaving them vulnerable to LH-P failure modes. Similarly, test-time adaptation methods (e.g., Tent (Wang et al., 2021)) fine-tune on target batches but cannot escape the original hypothesis space or fuse the knowledge required to solve LH-Ps.

## 3 DIAGNOSING LEARNING-HARDNESS WITH LEARNING-HARD INDEX

**Learning-Hard Index (LHI).** To efficiently diagnose LH-Ps without relying on their formal definition, we introduce the Learning-Hard Index (LHI). The LHI, a scalar in [0,1], is a practical metric to quantify a dataset's intrinsic complexity, providing a straightforward way to determine if a classification task is an LH-P. In general, a higher LHI indicates greater learning difficulty and, therefore, a higher likelihood that the task is learning-hard. Assuming nominally clean labels, we classify a task as an LH-P when its $LHI \geq 0.80$.

In contrast to model-centric measures (e.g., Rademacher complexity), LHI is data-centric. Because it can be computed before any training begins, LHI serves as a lightweight, model-agnostic score for comparing datasets and for deciding when specialized frameworks, such as MiL, are warranted.

*Quasi-LH-Ps.* For those datasets whose LHI falls in the [0.75, 0.80) range, we term their classification task as quasi-LH-Ps. Quasi-LH-Ps still suffer from the *Near-Universal Failure condition, but to a lesser degree.* Consequently, while many standard models may still underperform, the probability that a more powerful, well-tuned 'vanilla' model might find a satisfactory solution is considerably higher than for a true LH-P.

**LHI computing.** Let $X = \{(x_i, y_i)\}_{i=1}^m$ be a labeled dataset. We first obtain a locality-preserving embedding $X_r = f_{\mathrm{dm}}(X)$ via a local nonlinear dimension reduction map $f_{dm}$ (e.g., t-SNE McInnes et al. (2018).) We then group $X_r$ using a clustering algorithm $\Theta$ (e.g., $k$-means) to generate pseudolabels $y_{\mathrm{p}i}$, forming the pseudolabeled reference set $X_p = \{(x_i, y_{\mathrm{p}i})\}_{i=1}^m$.

The LHI is defined as $\mathrm{LHI}(X) = 1 - \mathrm{AMI}(X_r, X_p)$, where the Adjusted Mutual Information (AMI) is:

$$\mathrm{AMI}(X_r, X_p) = \frac{\mathrm{MI}(X_r, X_p) - \mathbb{E}[\mathrm{MI}(X_r, X_p)]}{\frac{1}{2}(H(X_r) + H(X_p)) - \mathbb{E}[\mathrm{MI}(X_r, X_p)]}. \tag{2}$$

Here, MI denotes mutual information and $H(\cdot)$ is Shannon entropy. Because AMI rewards embeddings that preserve local neighborhoods, it serves as a *robust* basis for the LHI.

Unlike global linear projections (e.g. PCA) blurring minority manifolds, the embedding map t-SNE maintains data locality better Han et al. (2022), and produces an LHI that faithfully reflects intrinsic task difficulty. Crucially, this facilitates clustering without biasing the metric: *Suppl. N* verifies no statistical difference between t-SNE and raw-feature LHI, confirming the metric captures intrinsic complexity of LH-Ps rather than artifacts.

**Cufoff:** Thresholding $\mathrm{LHI}(X) \geq 0.80$, i.e., when the clustering retains $\leq 20\%$ of neighborhood mutual information, reliably flags learning-hard tasks that demand specialized training (e.g., MiL) to achieve acceptable accuracy. A significant negative correlation ($r = -0.67, p = 0.035$) between LHI and performance validates $\mathrm{LHI} \geq 0.80$ as the critical threshold where standard model efficacy collapses (*Suppl. O*).

**Evaluate LH-P Data with LHI:** We evaluate LHI on 5 benchmarks spanning music, speech, health, and medicine: IRMAS Yu et al. (2020), CASIA (Li et al., 2016), SAVEE (Haq et al., 2008), Ovarian (Han et al., 2023), and a curated COVID-19 triage dataset. Table 1 summarizes key statistics; Although COVID19 falls slightly

below the 0.80 threshold, we include this to test MiL's sensitivity to solving a quasi-LH-P. More data details can be found in *suppl. G*.

Table 1: Datasets of learning-hard problems

| Dataset | (n, p) | Imbalance / sample rate | Classes | LHI |
|---|---|---|---|---|
| IRMAS | (6705, 518) | N | 11 | 90.7% |
| CASIA | (1200, 54) | N | 6 | 87.3% |
| SAVEE | (480, 54) | N | 7 | 81.4% |
| COVID-19 | (128, 48) | Y (57.03%: 37.5%:5.47%) | 3 | 78.5% |
| Ovarian | (266, 20531) | Y (98.50%: 1.5%) | 2 | 97.6% |

The LHI identifies *when* conventional training fails, but not *how* to succeed. Our core insight is that an LH-P can often be solved on a judiciously chosen customized small *subset* of the training set called a *traininglet* by fusing *label-aware structural knowledge for a query point for a minimal overfitting risk.* We formalize this idea using local Rademacher complexity (Bartlett & Mendelson, 2002). in Prop. 1

**Local Rademacher complexity.** For a sample $S = \{z_1, \ldots, z_n\}$ and a function class $\mathcal{F}$, define the radius-$r$ neighborhood $\mathcal{F}_r(f) = \{g \in \mathcal{F} : \|g - f\|_2 \leq r\}$. Its local Rademacher complexity is $\mathcal{R}_n\big(\mathcal{F}_r(f)\big) = \mathbb{E}_{S,\sigma}\Big[\sup_{g \in \mathcal{F}_r(f)} \frac{1}{n} \sum_{i=1}^n \sigma_i\, g(z_i)\Big]$, where each $\sigma_i$ is an independent Rademacher variable. Smaller $\mathcal{R}_n$ implies tighter generalization bounds in the neighborhood of $f$.

Prop.1 (*proof in Suppl. A*) provides the theoretical grounding for our approach, stating that every LH-P contains *a 'sweet-spot' model within a region of minimal overfitting risk (i.e., minimal local Rademacher complexity).* Our MiL is designed to systematically find this low-capacity region.

**Prop.1 (Low-capacity witness).** For any LH-P with hypothesis class $\mathbb{H}$ and any radius $r > 0$, there exists a model $f^\star \in \mathbb{H}$ such that $\mathcal{R}_n\big(\mathcal{F}_r(f^\star)\big) = \inf_{f \in \mathbb{H}} \mathcal{R}_n\big(\mathcal{F}_r(f)\big)$, meaning $f^\star$ minimizes the local Rademacher complexity over $\mathbb{H}$. Consequently, $f^\star$ and every model within its $r$-ball neighborhood enjoy the tightest generalization bound available in the entire hypothesis space.

*Why Prop. 1 matters.* Even though $\mathbb{H}$ is inflated by noise and nonlinearity, Prop. 1 guarantees at least one "sweet-spot" region where overfitting risk is minimal. The practical challenge is to reach that region without exhaustively searching $\mathbb{H}$.

Standing on Prop. 1, Prop. 2 (*proof in suppl. B*) shows that for any given test point, a model trained on a suitably crafted traininglet is more likely to match the ideal Bayes prediction than any model trained on the full dataset. This strategy provides a practical path to realizing the low-capacity "sweet spot" guaranteed by Proposition 1.

**Prop.2 (Traininglet sufficiency).** For any test point $p$, there exist a traininglet $S_p \subsetneq S$ and $\Theta_p \in \mathbb{H}$ such that the classifier trained only on this traininglet $h_{\Theta_p, S_p} \in \mathbb{H}$ satisfies $\Pr\big[h_{\Theta_p, S_p}(p) = f_{\text{Bayes}}(p)\big] > \sup_{\Theta \in \mathbb{H}} \Pr\big[h_{\Theta, S}(p) = f_{\text{Bayes}}(p)\big]$. Here $h_{\Theta, S'}$ is the model obtained by fitting hypothesis $\Theta$ on dataset $S'$, and $f_{\text{Bayes}}$ denotes the Bayes-optimal classifier. Hence, isolating the low-capacity traininglet $S_p$ and training locally yields a predictor whose Bayes-matching probability strictly exceeds that of every full-data model, exactly the strategy embodied in our MiL.

*Prop. 2* guarantees an ideal, low-capacity traininglet exists for any test point. The central challenge, which MiL solves, is to practically construct this Bayes-optimal subset via label-aware structural knowledge fusion.

## 4 MiL: Overfitting-resistant, Explainable, and Reproducible

**MiL core: knowledge fusion for each query:** The key idea of MiL is to perform label-aware structural knowledge fusion by constructing a tailored traininglet for each query. Since finding the lexicographically optimal traininglet is NP-hard (Theorem 1), we introduce two practical heuristics to implement this fusion process: 1) Naïve Traininglet Construction (NTC): A straightforward approach effective for relatively large and clean datasets where local geometry is a reliable guide. 2) Precision Traininglet Construction (PTC): A robust, multi-stage framework designed to handle the complexity of true LH-Ps, including small, noisy, or imbalanced data.

Both heuristics achieve this *label-aware structural knowledge fusion* by actively leveraging label information to refine local neighborhoods: whether through multi-metric intersection (in NTC) or discriminative probing, training sanitization, meta-fusion, and precision pruning (in PTC). This ensures the resulting traininglet isolates the specific manifold structure relevant to the query, landing in the low-capacity "sweet-spot" guaranteed by our theory.

**MiL learns locally via a learninglet.** MiL then fits a deterministic RKHS model (e.g., SVM or variants), a *learninglet* on each traininglet. This local approach is inherently *overfitting-resistant;* instead of demanding a

single complex model to generalize globally, MiL fits many simple models to low-complexity, query-specific data. The use of an SVM or its variant, which avoid the high-order nested structure of deep networks and relies on deterministic convex optimization, ensures that each prediction is both *reproducible and interpretable.* This pipeline provides a constructive method for realizing the existential guarantee of Proposition 2, as shown in Fig.2 that compares MiL with traditional ML.

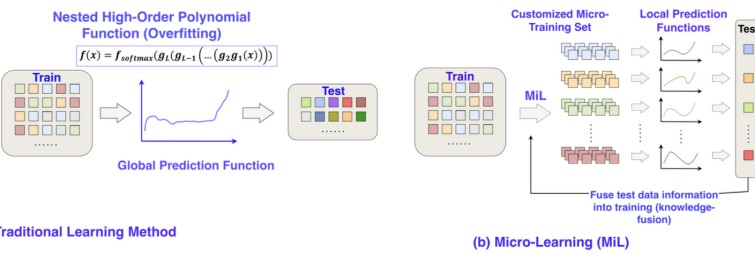

Figure 2: MiL learns a local predictor for each test point or batch, unlike traditional ML, which learns a single global function. This locality makes MiL inherently resistant to overfitting.

### Traininglet: Definition and Theory

**Definition 2** (Traininglet). *Let $X = \{(x_i, y_i)\}_{i=1}^n$ be the labeled training set and $\mathcal{Y} = \{1, \ldots, k\}$ its label set. Denote by $\mathrm{LHI}(\cdot) \in [0, 1]$ the Learning-Hard Index. For a query point $x'$, the* traininglet

$$\mathcal{T}_{x'} = \arg\min_{T \subseteq X} \big(\mathrm{LHI}(T), |T|\big) \quad s.t. \quad \mathcal{Y} \subseteq \{y_i : x_i \in T\}, \tag{3}$$

*minimizes the pair $\big(\mathrm{LHI}(T), |T|\big)$ lexicographically, first the lowest LHI, then the smallest size.*

Theorems 1 and 2, stated next, establish (i) traininglet decision problem (TRAININGLET-DEC) is NP-complete, implying the NP-hardness of finding the optimal traininglets, and (ii) the guaranteed existence of a low-capacity "sweet-spot" solution for every LH-P; detailed proofs are provided in *supplemental*.

**Theorem 1** (TRAININGLET-DEC is NP-complete). *Given a labeled set $X = \{(x_i, y_i)\}_{i=1}^n$, a set of required labels $\mathcal{Y}$, a budget $b \leq n$, and an LHI bound $\ell \in [0, 1]$, the problem of deciding*

$$\exists T \subseteq X : \ |T| \leq b, \ \mathrm{LHI}(T) \leq \ell, \ \mathcal{Y} \subseteq \{y_i : x_i \in T\} \qquad \text{(TRAININGLET-DEC)}$$

*is NP-complete, assuming $\mathrm{LHI}(\cdot)$ is computable in polynomial time.*

Before we move to Theorem 2 we need one technical fact. A sample is called *$\sigma$-noisy* if replacing its label by a fresh dummy label increases AMI by at least $\sigma > 0$. Removing such a point always lowers the overall LHI of a dataset $X$. Lemma 1 formalizes this monotonicity and is the key step used to build the low-complexity traininglets of Theorem 2.

**Lemma 1** (Removable-Noise Monotonicity; proof in *supplemental D*). *Let $Z \subseteq X$ be labeled data and $z = (x_z, y_z) \in Z$. If substituting a fresh dummy label $\perp$ for $y_z$ increases $\mathrm{AMI}$ by at least $\sigma > 0$, then $\mathrm{LHI}(Z \setminus \{z\}) \leq \mathrm{LHI}(Z) - \sigma$.*

For any LH-P and any batch of test queries we can always pick traininglets whose intrinsic complexity is strictly reduced, guaranteeing a move into the low-capacity regime promised by learning theory.

**Theorem 2** (Existence of Low-Complexity Traininglets). *Let $X$ be the training set of any learning-hard problem (LH-P) and let $x'_1, \ldots, x'_s$ be an arbitrary query batch. Then there exist traininglets $\mathcal{T}_{x'_1}, \ldots, \mathcal{T}_{x'_s} \subseteq X$ with $\min_j \mathrm{LHI}(\mathcal{T}_{x'_j}) < \mathrm{LHI}(X)$. Moreover, if $X$ contains a $\sigma$-noisy point in the sense of Lemma 1 (so $\sigma > 0$), the inequality sharpens to $\min_j \mathrm{LHI}(\mathcal{T}_{x'_j}) \leq \mathrm{LHI}(X) - \sigma$. proof in supplemental E.*

**NTC and PTC for knowledge fusion.** MiL employs two practical heuristics: naive traininglet construction (NTC) and precision traininglet construction (PTC) for knowledge fusion. NTC, the basis for our "Naive MiL" variant, creates a traininglet by intersecting small metric balls (e.g., Euclidean and correlation) and is effective primarily on relatively large, clean training data. PTC is a more robust heuristic for creating high-quality, low-LHI traininglets, especially for challenging data such as highly noisy, imbalanced, and small-sample data.

**Naïve traininglet construction (NTC).** NTC builds a traininglet for query $x'_i$ by intersecting several small metric balls so that retained points are simultaneously close to $x'_i$ in multiple geometric views of the data. Formally,

$$\mathcal{T}_{x'_i} = \bigcap_{j=1}^m \{x \in X : d_j(x, x'_i) < \varepsilon_j\}, \quad m \geq 2. \tag{4}$$

where $d_1$ and $d_2$ are typically Euclidean distance and Pearson correlation; a third view such as Wasserstein (images/audio) or cosine distance (sparse text) can be added when beneficial.

*Label rebalancing.* If the neighborhood $\mathcal{N}_\varepsilon(x_i')$ lacks any label $o$, we append the nearest sample of that label:

$$\mathcal{N}_\varepsilon'(x_i') = \mathcal{N}_\varepsilon(x_i') \cup \Big\{ \arg\min_{x \in \mathcal{S}_o} d_j(x, x_i') \Big\}. \tag{5}$$

*Limitations.* NTC presumes a large, clean dataset; the fixed radii $\varepsilon_j$ in equation 4 are rarely optimal, and noise within a ball can raise LHI even after rebalancing via equation 5. It either remains unknown how to select $\varepsilon_j$ for a batch of query points. These issues motivate the more robust Precision Traininglet Construction (PTC) introduced next (Fig .3).

**Precision Traininglet Construction (PTC) in Micro-Learning (MiL)**

Figure 3: Precision Traininglet Construction (PTC) in MiL consists of 4 steps: probing learning, training sanitization, meta-fusion, and precision pruning.

**Precision Traininglet Construction (PTC).** PTC operationalizes the guarantee of *Thm. 2*, identifying the low-capacity *sweet-spot* for a query via knowledge fusion. We denote the final traininglet produced by this algorithm as $\mathcal{T}_{x'}^{\mathrm{PTC}}$. It is constructed by forming the union $\mathcal{U}_{x'} = \bigcup_{j=1}^4 \mathcal{T}_{x'}^{(j)}$ of four meta-traininglets ($\mathcal{T}_{x'}^{(j)}$), and then pruning noisy samples in a 4-stage procedure: *Probing learning, Training sanitization, Meta-traininglet fusion, and precision pruning* (algorithm in *Suppl. G+*.)

*1. Probing learning.* We estimate the optimal neighborhood radius $k$ and batch size $z$ (queries processed jointly) by a Monte-Carlo (MC) search: over $M = 5 - 30$ random 80/20 splits of training data $X$. We evaluate every $(k, z)$ on Naive-MiL to maximize a target *D-index* Han et al. (2023) and retain only non-dominated pairs. Specifically, we random-split $X$ into an 80% *train-train* subset $X_{\mathrm{tr}}$ and a 20% *train-test* subset $X_{\mathrm{te}}$. Across a bounded grid of $(k, z)$ pairs, Naive-MiL predicts the labels of $X_{\mathrm{va}}$ from $X_{\mathrm{tr}}$.

We select the pair $(k^\star, z^\star)$ that maximizes $D$ (D-index) in each search. $(k^\star, z^\star) = \arg\max_{k,z} D_{\text{Naive-MiL}}\big(X_{\mathrm{tr}}, X_{\mathrm{te}}, k, z\big)$. The D-index, an interpretable ML assessment score bounded by $(0, 2]$, is defined for a $K$-class problem as $D = \frac{1}{K} \sum_{i=1}^K \Big[ \log_2\big(1 + \alpha_i\big) + \log_2\Big(1 + \frac{s_i + p_i}{2}\Big) \Big]$, where $\alpha_i$, $s_i$, and $p_i$ denote the accuracy, sensitivity, and specificity per class, respectively.

*2. Training sanitization.* Running Naive-MiL with $(k^\star, z^\star)$ on training data $X$ yields a deterministic prediction $\hat{y}_i$ for every sample $(x_i, y_i)$. This partitions $X$ into correctly and incorrectly predicted subsets ("good guys" and "bad guys"):

$$\mathcal{G} = \{x_i \in X \mid \hat{y}_i = y_i\}, \qquad \mathcal{B} = \{x_i \in X \mid \hat{y}_i \neq y_i\}. \tag{6}$$

*Noise pruning.* For each $x_b \in \mathcal{B}$ we remove both the error point and its $\epsilon$-ball neighbors $\mathcal{N}_\epsilon(x_b)$: $X^{\mathrm{clean}} = X \setminus \big(\mathcal{B} \cup \bigcup_{x_b \in \mathcal{B}} \mathcal{N}_\epsilon(x_b)\big)$. By Lemma 1, deleting each $\epsilon$-ball lowers the LHI by at least $\sigma > 0$; hence $\mathrm{LHI}(X^{\mathrm{clean}}) \leq \mathrm{LHI}(X) - \sigma$, moving the data toward the low-capacity "sweet-spot" required by Theorem 2.

The sanitization process prunes 18–41% of the training data across our five benchmarks, reducing the LHI by 6–20%. To prevent data loss for rare classes, a *minority-class safeguard* re-introduces the nearest 'good' instance ($\mathcal{G}$) for any class that is fully eliminated. This yields a lean, noise-free, and label-complete dataset for the subsequent PTC steps.

*3. Meta-traininglet fusion.* For every query $x_i'$, we fuse four meta-traininglets $\{\mathcal{T}_{x_i'}^{(j)}\}_{j=1}^4$ into a single, label-complete union to capture complementary knowledge structural views. $\mathcal{U}_{x_i'} = \bigcup_{j=1}^4 \mathcal{T}_{x_i'}^{(j)}$. This union (i) contains every class, (ii) is at most $3k' + |\mathcal{G}|$ points, and (iii) lowers LHI, and provide a compact, well-balanced basis for

PTC. The $1^{st}$ meta-traininglet $\mathcal{T}_{x_i'}^{(1)}$ is *a local ball* capturing geometric proximity. It is created using NTC with the optimal neighbor size $k'$ in the cleaned training data: $\mathcal{T}_{x_i'}^{(1)} = \text{NTC}(x_i', k', X^{\text{clean}})$.

The $2^{nd}$ and $3^{rd}$ meta-traininglets, $\mathcal{T}_{x_i'}^{(2)}$ and $\mathcal{T}_{x_i'}^{(3)}$, are *1-hop and 2-hop transfers*, injecting first-order semantic context and adding broader manifold structure, respectively. They are generated by performing nearest-neighbor search (NNS) on $\mathcal{G}$ (the set of "good guys" from training sanitization) to obtain each point's first- and second-closest neighbors, $\mathcal{N}_1(x_i')$ and $\mathcal{N}_2(x_i')$, and then merging their traininglets:

$$\mathcal{T}_{x_i'}^{(2)} = \bigcup_{x_i' \in \mathcal{G}} \mathcal{T}_{\mathcal{N}_1(x_i)}, \qquad \mathcal{T}_{x_i'}^{(3)} = \bigcup_{x_i' \in \mathcal{G}} \mathcal{T}_{\mathcal{N}_2(x_i')} \tag{7}$$

The $4^{th}$ meta-traininglet $\mathcal{T}_{x_i'}^{(4)}$ is *a random anchor* plugging residual topology gaps. It is formed by randomly selecting a "good guy" $x_g \in \mathcal{G}$ and combining it with its traininglet, $\mathcal{T}_{x_i'}^{(4)} = \mathcal{T}_{x_g}$.

*4. Precision pruning.* Remove any point within a neighbor radius of $B$ ('bad guys') to obtain the *final traininglet* $\mathcal{T}_{x_i'}^{\text{PTC}} = \mathcal{U}_{x_i'} \setminus \bigcup_{b \in B} \mathcal{N}(b)$. This shrinks LHI by removing additional noise or outliers.

**Why PTC works.** PTC knowledge fusion has theoretical guarantees. *Stage 1* aligns neighborhoods with labels; by *Prop. 1*, it lands in an r-ball of minimal local Rademacher radius, and *Prop. 2* guarantees that the resulting traininglet outperforms any fulldata model, tightening the generalization bound; *Stage 2* excises high-entropy samples and their neighbors, lowering the empirical VC dimension; *Stage 3* re-establishes full label coverage, guaranteeing an *informed traininglet* (Thm. 2); *Stage 4* removes outliers, tightening the generalization bound to $\mathcal{O}(1/\sqrt{|T|})$ ($T$: final traininglet size.)

*Prop.3* (*suppl. F*) resolves *distribution shift* by strictly contracting the training–test total-variation distance, creating a query-aligned local distribution that ensures reliable generalization on LH-Ps.

**Prop. 3 (PTC contracts the training–test gap).** Let $P_{\text{tr}}$ and $P_{\text{te}}$ denote the training and test distributions of an LH-P. After applying PTC within the MiL pipeline, the resulting distribution $P_{\text{PTC}}$ satisfies the strict total variation contraction, i.e., $P_{\text{tr}} \xrightarrow{\text{PTC}} P_{\text{te}}$. $\left\| P_{\text{PTC}} - P_{\text{te}} \right\|_{\text{TV}} < \left\| P_{\text{tr}} - P_{\text{te}} \right\|_{\text{TV}}$.

**Explainability and reproducibility of MiL's learninglet.** MiL's local learner ( learninglet) is a multiclass SVM, a choice that ensures both reproducibility and interpretability. Reproducibility is guaranteed via deterministic convex optimization. Interpretability stems from the SVM's decision function for any pair of classes '$i$' and '$k$': $f_{ik}(x) = \sum_j \alpha_j^{ik} y_j^{ik} K(x_j^{ik}, x) + b_{ik}$. This formulation provides a transparent, instance-based explanation, as the prediction is a direct function of the query's kernelized similarity to a few support vectors from the tailored traininglet.

**SVM-micro-CNN-let (SC-let).** To endow MiL with representation learning while retaining the determinism of large-margin theory, we replace each SVM with an SVM-micro-CNN-let (SC-let). This *learninglet* uses a compact CNN module (e.g., a 3x3 CNN-let, ResNet-let or even a ResNet) to learn a feature map, which is then fed to a linear SVM head. This hybrid design retains the reproducibility and RKHS explainability of classical SVMs while gaining the expressive power of CNNs to disentangle complex local patterns from small traininglets. Crucially, for image data, all nearest neighbor searches for traininglet construction are performed in a pretrained CNN-mapped feature space, ensuring comparisons are based on semantic similarity rather than misleading pixel-by-pixel calculations. Note NTC is recommended for high-dimensional image data for cost.

**MiL Complexity.** MiL complexity model represents a deliberate trade-off, making it a practical for high-stakes, small/mid-sized LH-Ps. Its primary limitation is the significant, one-time offline preprocessing cost of $\mathcal{O}(Mn^2p)$ in the PTC phase ($M$: Number of Monte-Carlo (MC) draws) While this can be expensive for very large datasets, this upfront investment enables highly efficient and embarrassingly parallel online inference for each query. Furthermore, MiL's memory complexity is only $\mathcal{O}(np)$, a significant advantage over methods requiring prohibitive $\mathcal{O}(n^2)$ storage like kernel SVMs. This two-phase design: a high but justifiable one-time cost for fast, scalable, and memory-efficient *learninglet inference* makes MiL feasible for challenging problems where other powerful methods are often computationally intractable.

# 5 RESULTS: MASTERING LH-PS WITH MIL

**Baselines.** We evaluate MiL's performance across the five benchmarks in Table 1. MiL is compared with *15 baselines* chosen to cover the three dominant paradigms for small or noisy data: *(i) Classical non-parametrics*: SVM, Random Forest, Extra-Trees, Naïve Bayes, DNN; *(ii) Mainstream static DL*: CNN, LSTM, GRU, Bi-LSTM, Bi-GRU; *(iii) Hybrid/capsule refinements*: Conv-LSTM, Conv-GRU, Conv-BiLSTM, Conv-BiGRU, CapsNet (LeCun et al., 2015; Sabour et al., 2017; Cho et al., 2014). These paradigms isolate the efficacy of local knowledge fusion. MiL (84.3% F1) surpasses domain-specific IRMAS SOTA (68.5% Yu et al., 2020)); for the novel medical tasks, standard DL represents the effective state-of-the-art.

Online TTA methods (e.g., Tent, T3A) are omitted: they assume large, stationary target batches and fixed feature extractors, assumptions that fail in LH-Ps where queries are single and highly shifted. Hyper-parameters are tuned in study by nested grid search (*Suppl. L*).

**MiL wins statistically.** We report mean over five repeated 5-fold CV runs (IRMAS, CASIA, COVID-19, Ovarian) and a single 10-fold CV (SAVEE), following established practice on small-sample speech corpora.

Table 2: Performance of MiL on five benchmarks

| Dataset | D-index | Acc | Sen | Prec | F1 |
|---------|---------|-----|-----|------|-----|
| IRMAS | 1.8162 | 0.8431 | 0.8387 | 0.8449 | 0.8431 |
| CASIA | 1.7949 | 0.8283 | 0.8314 | 0.8297 | 0.8283 |
| SAVEE | 1.7015 | 0.7625 | 0.7365 | 0.7458 | 0.7625 |
| COVID19 | 1.9424 | 0.9544 | 0.9644 | 0.9632 | 0.9544 |
| Ovarian | 1.7939 | 0.9815 | 1.0000 | 0.9811 | 0.9815 |

Table 2 reports MiL's performance across five benchmarks, showing it surpasses both classical ML and DL baselines on every dataset. A one-tailed Mann–Whitney $U$-test on the composite *Metric-integrated Lift* score confirms this superiority: MiL's median of 0.97 (95% CI: 0.95-1.00) significantly exceeds the 0.77 (95% CI: 0.76-0.84) of the best DL baseline ($U = 23, p = 1.6 \times 10^{-2}, \delta_{\text{Cliff}} = 0.84$). A more granular test on all 35 raw metric values yields an even more dominant result ($U = 1082, p < 2 \times 10^{-8}, \delta_{\text{Cliff}} \approx 0.77$), indicating an 89% probability that MiL outperforms the DL model on any given metric ($P(\text{ours} > \text{DL}) \approx 0.89$). MiL therefore statistically outperforms every convolutional, recurrent, and capsule DL model, providing concise, effect-size-centered evidence of its architectural superiority. Similarly, A battery of 25 Bonferroni-adjusted Mann-Whitney tests confirms MiL's complete stochastic dominance over classical ML baselines, with its knowledge-fused train-

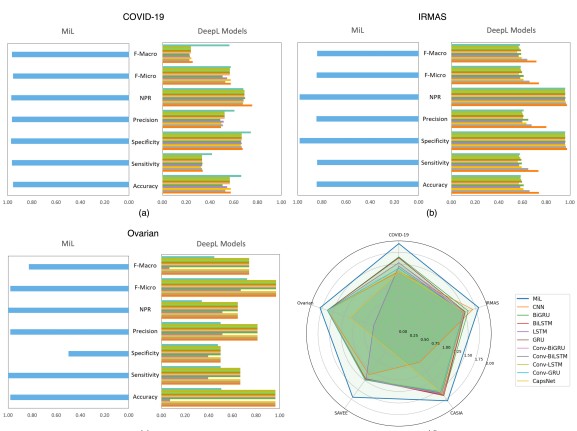

Figure 4: MiL demonstrates superior performance over 11 DL models. *(a-c)* Comparison on key metrics for the COVID-19, IRMAS, and Ovarian datasets. *(d)* D-index values across all 5 benchmarks, confirming MiL's superiority, especially on small-sample tasks.

inglets maintaining a solid decision boundary even under extreme imbalance where rivals falter (see *Suppl. J*)

**MiL vs DL.** Fig 4 contrasts MiL with 11 DL baselines across our 3 benchmarks (results of 5 benchmarks in *Suppl. H*). The results demonstrate MiL's superiority: even in its naïve form on the larger IRMAS dataset, MiL tops every DL model. This performance gap widens dramatically on small-sample tasks like COVID-19 and Ovarian, where DL struggle to generalize. Beyond raw accuracy, MiL provides advantages that DL cannot: deterministic training, transparent decision boundaries, and inherent resistance to overfitting.

**MiL vs meta-learning, SAM, pretraining, and LNN.** Across the five benchmarks, MiL consistently outperforms meta-learning baselines: *ProtoNet* and *MAML* (*Suppl. M*), raising average accuracy from 67.8% to 87.4% (and F1 from 64.6% to 87.4%), with per-dataset accuracy gains ranging from about 7 to 33.7 percentage points and consistent increases in D-index, sensitivity, and precision. They both have poor performance on the small-sample data: *COVID-19* and *Ovarian*. A paired $t$-test across five benchmarks confirms MiL significantly outperforms the best baseline ($p \approx 0.01$) with a large effect size. MiL excels by building a clean, query-specific local model, avoiding the single, noise-sensitive global model used by meta-learners. Similarly, MiL statistically outperforms SAM across the benchmarks, as SAM relies on a single, noise-sensitive global model and with poor reproducibility (*Suppl. P*). *Suppl. Q and U* also show MiL statistically outperforms pretraining and LNN (*liquid neural networks*) models on LH-P benchmarks.

**Traininglet visualization.** Fig. 5 contrasts the highly entangled global datasets (baselines shown in Fig. 1(b) and *Suppl. G*) with MiL's tailored traininglets for IRMAS, COVID-19, and Ovarian queries. For both single-sample inference and optimized batches (batch size $z : 212, 32, 25$ respectively), the traininglets exhibit exceptional class separability. Quantitatively, the LHI plummets from the intractable global baseline ($\geq 0.79$) to a solvable local regime ($\leq 0.26$). This drastic reduction empirically validates *Thm 2* (strict complexity reduction) and serves as the physical realization of *Prop. 1's* theoretical *sweet-spot*. By isolating these simplified, query-aligned sub-distributions, MiL effectively contracts the training-test total variation distance (*Prop. 3*), converting a globally hard problem into a sequence of locally trivial ones.

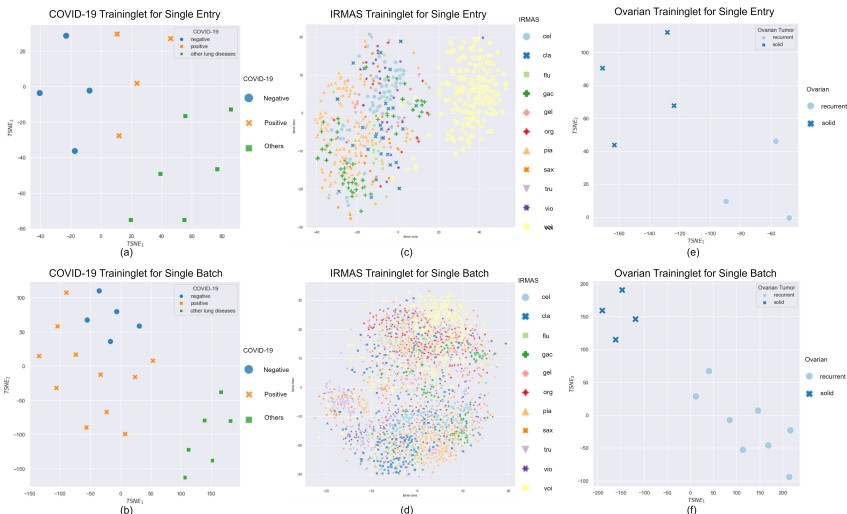

Figure 5: High-quality, separable traininglets constructed by MiL for the COVID-19 (a, b), IRMAS (c, d), and Ovarian (e, f) datasets. Examples for both single queries (a, c, e) and test batches (b, d, f) demonstrate exceptional class separability, validating the effectiveness of label-aware structural knowledge fusion.

**Ablation studies.** Our MiL ablation study strongly supports the key design choices of the four-stage PTC pipeline. For example, for the COVID-19 data ablation, a paired Wilcoxon signed-rank analysis across the eight evaluation metrics (*Suppl. I*) shows that removing any single PTC component leads to a statistically significant drop in performance compared to the full *MiL* pipeline (all $p = 0.003906 < 0.0125$ after Bonferroni correction over four tests), statistically validating that all four PTC stages are integral and complementary, with each contributing a significant and non-redundant performance gain. We omit ablation for IRMAS and Ovarian data as their NTC method lacks separable pipeline stages.

# 6   DISCUSSION AND CONCLUSION

We formalized LH-Ps that defeat most learners yet become solvable once latent knowledge fused, and introduced MiL for LH-Ps. MiL builds a well-tailored *traininglet* for each query by fusing relevant label-aware and structural knowledge upon which a deterministic learninglet infers a local decision function, to achieve overfitting-resistant, reproducible, and interpretable learning. MiL mitigates distribution shift and adversarial attacks via dynamic, query-aligned traininglets, statistically outperforming SOTA baselines across benchmarks.

**Limitation:** *1. Scalability.* MiL's primary limitation is the complexity of its PTC phase: $\mathcal{O}(Mn^2p)$, which restricts its application to small- and mid-sized LH-Ps. While potential solutions exist, they involve significant trade-offs: larger traininglets risk compromising MiL's overfitting resistance, while intensive GPU acceleration may sacrifice the deterministic reproducibility that is a key feature of our *learninglet*. Future work could explore scalable approximation algorithms for the NP-hard *traininglet* selection for this and trainglet resuse techniques. *2. MiL failure.* MiL's effectiveness is also limited on *high-dimensional, noisy data* like vectorized text. A preliminary study on a large SEC 8K dataset ($\sim$18k samples, $\sim$ 1k features, *Suppl. K*) showed only marginal improvement over existing models with more computing. We hypothesize that high vectorization noise degrades effective knowledge fusion in traininglets construction even if dimension-reduction-de-noising is employed. It implies more customized de-noising or context retrieval is needed for such data for MiL.

**Extension and Future:** We applied the proposed *SC-let* to CIFAR-100 data, where the *learninglet* is implemented as a ResNet and each *traininglet* is calculated with NTC for each test point. We achieved 80.24% accuracy (*Suppl. V*) where each traininglet with only 50 entries, suggesting that MiL can be an effective model for large image data, largely retaining its benefits of reproducibility and overfitting-resistance. A promising direction for future work is to design more compact or inherently interpretable *learning-lets* to ensure MiL's explainability is not compromised when scaling to such complex vision tasks, as well as extending MiL, LHI, *traininglets* and *learning-lets* to Learning-Hard regression problems (*Suppl. T*)

We aim to scale MiL to large-scale LH-Ps by integrating approximate nearest-neighbor search (e.g., *Annoy*) and designing optimized *learning-lets* for big data. Crucially, we aim to quantify the inherent trade-offs between improved scalability and MiL's core advantages: its overfitting-resistance, reproducibility, and explainability. Striking this balance is essential to establish trustworthy MiL for high-stakes AI. **code and data:** https://anonymous.4open.science/r/iclr26-anon-code-9DB6/.

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
