# Micro-Learning for Learning-Hard Problems,
## *Supplemental*

# 1 Supplemental A

## Proposition 1

**Proposition 1 (Low-capacity witness).** *For any LH-P with hypothesis class $\mathbb{H}$ and any radius $r > 0$, there exists a model $f^\star \in \mathbb{H}$ such that*

$$\mathcal{R}_n\big(\mathcal{F}_r(f^\star)\big) = \inf_{f \in \mathbb{H}} \mathcal{R}_n\big(\mathcal{F}_r(f)\big), \tag{1}$$

*meaning $f^\star$ minimizes the local Rademacher complexity over $\mathbb{H}$. Consequently, $f^\star$ and every model within its $r$-ball neighborhood enjoy the tightest generalization bound available in the entire hypothesis space.*

*Proof.* Fix the sample $S = \{z_1, \ldots, z_n\}$ and endow $\mathbb{H}$ with the empirical $\ell_2$ metric $d(f,g) = \big(\frac{1}{n}\sum_{i=1}^n (f(z_i) - g(z_i))^2\big)^{1/2}$.

**(i) Compactness.** Assume (a) uniform boundedness $|h(z_i)| \leq B < \infty$ for all $h \in \mathbb{H}$ and (b) $\mathbb{H}$ is closed in $d$ (these hold, e.g., for RKHS balls and bounded-weight networks). The evaluation map $\mathbf{v} : \mathbb{H} \to [-B, B]^n$, $h \mapsto (h(z_1), \ldots, h(z_n))$ is an isometry, so its image is a closed subset of the compact cube $[-B, B]^n$ and therefore compact; hence $(\mathbb{H}, d)$ is compact.

**(ii) Continuity.** Let $G(f) = \mathcal{R}_n\big(\mathcal{F}_r(f)\big)$. For any $f_1, f_2 \in \mathbb{H}$,

$$|G(f_1) - G(f_2)| \leq d(f_1, f_2),$$

because the support function of an $\ell_2$-ball is 1-Lipschitz (Ledoux and Talagrand, 1991, Prop.2.1). Thus $G$ is continuous.

**(iii) Extreme-value argument.** A continuous real-valued function on a compact set attains its infimum; therefore $\exists f^\star \in \mathbb{H}$ such that $G(f^\star) = \inf_{f \in \mathbb{H}} G(f)$, proving (1). $\qquad\square$

## References

Ledoux, M., and Talagrand, M. 1991. *Probability in Banach Spaces: Isoperimetry and Processes.* In Ergebnisse der Mathematik und ihrer Grenzgebiete. Springer-Verlag Berlin Heidelberg.

## 2    Supplemental B

**Proposition 2 (Traininglet sufficiency).**   *For any test point $p$, there exist a traininglet $S_p \subsetneq S$ and a hypothesis $\Theta_p \in \mathbb{H}$ such that the classifier trained only on this traininglet, $h_{\Theta_p, S_p}$, satisfies,*

$$\Pr[h_{\Theta_p, S_p}(p) = f_{\text{Bayes}}(p)] \; > \; \sup_{\Theta \in \mathbb{H}} \Pr[h_{\Theta, S}(p) = f_{\text{Bayes}}(p)] \tag{2}$$

*Proof.* The proof follows from a standard leave-one-out (LOO) stability argument. Assume the learning algorithm is uniformly $\beta$-stable Bousquet and Elisseeff (2002), so the standard LOO bound applies. Let $I(z; p)$ be the influence of a point $z \in S$ on the misclassification of $p$. Because the best full-data success probability $q < 1$, we have $\sum_{z \in S} I(z; p) > 0$. Let $z_{\max} = \arg\max_{z \in S} I(z; p)$, for which $I(z_{\max}; p) > 0$.

Let $S_p = S \setminus \{z_{\max}\}$. Stability yields the bound:

$$\Pr[h_{\Theta, S_p}(p) \neq y_p] \leq \Pr[h_{\Theta, S}(p) \neq y_p] - I(z_{\max}; p).$$

This implies the success probability on $S_p$ is strictly larger than on $S$ by at least $I(z_{\max}; p) > 0$. By choosing $\Theta_p$ as the empirical risk minimizer on $S_p$, we satisfy the strict inequality in (2).    $\square$

## References

Bousquet, O., and Elisseeff, A. 2002. Stability and Generalization. *Journal of Machine Learning Research* 2: 499–526.

# 3 Supplemental C

**A Rigorous Proof of the NP-Completeness of the Traininglet Decision Problem**

### Abstract

We present a formal proof that finding an optimal "traininglet" is an NP-hard problem. We establish this by proving that the corresponding decision problem, TRAININGLET-DEC, is NP-complete. The proof of NP-hardness is achieved via a polynomial-time reduction from a bounded-occurrence version of 3-Satisfiability, (E3,3)-SAT. The reduction's key innovation is the use of a large set of "anchor points" to stabilize the LHI calculation. This allows for a robust "LHI budget" ($\ell < \epsilon$) that can distinguish between the minor feature collisions inherent in a valid construction and the major collisions that signify a logical contradiction, which we formalize in a quantitative supporting lemma.

## Introduction and Background

Modern AI increasingly confronts *high-complexity data* whose non-linearity, noise, imbalance, or scarcity overwhelm conventional models. We argue that many such tasks belong to a distinct, under-studied class we call *Learning-Hard Problems (LH-Ps)*. An LH-P is characterized by two simultaneous properties:

1. *Near-universal failure* – almost every model in a broad hypothesis space delivers only mediocre or poor performance;

2. *Latent solvability* – there exists at least one model that can achieve high-quality results once appropriate domain knowledge is fused into the training process.

To diagnose such problems a priori, we introduce a data-centric metric, the Learning-Hard Index (LHI).

**Definition 1** (Learning-Hard Index (LHI))**.** Let $X = \{(x_i, y_i)\}_{i=1}^{m}$ be a labeled dataset. We first obtain a locality-preserving embedding $X_r = f_{\mathrm{dm}}(X)$ (e.g., t-SNE or UMAP), then cluster $X_r$ (e.g., with $k$-means) to produce pseudolabels $y_{\mathrm{p}i}$. The *Learning-Hard Index* is

$$\mathrm{LHI}(X) = 1 - \mathrm{AMI}(y, y_p) \tag{3}$$

where $\mathrm{AMI}(y, y_p)$ is the Adjusted Mutual Information between the true labels $y$ and the pseudo-labels $y_p$. An LHI near 1 indicates that the geometric structure of the data is in high disagreement with the label structure, signaling a difficult learning task.

The core insight for solving LH-Ps is that a global model is often suboptimal. Instead, the solution lies in finding a small, well-chosen subset of data, which we formalize as a "traininglet." This is the central object of study in the Micro-Learning (MiL) framework.

**Definition 2** (Traininglet)**.** Let $X = \{(x_i, y_i)\}_{i=1}^{n}$ be the training set and $\mathcal{Y}$ its label set. A *traininglet* $\mathcal{T} \subseteq X$ is a subset that is optimal with respect to three criteria:

(i) **Low Complexity:** It should have a low intrinsic complexity, measured by $\mathrm{LHI}(\mathcal{T})$.

(ii) **Small Size:** It should be as small as possible, measured by $|\mathcal{T}|$.

(iii) **Full Coverage:** It must contain at least one example for every class in the original problem, i.e., $\mathcal{Y} \subseteq \{y_i : x_i \in \mathcal{T}\}$.

The optimization problem, OPTIMAL-TRAININGLET, seeks to find the subset $T \subseteq X$ that minimizes the pair $(\text{LHI}(T), |T|)$ lexicographically while satisfying the full coverage constraint.

This mini-paper addresses the fundamental computational question: how hard is it to find an optimal traininglet? We answer this by proving that the problem is NP-hard. This result provides a crucial theoretical foundation for the MiL framework, justifying the need for heuristic or approximation-based methods to find effective traininglets in practice.

## NP-Completeness of Finding an Optimal Traininglet

We now prove that finding an optimal traininglet is computationally intractable by analyzing the complexity of its decision version, TRAININGLET-DEC.

**Definition 3** (TRAININGLET-DEC). Given a dataset $X = \{(x_i, y_i)\}_{i=1}^{N}$, a set of required labels $\mathcal{Y}$, an integer budget $b$, and a complexity bound $\ell \in [0, 1]$, does there exist a subset $T \subseteq X$ such that:

(i) $|T| \leq b$ (Budget constraint),

(ii) $\mathcal{Y} \subseteq \{y_i : x_i \in T\}$ (Label coverage constraint), and

(iii) $\text{LHI}(T) \leq \ell$ (LHI constraint)?

**Theorem 1.** *The decision problem* TRAININGLET-DEC *is NP-complete.*

**Theorem 2.** *The optimization problem* OPTIMAL-TRAININGLET *is NP-hard.*

The second theorem follows directly from the first. Thus, our main task is to prove Theorem 1. Our idea flow of Theorem 1 proof is sketched in Figure 1.

The proof proceeds in a four–step cycle: **(i)** A tight budget, together with unique variable- and clause-identity labels, forces every feasible solution to contain *exactly one* point per label. **(ii)** The LHI bound is engineered so that any logically invalid choice provokes a feature–label collision. **(iii)** A large block of orthogonal anchor points stabilises LHI, rendering small perturbations negligible. **(iv)** Harmless cross–clause reuse perturbs the index by only $O\big(1/(N'\log N')\big)$, whereas a genuine variable–clause collision changes it by $\Theta(1/N')$. Choosing $\epsilon$ in this gap separates *YES* from *NO* instances, completing the reduction.

## Proof of Theorem 1

The proof has two parts: establishing membership in NP and proving NP-hardness.

### Part 1: NP Membership

A candidate solution $T \subseteq X$ can be verified in polynomial time: (1) Check $|T| \leq b$ in $O(|T|)$ time. (2) Check $\mathcal{Y} \subseteq \{y_i : x_i \in T\}$ in $O(|T| + |\mathcal{Y}|)$ time. (3) Compute $\text{LHI}(T)$ and check if it is $\leq \ell$, which is assumed to be

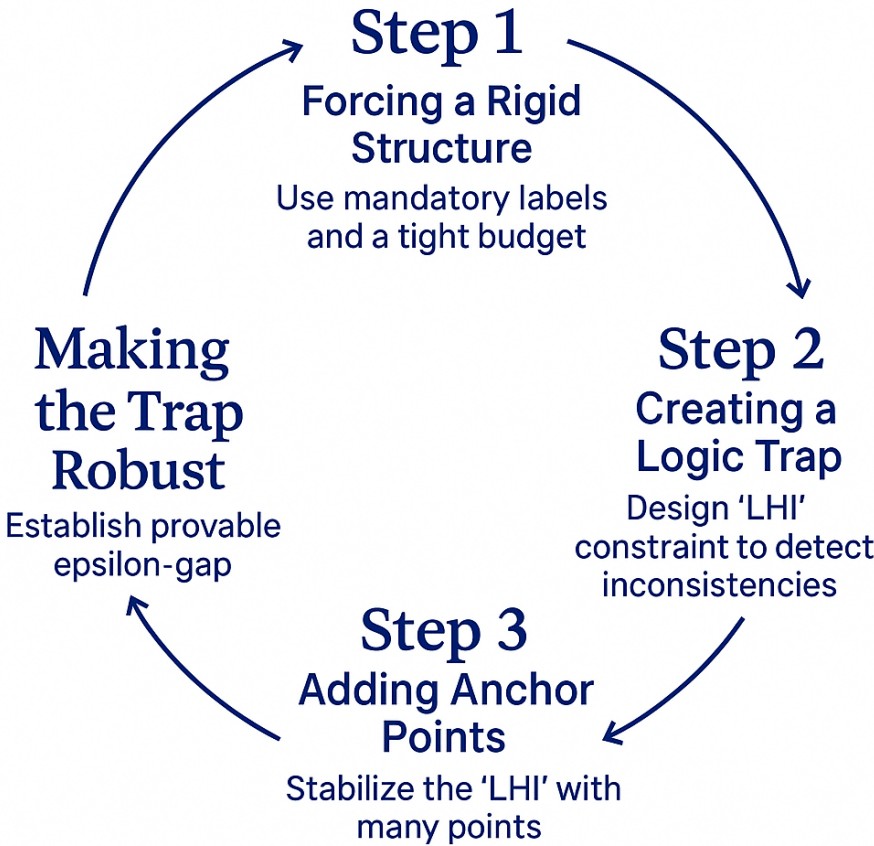

Figure 1: Idea-flow overview of the NP-completeness proof. Four–step reduction outline: budget + unique labels enforce one-point-per-label; the LHI bound turns logical errors into feature collisions; orthogonal anchors stabilise LHI; a calibrated $\epsilon$ exploits the $O(1/(N' \log N'))$ vs. $\Theta(1/N')$ gap to distinguish valid from invalid solutions.

a polynomial-time operation.[1] Since verification is polynomial, TRAININGLET-DEC is in NP.

**Part 2: NP-Hardness**

We prove NP-hardness via a polynomial-time reduction from (E3,3)-SAT, a known NP-complete fragment of 3-SAT where every variable appears in at most three clauses.[2]

**Idea Flow of the Reduction.** The core challenge is that the three constraints of TRAININGLET-DEC are difficult to balance. Our strategy overcomes this with a three-part construction:

1. **Forcing a Rigid Structure.** We introduce mandatory, unique labels for each variable choice and each clause condition. By setting the budget $b$ to be exactly the total number of these mandatory labels, we force any valid solution to have a precise, predictable structure. This is formalized in Lemma 1.

2. **Creating a Logic Trap.** We use the 'LHI' constraint as a logic gate. We design the features of our "gadget" points such that a logically inconsistent choice creates a feature collision between points with different mandatory labels.

3. **Making the Trap Robust with an $\epsilon$-Gap.** A perfect 'LHI=0' trap is brittle. We solve this by relaxing the bound to LHI $< \epsilon$. We add a large number of "anchor points" to the dataset to stabilize the LHI calculation. This creates a provable "epsilon-gap": logically consistent solutions have a tiny, near-zero LHI, while logically inconsistent solutions have a much larger LHI. This is formalized in Lemma 2.

**The Reduction (Formal Details).** Given a (E3,3)-SAT instance $\phi$ with $n$ variables and $m$ clauses, we construct an instance of TRAININGLET-DEC.

**1. The Gadgets (Data Points in $X$):**

- **Variable Gadgets:** For each variable $x_i$, create two points: $P_{i,T} = (f_{i,T}, \eta_i)$ and $P_{i,F} = (f_{i,F}, \eta_i)$. They have unique, orthogonal base feature vectors and share a mandatory label $\eta_i$.

- **Clause Gadgets:** For each clause $C_j = (L_1 \lor L_2 \lor L_3)$, create three points, $H_{j,k}$, all with mandatory label $c_j$. The feature of $H_{j,k}$ is identical to the feature of the variable point that would make the literal **false** (i.e., $f_{i,F}$ for $x_i$ and $f_{i,T}$ for $\neg x_i$).

- **LHI Anchor Gadget:** Add $N = (n+m)^3$ "anchor" points, $A_k = (g_k, \delta_k)$, each with a unique orthogonal feature and unique label (where any polynomial in $n$ and $m$ with exponent $c > 2$ suffices). Each anchor feature $g_k$ lives in its own new coordinate disjoint from all $f_{i,T}, f_{i,F}$, so no anchor ever collides with a gadget feature.

**2. Parameters $(\mathcal{Y}, b, \ell)$:**

- **Required Labels $(\mathcal{Y})$:** The set of all $N + n + m$ mandatory labels: $\mathcal{Y} = \{\eta_i\} \cup \{c_j\} \cup \{\delta_k\}$.

---

[1] For instance, a pipeline of Barnes-Hut t-SNE ($O(|T| \log |T|)$), DBSCAN with a k-d tree ($O(|T| \log |T|)$), and AMI calculation ($O(|T|)$) is polynomial.

[2] See Tovey (1984) for NP-completeness of (E3,3)-SAT. For a broader survey, see e.g., Berman et al. (2003).

- **Budget** ($b$): $b = N + n + m$. For clarity, we denote this total size by $N'$ throughout the proof.

- **LHI Bound** ($\ell$): $\ell = \epsilon$, a small positive constant defined in Lemma 2.

**Proof of Equivalence.**

**Lemma 1** (Structure of Any Valid Solution). *Any valid traininglet $T$ for the constructed instance must have size exactly $N'$ and consist of: all $N$ anchor points, exactly one variable point for each of the $n$ variables, and exactly one satisfaction point for each of the $m$ clauses.*

*Proof.* The label coverage for $\mathcal{Y}$ requires at least one point for each of the $N'$ mandatory labels, setting a minimum size of $N'$. The budget $b$ is set to exactly this minimum. By the pigeonhole principle, the mapping from mandatory labels to points in $T$ must be a bijection. This forces the rigid structure. $\square$

**Lemma 2** (LHI Epsilon-Gap). *Let $T_{good}$ be a traininglet constructed from a satisfying assignment, and $T_{bad}$ be a traininglet containing at least one variable-clause feature collision. For sufficiently large $N'$, there exists an $\epsilon$ such that $\mathrm{LHI}(T_{good}) < \epsilon$ and $\mathrm{LHI}(T_{bad}) > \epsilon$.*

*Proof.* The LHI is $1 - \mathrm{AMI}$. (1) A "good" solution's only impurity comes from cross-clause collisions, where at most 3 points share a feature. This creates a small LHI perturbation of order $O(1/(N' \log N'))$. (2) A "bad" solution contains a major variable-clause collision, merging two distinct mandatory label groups. This creates a larger LHI perturbation of order $\Theta(1/N')$.[3] For large $N'$, a gap is guaranteed. We can choose $\epsilon := 1/(4N')$. $\square$

**() A satisfying assignment for $\phi$ implies a YES-instance for Traininglet-Dec.** Assume a satisfying assignment exists. Construct $T$ by selecting all $N$ anchors, the $n$ variable points matching the assignment, and for each clause, one satisfaction point for a satisfying literal. $|T| = N' = b$ and all labels are covered. The only LHI impurity is from bounded cross-clause collisions, so by Lemma 2, $\mathrm{LHI}(T) < \epsilon$. A valid traininglet exists.

**() A YES-instance for Traininglet-Dec implies a satisfying assignment for $\phi$.** Assume a valid traininglet $T$ exists. By Lemma 1, we know its structure.

1. **Construct Assignment:** For each variable $i$, since exactly one of $P_{i,T}$ or $P_{i,F}$ is in $T$, define the assignment accordingly.

2. **Verify Assignment:** Assume for contradiction a clause $C_j$ is not satisfied. This means all its literals are false. Let $L_{j,k} = x_i$ be one such literal. Since $x_i$ is false, $P_{i,F} \in T$. The satisfaction point for this literal, $H_{j,k}$, has feature $f_{i,F}$. Thus, $T$ would contain both $P_{i,F}$ (label $\eta_i$) and $H_{j,k}$ (label $c_j$), a major collision. By Lemma 2, this means $\mathrm{LHI}(T) > \epsilon$, contradicting that $T$ is a valid solution. The same argument holds for negative literals. The contradiction implies all clauses must be satisfied.

The assignment is satisfying.

**Conclusion** We have shown that a satisfying assignment for a (E3,3)-SAT formula exists if and only if a valid traininglet exists for the instance we constructed. The transformation is polynomial. Therefore, Traininglet-Dec is NP-hard. Since it is also in NP, it is NP-complete.

---

[3]See Vinh, Epps, and Bailey (2010) for bounds on the change in mutual information.

# References

Berman, P., Karpinski, M., & Scott, A. D. (2003). Approximation hardness of bounded degree MIN-CSP and MIN-BISECTION. *Electronic Colloquium on Computational Complexity (ECCC)*, 10(009).

Tovey, C. A. (1984). A simplified NP-complete satisfiability problem. *Discrete Applied Mathematics*, 8(1), 85-89.

Vinh, N. X., Epps, J., & Bailey, J. (2010). Information theoretic measures for clusterings comparison: Variants, properties, normalization and correction for chance. *Journal of Machine Learning Research*, 11(Oct), 2837-2854.

# 4   Supplemental D

**Lemma 1** (Removable-Noise Monotonicity; proof in *supplemental*)**.** *Let $Z \subseteq X$ be labeled data and $z = (x_z, y_z) \in Z$. If substituting a fresh dummy label $\perp$ for $y_z$ increases* AMI *by at least $\sigma > 0$, then*

$$\text{LHI}(Z \setminus \{z\}) \;\leq\; \text{LHI}(Z) - \sigma. \tag{4}$$

*Proof.* Recall that for any labelled sample $S$

$$\text{LHI}(S) \;=\; 1 - \text{AMI}(S) \;=\; 1 - \frac{I(Y;C) - \mathbb{E}[I(Y;C)]}{\frac{1}{2}\big(H(Y) + H(C)\big) - \mathbb{E}[I(Y;C)]},$$

where $Y$ is the ground-truth label partition, $C$ is the cluster/feature partition produced by the locality-preserving embedding, $I(\cdot;\cdot)$ is mutual information and $H(\cdot)$ Shannon entropy.

**Step1 – AMI increases by $\sigma$.** By assumption, substituting the dummy label $\perp$ for the single point $z$ leaves the feature partition unchanged but converts its label to a singleton, so that

$$\text{AMI}(Z \setminus \{z\}) \;\geq\; \text{AMI}(Z) + \sigma.$$

**Step2 – Normalisation denominator is invariant.** Both the entropy term $H(Y)$ and its expectation in the AMI denominator depend only on the *multiset of label counts*. Relabelling $z$ from $y_z$ to a fresh symbol $\perp$ removes one count from the $y_z$ bucket and puts it into a new singleton bucket; the total number of labels and hence the multiset of counts is unchanged. Consequently the denominator $D := \frac{1}{2}\big(H(Y) + H(C)\big) - \mathbb{E}[I(Y;C)]$ is identical for $Z$ and $Z \setminus \{z\}$.

**Step3 – Translating the AMI gap into an LHI gap.** With the same denominator $D$ we have

$$\text{LHI}(Z \setminus \{z\}) = 1 - \text{AMI}(Z) - \frac{\sigma}{D} \;=\; \text{LHI}(Z) - \frac{\sigma}{D}.$$

Since $0 < D \leq 1$, the factor $1/D \geq 1$ and therefore $\text{LHI}(Z \setminus \{z\}) \leq \text{LHI}(Z) - \sigma$.

**Step4 – Tightness.** The bound is tight: equality is achieved when (i) $z$ forms its own (mis-)cluster in $C$ and (ii) all other label/cluster intersections remain pure, so that the only change in mutual information stems from isolating $z$.

Hence removing a $\sigma$-noisy sample can only *decrease* the Learning-Hard Index, and by at least $\sigma$. $\qquad\square$

# 5  Supplemental E

**Theorem 2** (Existence of Low-Complexity Traininglets). *Let $X$ be the training set of any learning-hard problem (LH-P) and let $x'_1, \ldots, x'_s$ be an arbitrary query batch. Then there exist traininglets $\mathcal{T}_{x'_1}, \ldots, \mathcal{T}_{x'_s} \subseteq X$ with*

$$\min_j \mathrm{LHI}\big(\mathcal{T}_{x'_j}\big) \; < \; \mathrm{LHI}(X). \tag{5}$$

*Moreover, if $X$ contains a $\sigma$-noisy point in the sense of Lemma 1 (so $\sigma > 0$), the inequality sharpens to $\min_j \mathrm{LHI}\big(\mathcal{T}_{x'_j}\big) \; \leq \; \mathrm{LHI}(X) - \sigma$.*

*Proof of Theorem 2.* We establish two statements.

**A. Strict LHI reduction (always).** Let $Y = \{y_i : (x_i, y_i) \in X\}$ and $k := |Y|$. Start with $S \leftarrow X$ and scan its points once:

1. For $(x, y) \in S$ delete it *iff* (i) $S \setminus \{(x, y)\}$ still covers all labels and (ii) $\mathrm{LHI}(S \setminus \{(x, y)\}) < \mathrm{LHI}(S)$.

At least one deletion succeeds (otherwise $X$ is already minimal). The final set $S^\star \subsetneq X$ keeps full coverage yet satisfies $\mathrm{LHI}(S^\star) < \mathrm{LHI}(X)$. Choose the *same* $S^\star$ for every query:

$$\min_j \mathrm{LHI}\big(\mathcal{T}_{x'_j}\big) = \mathrm{LHI}(S^\star) < \mathrm{LHI}(X).$$

**B. $\sigma$–gap when a $\sigma$-noisy point exists.** Assume $X$ contains a $\sigma$-noisy sample (Lemma 1); pick one and denote it $z^{(1)}$. Iterate

$$S^{(t+1)} \; \leftarrow \; S^{(t)} \setminus \{z^{(t+1)}\}, \qquad t = 0, 1, \ldots$$

while $S^{(t+1)}$ still covers all $k$ labels. Coverage can fail after at most $k-1$ removals, so $t_{\max} \leq k-1$. Lemma 1 guarantees at each step $\mathrm{LHI}(S^{(t+1)}) \leq \mathrm{LHI}(S^{(t)}) - \sigma$, hence

$$\mathrm{LHI}\big(S^{(t_{\max})}\big) \leq \mathrm{LHI}(X) - \sigma.$$

Assign $\mathcal{T}_{x'_j} := S^{(t_{\max})}$ for every query $j$ to obtain $\min_j \mathrm{LHI}(\mathcal{T}_{x'_j}) \leq \mathrm{LHI}(X) - \sigma$.

**Complexity.** Both procedures perform $O(n)$ deletions and at most $n$ LHI evaluations, matching the polynomial-time assumption for $\mathrm{LHI}(\cdot)$.

$\square$

# 6   Supplemental F

## Proposition 3

**Proposition3 (PTC contracts the training–test gap).** Let $P_{\mathrm{tr}}$ and $P_{\mathrm{te}}$ denote the training and test distributions of a learning-hard problem. After applying Precision Traininglet Construction (PTC) within the Overfitting-Free MiL pipeline, the resulting distribution $P_{\mathrm{PTC}}$ satisfies the strict total-variation contraction

$$\left\| P_{\mathrm{PTC}} - P_{\mathrm{te}} \right\|_{\mathrm{TV}} \; < \; \left\| P_{\mathrm{tr}} - P_{\mathrm{te}} \right\|_{\mathrm{TV}}.$$

(In other words, $P_{\mathrm{tr}} \xrightarrow{\mathrm{PTC}} P_{\mathrm{te}}$.)

### *Proof:*

Let $P_{\mathrm{tr}}$ and $P_{\mathrm{te}}$ denote the training and test distributions of a learning-hard problem. After applying (PTC) in the Overfitting-Free MiL pipeline, the distribution of the remaining data $P_{\mathrm{PTC}}$ satisfies

$$\left\| P_{\mathrm{PTC}} - P_{\mathrm{te}} \right\|_{\mathrm{TV}} \; < \; \left\| P_{\mathrm{tr}} - P_{\mathrm{te}} \right\|_{\mathrm{TV}}.$$

### Assumptions

**A1** *Cluster assumption.* Each class supports a disjoint, high-density cluster in $P_{\mathrm{te}}$; the space between clusters has density 0.

**A2** *Consistency.* Every point correctly classified by the Bayes rule lies in the support of $P_{\mathrm{te}}$.

**A3** *Misclassification mass.* The set $\mathcal{B}$ of mis-classified training points has strictly positive probability under $P_{\mathrm{tr}}$.

|  |  |  |
|---|---|---|
| **Notation** | $\mathcal{B}$ | Bad (mis-classified) points, §3.2. |
|  | $\mathcal{N}_\epsilon(x)$ | Closed $\epsilon$-ball around $x$. |
|  | $\|\cdot\|_{\mathrm{TV}}$ | Total-variation distance: $\|P-Q\|_{\mathrm{TV}} = \sup_A |P(A)-Q(A)|$. |

**Key lemma (Devroye etal. 2018, Lemma2.1)**   For any distributions $P, Q$ and measurable set $A$, $\|P - Q\|_{\mathrm{TV}} - \|P^{\backslash A} - Q\|_{\mathrm{TV}} = P(A) - Q(A)$. Hence if $Q(A) = 0$ and $P(A) > 0$, removing $A$ strictly decreases TV distance.

**Proof   Step 1 — Training clean contracts TV.** Remove $\mathcal{B}$ and their $\epsilon$-balls to obtain $X^{\mathrm{clean}}$ (distribution $P_{\mathrm{clean}}$). AssumptionsA1–A2 imply $P_{\mathrm{te}}(\mathcal{B}) = 0$. By the key lemma,

$$\|P_{\mathrm{clean}} - P_{\mathrm{te}}\|_{\mathrm{TV}} = \|P_{\mathrm{tr}} - P_{\mathrm{te}}\|_{\mathrm{TV}} - P_{\mathrm{tr}}(\mathcal{B}) < \|P_{\mathrm{tr}} - P_{\mathrm{te}}\|_{\mathrm{TV}}. \tag{1}$$

**Step 2 — Precision pruning contracts TV again.** For each query $x_i'$ let $\mathcal{T}_{x_i'}^{\mathrm{PTC}} \subset X^{\mathrm{clean}}$ be the final traininglet and define $P_{\mathrm{PTC}} = \frac{1}{n} \sum_i P_{\mathcal{T}_{x_i'}^{\mathrm{PTC}}}$. Precision pruning removes only points whose $\epsilon$-balls intersect $\mathcal{B}$; call the removed set $\mathcal{C} \subset \bigcup_{b \in \mathcal{B}} \mathcal{N}_\epsilon(b)$. Again $P_{\mathrm{te}}(\mathcal{C}) = 0$ and $P_{\mathrm{clean}}(\mathcal{C}) > 0$, so

$$\|P_{\mathrm{PTC}} - P_{\mathrm{te}}\|_{\mathrm{TV}} < \|P_{\mathrm{clean}} - P_{\mathrm{te}}\|_{\mathrm{TV}}. \tag{2}$$

**Step 3 — Transitive contraction.** Combining (1) and (2):

$$\|P_{\mathrm{PTC}} - P_{\mathrm{te}}\|_{\mathrm{TV}} < \|P_{\mathrm{clean}} - P_{\mathrm{te}}\|_{\mathrm{TV}} < \|P_{\mathrm{tr}} - P_{\mathrm{te}}\|_{\mathrm{TV}},$$

which proves the proposition. $\square$

# Supplemental G+: Precision Traininglet Construction (PTC) in MiL

**Knowledge Fusion in PTC: plain english.** PTC proceeds in four stages. *Probing learning (data-driven radii).* Over repeated 80/20 splits of the training set, Naïve-MiL is run on a grid of neighborhood sizes and joint batch sizes. We retain non-dominated $(k, z)$ pairs that maximize the class-discriminative D-index, replacing ad-hoc radius choices with a label-aware, data-driven calibration.

*Training sanitization (noise removal).* Using the selected $(k^\star, z^\star)$, every training sample receives a deterministic prediction. Incorrectly predicted "error points" and their $\epsilon$-ball neighbors are pruned, yielding a cleaned pool $X_{\text{clean}}$; a minority-class safeguard re-adds the nearest "good" instance when needed, preserving label coverage while reducing LHI.

*Meta-traininglet fusion (coverage + structure).* For a query $x'$, we build four complementary meta-traininglets and take their union: (i) a local ball around $x'$ in $X_{\text{clean}}$; (ii) 1-hop transfer from the closest correctly predicted sample; (iii) 2-hop transfer for broader manifold context; and (iv) a random anchor to plug residual topology gaps. The union is small, label-complete, and aligned with the query's local geometry.

*Precision pruning (final cut).* Elements lying in neighborhoods of residual error anchors are removed, producing the precision traininglet $\mathcal{T}_{x'}^{\text{PTC}}$ with reduced LHI and improved distribution match. An RKHS learner (e.g., SVM or SVM–micro–CNN–let) is then fit on $\mathcal{T}_{x'}^{\text{PTC}}$ to yield a deterministic learning-let $f_{x'}$; the collection $\{f_{x'}\}$ across queries constitutes the global MiL predictor—overfitting-resistant, reproducible, and interpretable by design.

---

**Algorithm 1** Precision Traininglet Construction (PTC)

---

**Require:** Training set $X = \{(x_i, y_i)\}_{i=1}^n$; query batch $Q$; grids $\mathcal{K}, \mathcal{Z}$; Monte-Carlo budget $M$; pruning–noise radius $r$

**Ensure:** Precision traininglets $\{\mathcal{T}_{x'}^{\text{PTC}}\}_{x' \in Q}$

0: initialize $S[k, z] \leftarrow 0 \quad \forall (k, z) \in \mathcal{K} \times \mathcal{Z}$

0: **for** $m = 1$ **to** $M$ **do**

0:     $(X_{\text{tr}}, X_{\text{te}}) \leftarrow \text{SPLIT}(X, 0.8)$

0:     **for all** $k \in \mathcal{K}$, $z \in \mathcal{Z}$ **do**

0:         $S[k, z] \mathrel{+}= D_{\text{Naive-MiL}}(X_{\text{tr}}, X_{\text{te}}, k, z)$

0:     **end for**

0: **end for**

0: $(k^\star, z^\star) \leftarrow \arg\max_{k,z}(S[k, z]/M)$

0: $\widehat{y} \leftarrow \text{NAIVE\_MIL\_PREDICT}(X, k^\star, z^\star)$

0: $G \leftarrow \{x_i : \widehat{y}_i = y_i\}$, $B \leftarrow \{x_i : \widehat{y}_i \neq y_i\}$

0: $X_{\text{clean}} \leftarrow X \setminus \left(B \cup \bigcup_{b \in B} \mathcal{N}_r(b)\right)$

0: **for all** $x' \in Q$ **do**

0:     $\mathcal{T}^{(1)} \leftarrow \text{NTC}(x', k^\star, X_{\text{clean}})$

0:     $\mathcal{T}^{(2)} \leftarrow \bigcup_{g \in \text{NN}_1(x', G)} \text{NTC}(g, k^\star, X_{\text{clean}})$

0:     $\mathcal{T}^{(3)} \leftarrow \bigcup_{g \in \text{NN}_2(x', G)} \text{NTC}(g, k^\star, X_{\text{clean}})$

0:     $\mathcal{T}^{(4)} \leftarrow \text{NTC}(\text{Random}(G), k^\star, X_{\text{clean}})$

0:     $\mathcal{U} \leftarrow \bigcup_{j=1}^4 \mathcal{T}^{(j)}$

0:     $\mathcal{T}_{x'}^{\text{PTC}} \leftarrow \mathcal{U} \setminus \bigcup_{b \in B} \mathcal{N}_r(b)$

0: **end for**

0: **return** $\{\mathcal{T}_{x'}^{\text{PTC}}\}_{x' \in Q}$ $=0$

---

# Supplemental G: Benchmark LH-P Datasets

This section provides detailed descriptions of the five benchmark datasets used in our study to evaluate the performance of Micro-Learning on LH-Ps.

**IRMAS Dataset**

The *IRMAS (Instrument Recognition in Musical Audio Signals)* dataset was created for studying musical instrument recognition. It contains 6,705 audio files, each 3 seconds long, with a sampling rate of 44.1 kHz, 16-bit, stereo wav audio format. The dataset covers 11 instruments. The dataset is relatively balanced. Table S1 provides a summary, showing that the voice class has the largest number of samples (778), while the cello class has the smallest (388).

Table S1: IRMAS dataset summary.

| Instrument | Abbreviation | Sample Size |
|------------|--------------|-------------|
| Cello | cel | 388 |
| Clarinet | cla | 505 |
| Flute | flu | 451 |
| Acoustic Guitar | acg | 647 |
| Electric Guitar | elg | 760 |
| Organ | org | 682 |
| Piano | pia | 721 |
| Saxophone | sax | 626 |
| Trumpet | tru | 577 |
| Violin | vio | 580 |
| Voice | voi | 778 |

**CASIA Dataset**

For our speech emotion recognition task, we used the *CASIA (Chinese Emotional Speech Corpus)* dataset. This dataset was specifically designed for this task and contains emotional speech samples in Mandarin Chinese. The dataset consists of recordings from four professional speakers, with 200 samples in each of the six emotion categories: neutral, angry, fear, happy, sad, and surprise. Overall, the corpus includes 1200 emotional speech sentences and has a sampling rate of 48-kHz with 16-bit quantification.

Table S2: CASIA dataset summary.

| Emotion | Sample Size |
|---------|-------------|
| Neutral | 200 |
| Angry | 200 |
| Fear | 200 |
| Happy | 200 |
| Sad | 200 |
| Surprise | 200 |

**SAVEE Dataset**

*SAVEE* is an English speech dataset that includes recordings from four speakers, with seven different emotions represented: anger (A), disgust (D), fear (F), happiness (H), sadness (Sa), surprise (Su), and neutral (N). The neutral class has 120 samples, while the other emotion classes have 60 samples each, resulting in a total of 480 samples in the dataset.

Table S3: SAVEE dataset summary.

| Emotion | Abbreviation | Sample Size |
|---|---|---|
| Anger | A | 60 |
| Disgust | D | 60 |
| Fear | F | 60 |
| Happiness | H | 60 |
| Sadness | Sa | 60 |
| Surprise | Su | 60 |
| Neutral | N | 120 |

**COVID-19 Cough Dataset**

We curated the *COVID-19* dataset for developing an AI system for cough recording diagnosis. The dataset consists of 128 audio files categorized into three groups: COVID-19 positive, COVID-19 negative, and other tracheal or pulmonary diseases. Each observation is a one-second-long stereo wav audio file with a sampling rate of 48 kHz and bitrate of 32 bits. Table S4 provides a summary of the dataset.

Table S4: COVID-19 cough dataset summary.

| Recording Category | Sample Size |
|---|---|
| COVID-19 Positive | 48 |
| COVID-19 Negative | 73 |
| Other Tracheal or Pulmonary Diseases | 7 |

**Ovarian Tumor RNA-seq Dataset**

The *Ovarian* dataset is specifically designed for ovarian tumor diagnosis and consists of RNA-seq data from TCGA, including 262 recurrent ovarian tumors and 4 solid ovarian tumors across 20,531 genes. Unlike the other datasets, *Ovarian* does not require preprocessing before training. However, it presents a unique challenge as an extreme learning-hard problem, despite having only two classes. This is due to its small size (266 entries), high dimensionality (20,531 gene features), and severe imbalance (the majority class represents 98.5% of the dataset).

Table S5: Ovarian tumor RNA-seq dataset summary.

| Ovarian Tumor Category | Sample Size |
|---|---|
| Recurrent | 262 |
| Solid | 4 |

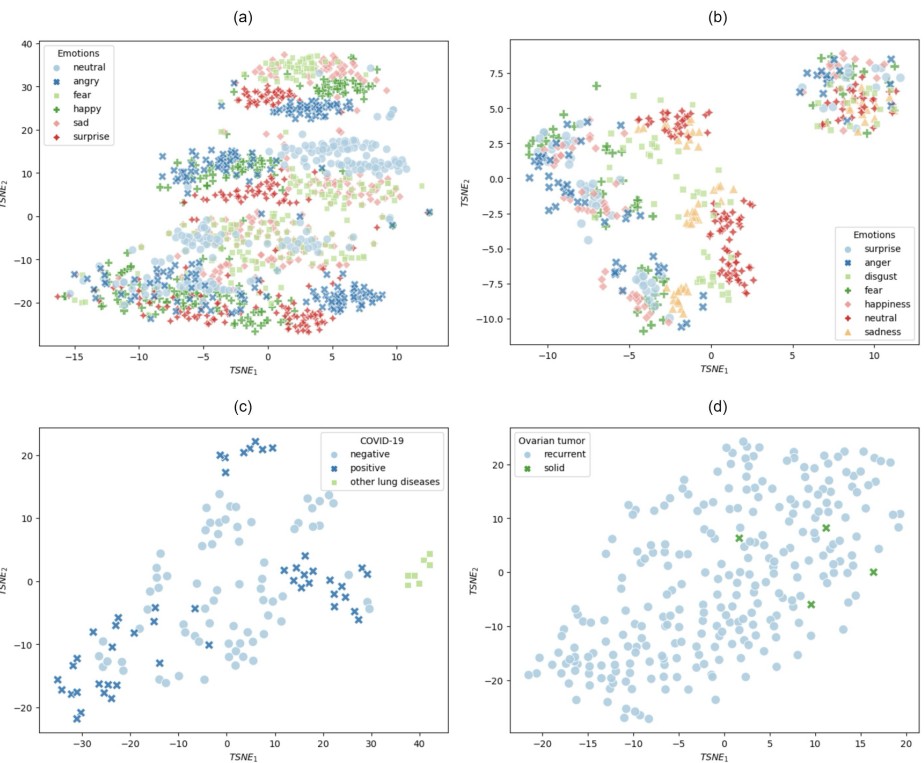

Figure 2: t-SNE visualizations of small/nonlinear datasets: CASIA, SAVEE, COVID-19, and Ovarian. The plots reveal significant separation difficulty, with mingled classes and few detectable boundaries between clusters. COVID-19 has a distinguishable third class, while Ovarian has extremely high separation difficulty, with randomly distributed observations.

# 7 Supplemental H

## Comparisons of MiL with other 11 deep learning baselines

Figure 3 offers a side-by-side evaluation of MiL against eleven state-of-the-art deep-learning (DL) architectures over five heterogeneous benchmarks: COVID-19 cough classification, IRMAS instrument recognition, CASIA Chinese speech emotion, SAVEE English speech emotion, and Ovarian cancer mass-spectra. Panels (a)–(e) report seven standard metrics—Accuracy, Sensitivity, Specificity, Precision, Negative-Predictive Rate, F-Micro, and F-Macro. In every dataset MiL's blue bars saturate the left axis, whereas the DL models (clustered bars on the right) exhibit wider variance and frequent metric shortfalls, especially on small or highly imbalanced data (COVID-19, Ovarian). Panel (f) condenses performance into a radar plot: MiL encloses the largest area, signalling superior and more stable generalisation across domains, while individual DL models form tighter, dataset-specific contours that betray overfitting and metric inconsistency

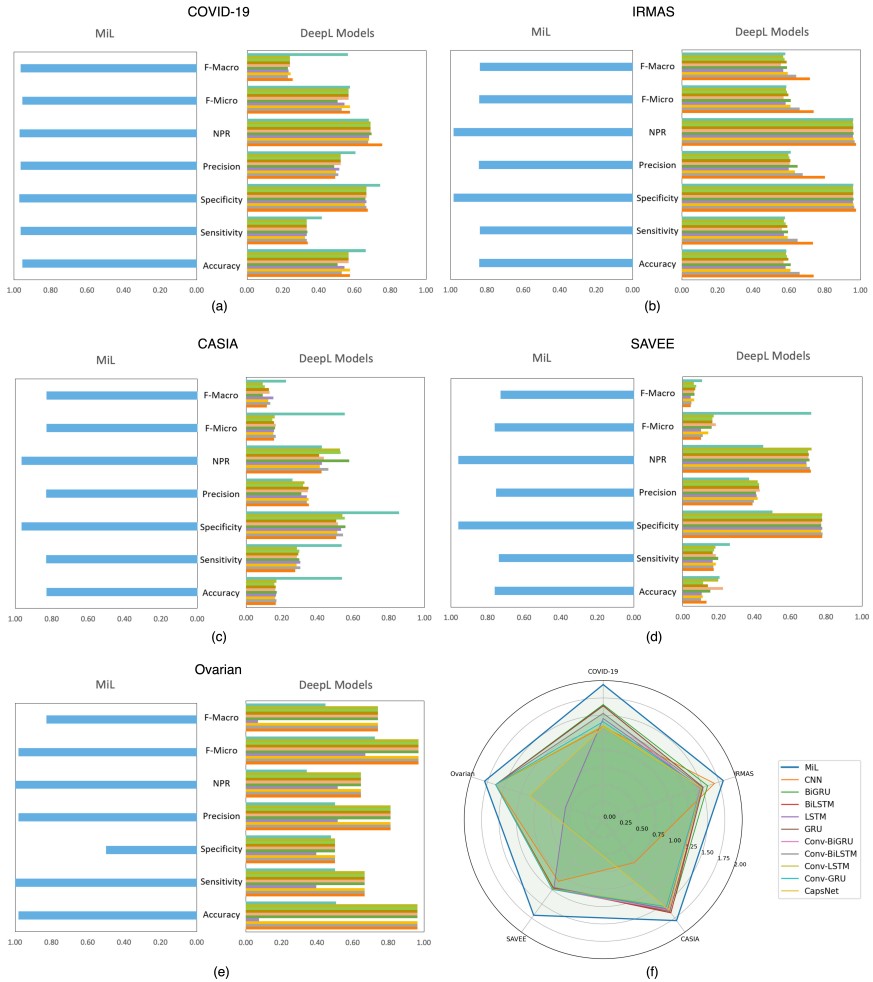

Figure 3: MiL vs. eleven deep-learning baselines. (a–e) Metric-wise comparison on five datasets; blue bars are MiL, grouped coloured bars are CNN, LSTM, GRU, BiLSTM, BiGRU, Conv-LSTM, Conv-GRU, Conv-BiLSTM, Conv-BiGRU, CapsNet, and OFL. (f) Radar chart of mean normalised performance across datasets; MiL (blue) spans the largest region, confirming its cross-domain robustness and consistent outperformance of DL peers

# 8    Supplemental I

**MiL Ablation study on COVID-19 data**

Table S6: Ablation study on MiL components (**higher is better**).

| Variant | D-Index | Acc. | TPR | TNR | PPV | NPV | $F_{\mathbf{micro}}$ | $F_{\mathbf{macro}}$ |
|---|---|---|---|---|---|---|---|---|
| A1: No Probing | 1.8472 | 0.8837 | 0.9054 | 0.9203 | 0.9180 | 0.9294 | 0.8837 | 0.9059 |
| A2: No Sanitisation | 1.8720 | 0.8991 | 0.9243 | 0.9346 | 0.9265 | 0.9357 | 0.8991 | 0.9219 |
| A3: No Fusion | 1.8784 | 0.9068 | 0.9232 | 0.9365 | 0.9345 | 0.9444 | 0.9068 | 0.9240 |
| A4: No Pruning | 1.8796 | 0.9068 | 0.9258 | 0.9382 | 0.9333 | 0.9431 | 0.9068 | 0.9248 |
| Baseline (keep all components) | **1.9424** | **0.9544** | **0.9644** | **0.9709** | **0.9632** | **0.9697** | **0.9544** | **0.9633** |

# 9. Supplemental J

Comparative Analysis of MIL and traditional ML Models (with visualizations of learning performance, traininglets, and confusion matrices)

### 8.1 COVID-19 diagnosis (COVID-19)

**Table S8. Performance summary of MIL and its peers using 5-fold cross-validation (COVID-19 data)**

| Measures\Models | MIL | SVM | RF | ET | NB | DNN |
|---|---|---|---|---|---|---|
| D-index | 1.9424 | 1.7118 | 1.8029 | 1.8234 | 1.7156 | 1.8350 |
| Accuracy | 0.9544 | 0.8230 | 0.8458 | 0.8617 | 0.7770 | 0.8771 |
| Sensitivity | 0.9644 | 0.7136 | 0.8847 | 0.8958 | 0.8435 | 0.8806 |
| Specificity | 0.9709 | 0.8906 | 0.9027 | 0.9112 | 0.8643 | 0.9240 |
| Precision | 0.9632 | 0.8737 | 0.8894 | 0.9040 | 0.8446 | 0.9085 |
| NPR | 0.9697 | 0.8995 | 0.9059 | 0.9168 | 0.8654 | 0.9259 |
| F-Micro | 0.9544 | 0.8230 | 0.8458 | 0.8617 | 0.7770 | 0.8771 |
| F-Macro | 0.9633 | 0.7311 | 0.8801 | 0.8936 | 0.8329 | 0.8843 |

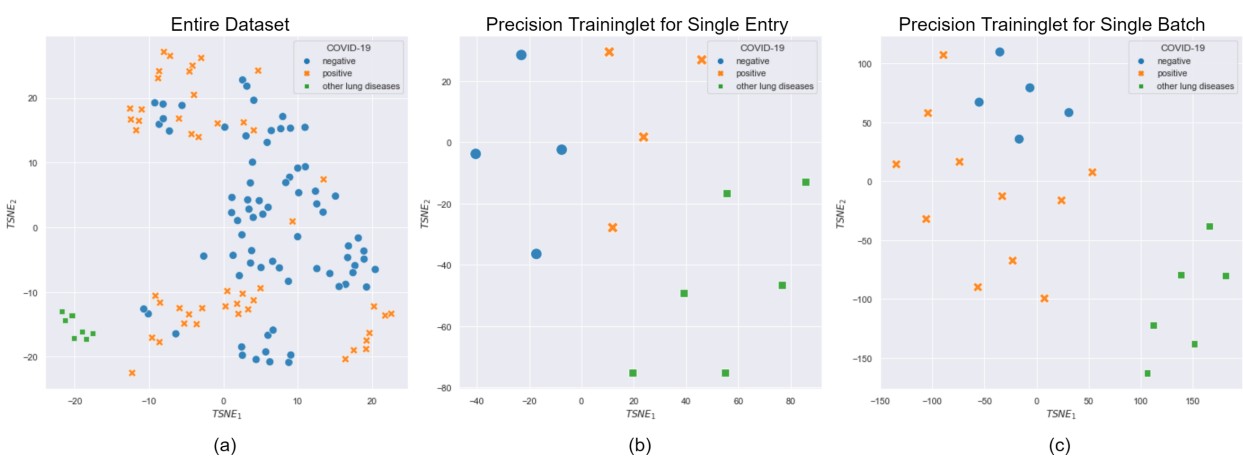

**Figure S10** Performance comparison of MIL with five peers on the COVID-19 dataset. Figures (c) and (d) present MIL's normalized and classic confusion matrices respectively.

**Figure S11.** Comparative Traininglet Analysis of MIL on the COVID-19 Dataset. In the t-SNE visualization (Figure (a)), the complete dataset displays significant nonlinearity and vague demarcation between 'negative' and 'positive' classes. Notably, the precision *traininglet* of a single entry (Figure (b)) and a test batch (Figure (c)) exhibit a clear distinction among the three classes, with increasingly pronounced boundaries.

## Predominant Musical Instrument Recognition in Polyphonic Music (IRMAS data)

**Table S9. IRMAS 5-folds Performance Summary**

| Measures\Models | MIL | SVM | RF | ET | NB | DNN |
|---|---|---|---|---|---|---|
| D-Index | 1.8162 | 1.5310 | 1.5153 | 1.5201 | 1.1662 | 1.4926 |
| Accuracy | 0.8431 | 0.6213 | 0.6103 | 0.6143 | 0.3676 | 0.5909 |
| Sensitivity | 0.8387 | 0.6052 | 0.5926 | 0.5957 | 0.3484 | 0.5802 |
| Specificity | 0.9842 | 0.9620 | 0.9607 | 0.9610 | 0.9363 | 0.9590 |
| Precision | 0.8449 | 0.6184 | 0.6319 | 0.6488 | 0.3660 | 0.5828 |
| NPR | 0.9843 | 0.9622 | 0.9613 | 0.9618 | 0.9370 | 0.9591 |
| F-Micro | 0.8431 | 0.6213 | 0.6103 | 0.6143 | 0.3676 | 0.5909 |
| F-Macro | 0.8397 | 0.6053 | 0.5944 | 0.5971 | 0.3336 | 0.5792 |

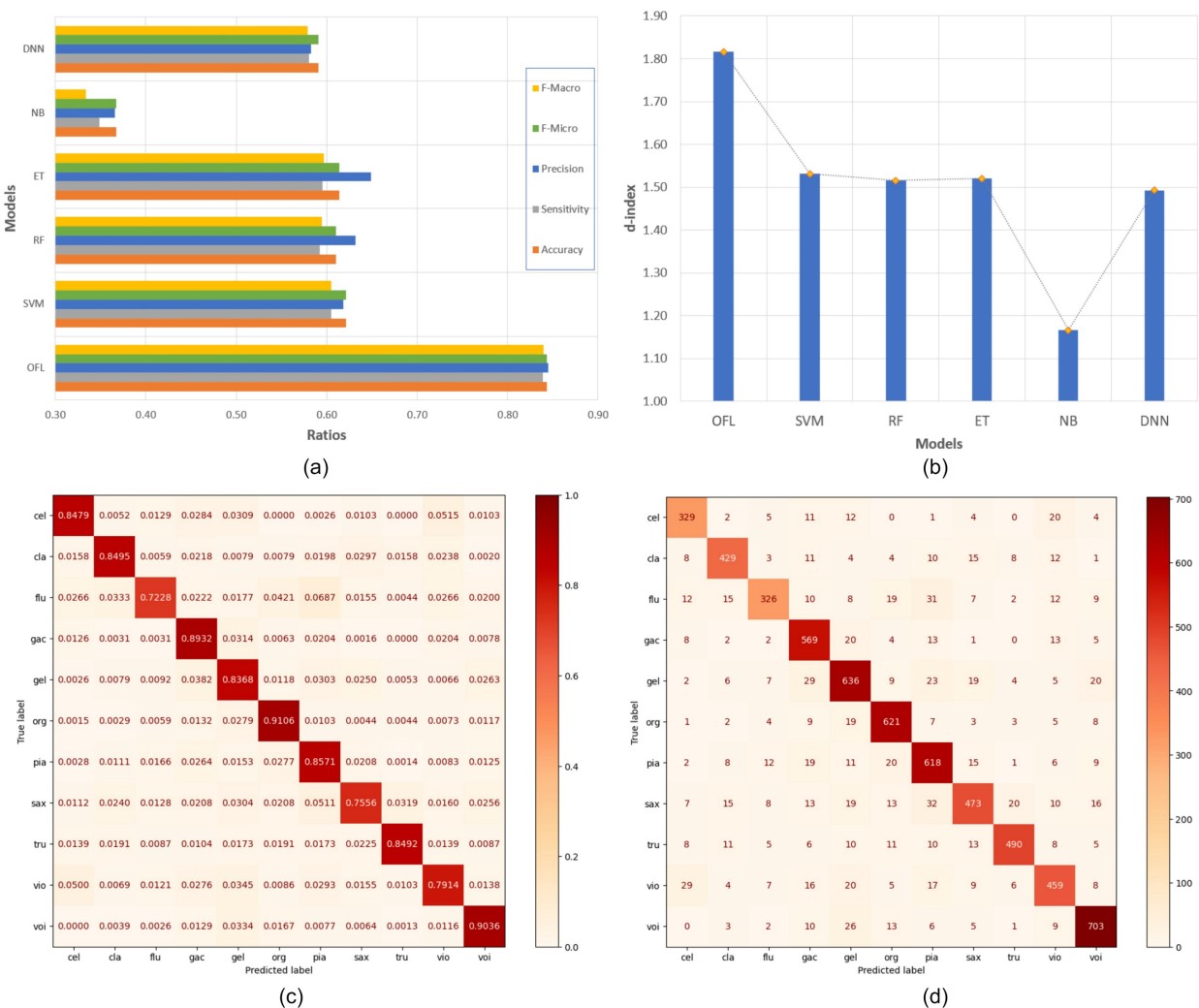

**Figure S12.** The comparisons of Naïve MIL performance with peer ML methods models on the IRMAS datasets in terms of traditional metrics and d-index values as well as the confusion matrices of Naïve MIL. Figures (c) and (d) present Naïve MIL's normalized and classic confusion matrices respectively.

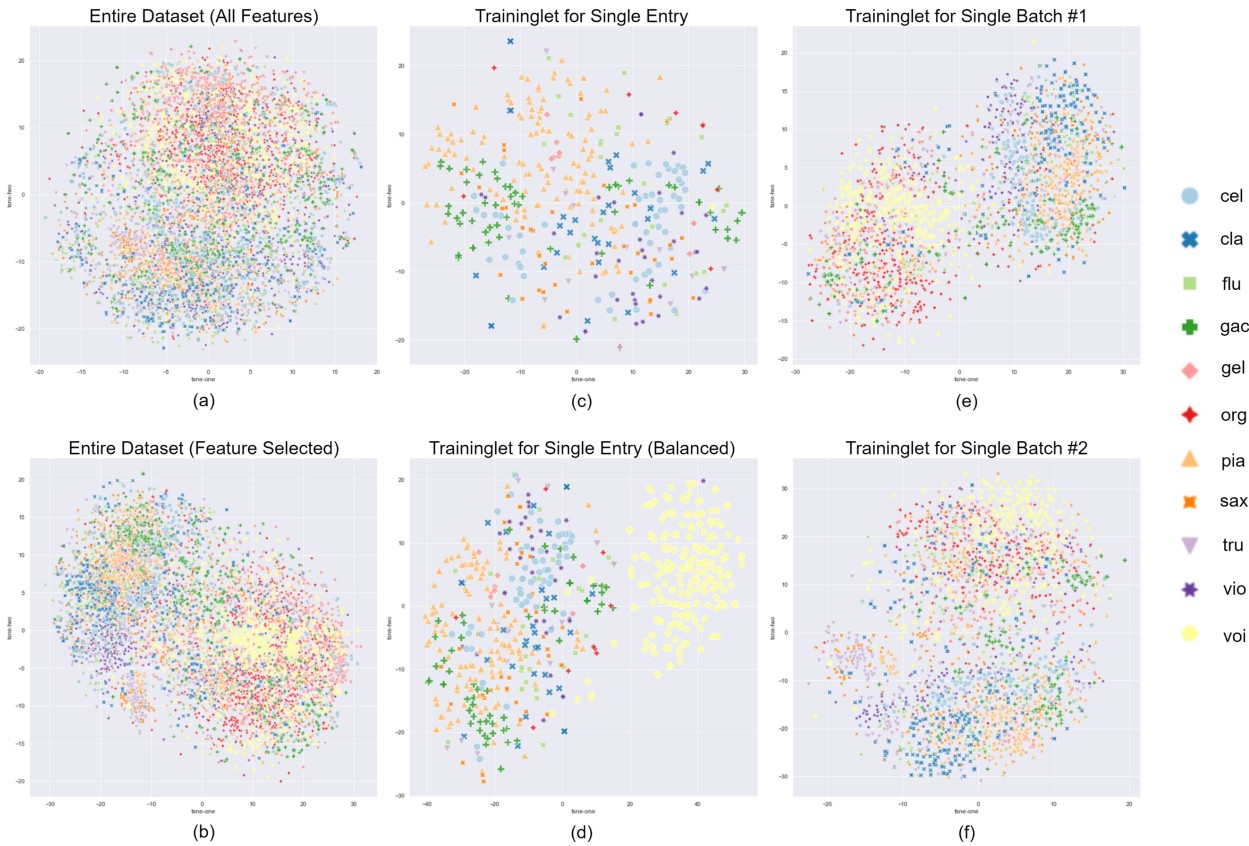

**Figure S13.** The comparisons of t-SNE visualizations of the original data before/after probing feature selection (a-b) and the t-SNE visualizations of different traininglet in the Naïve MIL (Figure c-f). Figure (a)/(b): data before/after probing feature selection. Figure (c): the T-SNE visualization of a customized training set (traininglet) for a single entry generated by the Naïve MIL. Figure (d): the T-SNE visualization of the customized training set with an adding-back neighbor set. Figure (e)(f): the T-SNE visualization the customized batch training sets in the Naïve MIL for each batch.

## Speech Emotion Recognition (CASIA and SAVEE)

### Table S10. CASIA Performance Summary

| Measures\Models | MIL | SVM | RF | ET | NB | DNN |
|---|---|---|---|---|---|---|
| D-Index | 1.7949 | 1.7703 | 1.7223 | 1.7232 | 1.4588 | 1.7736 |
| Accuracy | 0.8283 | 0.8083 | 0.7708 | 0.7717 | 0.5758 | 0.8108 |
| Sensitivity | 0.8314 | 0.8124 | 0.7747 | 0.7751 | 0.5766 | 0.8155 |
| Specificity | 0.9657 | 0.9617 | 0.9543 | 0.9545 | 0.9152 | 0.9622 |
| Precision | 0.8297 | 0.8101 | 0.7745 | 0.7756 | 0.6014 | 0.8144 |
| NPR | 0.9657 | 0.9618 | 0.9544 | 0.9546 | 0.9158 | 0.9623 |
| F-Micro | 0.8283 | 0.8083 | 0.7708 | 0.7717 | 0.5758 | 0.8108 |
| F-Macro | 0.8284 | 0.8080 | 0.7696 | 0.7705 | 0.5757 | 0.8114 |

### Table S11. SAVEE Performance Summary

| Measures\Models | MIL | SVM | RF | ET | NB | DNN |
|---|---|---|---|---|---|---|
| D-Index | 1.7015 | 0.9503 | 1.5843 | 1.5723 | 1.4612 | 1.0395 |
| Accuracy | 0.7625 | 0.2667 | 0.6771 | 0.6708 | 0.5917 | 0.3063 |
| Sensitivity | 0.7365 | 0.1899 | 0.6396 | 0.6247 | 0.5429 | 0.2724 |
| Specificity | 0.9603 | 0.8694 | 0.9446 | 0.9429 | 0.9306 | 0.8842 |
| Precision | 0.7458 | 0.0000 | 0.6841 | 0.7010 | 0.0000 | 0.0000 |
| NPR | 0.9608 | 0.8865 | 0.9480 | 0.9475 | 0.9330 | 0.8897 |
| F-Micro | 0.7625 | 0.2667 | 0.6771 | 0.6708 | 0.5917 | 0.3063 |
| F-Macro | 0.7270 | 0.0971 | 0.6298 | 0.6242 | 0.5391 | 0.2076 |

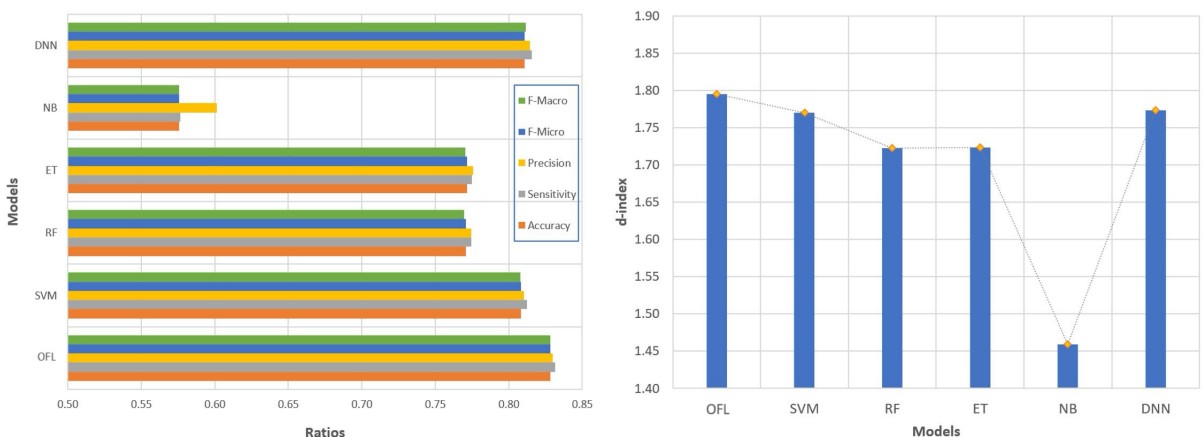

**Figure S14** Comparisons of MIL with five widely used ML models on the CASIA datasets in terms of traditional measures and d-index values. The left plot compares the performance of MIL and five ML models with 5 traditional classification measures. The right plot evaluates the performance of MIL and ML models with d-index values.

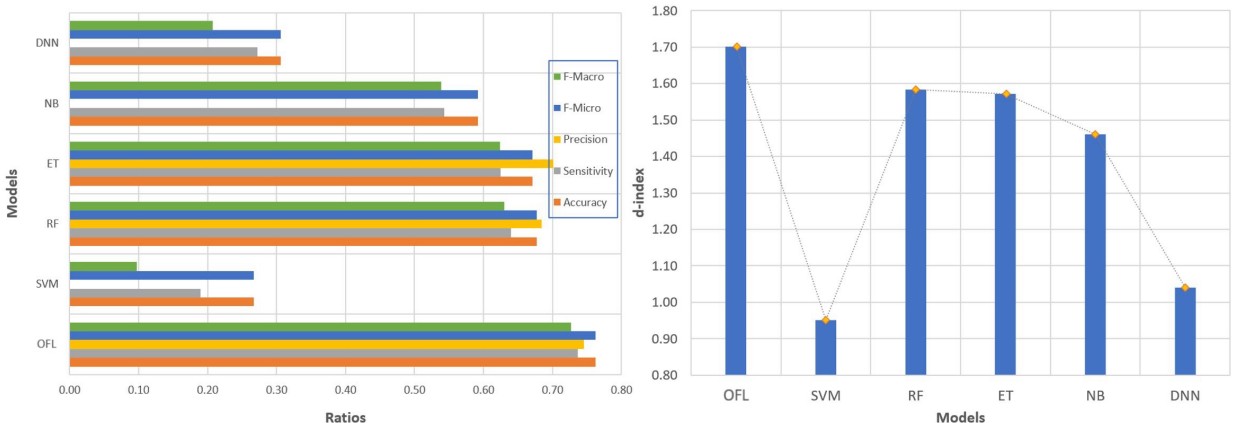

**Figure S15** Comparisons of MIL with five widely used ML models on the SAVEE datasets in terms of traditional measures and d-index values. The left plot compares the performance of MIL and five ML models with 5 traditional classification measures. The right plot evaluates the performance of MIL and ML models with d-index values.

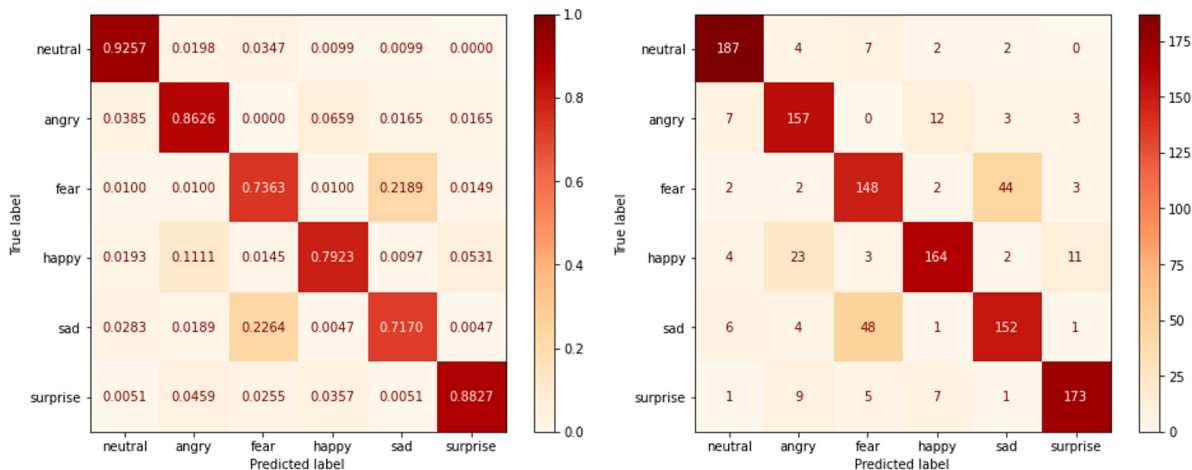

**Figure S16** The confusion matrix of MIL on the CASIA data under the five-fold cross validation. The left plot shows normalized confusion matrix over five folds, where the right plot shows non-normalized confusion matrix over five folds.

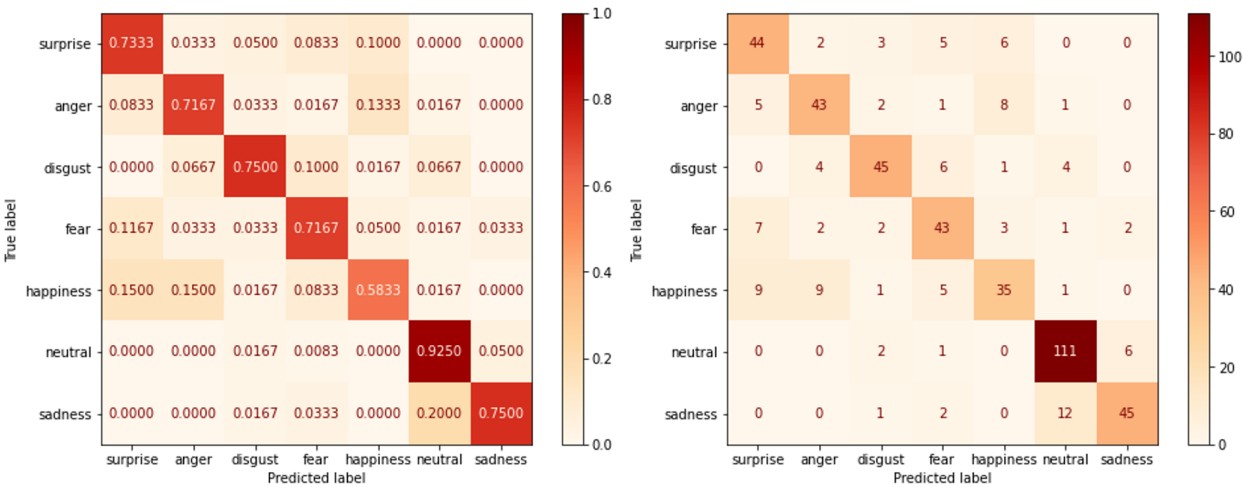

**Figure S17** The confusion matrix of MIL on the SAVEE data under the 10-fold cross validation. The left plot shows normalized confusion matrix over five folds, where the right plot shows non-normalized confusion matrix over 10 folds.

# Disease diagnosis (Ovarian)

**Table S12. Performance summary of MIL and its peers on the ovarian dataset**

| Measures\Models | MIL | SVM | RF | ET | NB | DNN |
|---|---|---|---|---|---|---|
| D-Index | 1.7939 | 1.5580 | 1.5580 | 1.5580 | 1.5580 | 1.5580 |
| Accuracy | 0.9815 | 0.9630 | 0.9630 | 0.9630 | 0.9630 | 0.9630 |
| Sensitivity | 1.0000 | 1.0000 | 1.0000 | 1.0000 | 1.0000 | 1.0000 |
| Specificity | 0.5000 | 0.0000 | 0.0000 | 0.0000 | 0.0000 | 0.0000 |
| Precision | 0.9811 | 0.9630 | 0.9630 | 0.9630 | 0.9630 | 0.9630 |
| NPR | 1.0000 | Nan | Nan | Nan | Nan | Nan |
| F-Micro | 0.9815 | 0.9630 | 0.9630 | 0.9630 | 0.9630 | 0.9630 |
| F-Macro | 0.8286 | 0.4906 | 0.4906 | 0.4906 | 0.4906 | 0.4906 |

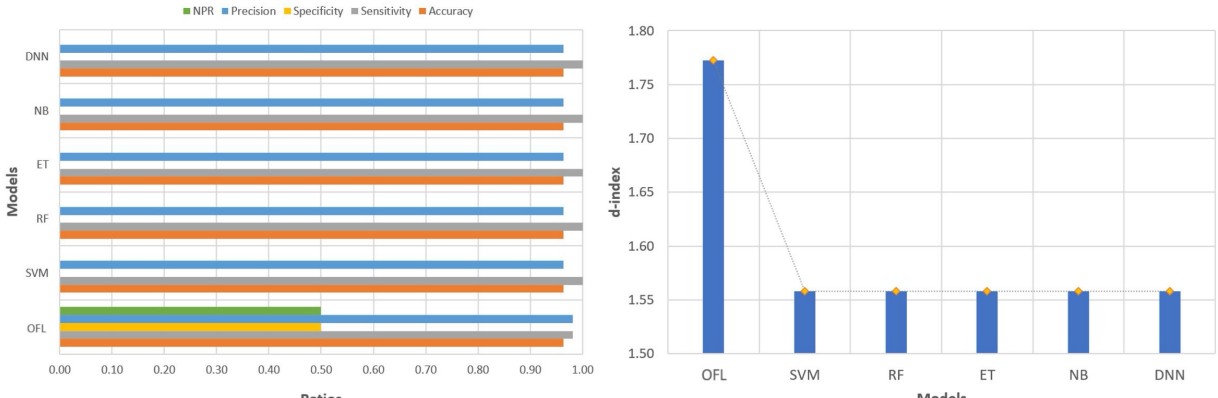

**Figure S18**. Naïve MIL and five ML models were compared on the Ovarian datasets using traditional metrics and d-index values. The left plot shows their performance using 5 classification measures, while the right plot uses d-index values.

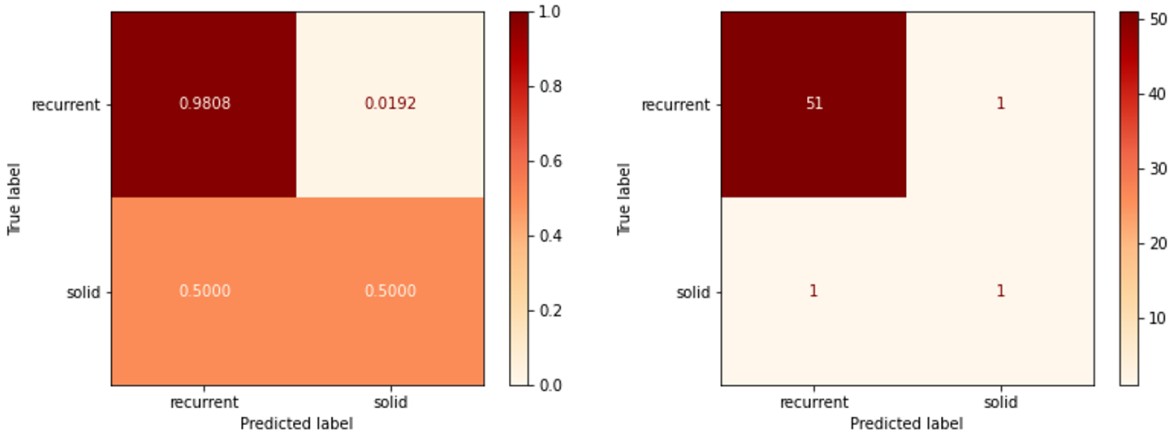

**Figure S19**. The confusion matrix of Naïve MIL on the Ovarian data under the five-fold cross validation. The left plot shows normalized confusion matrix over five folds, where the right plot shows non-normalized confusion matrix over five folds.

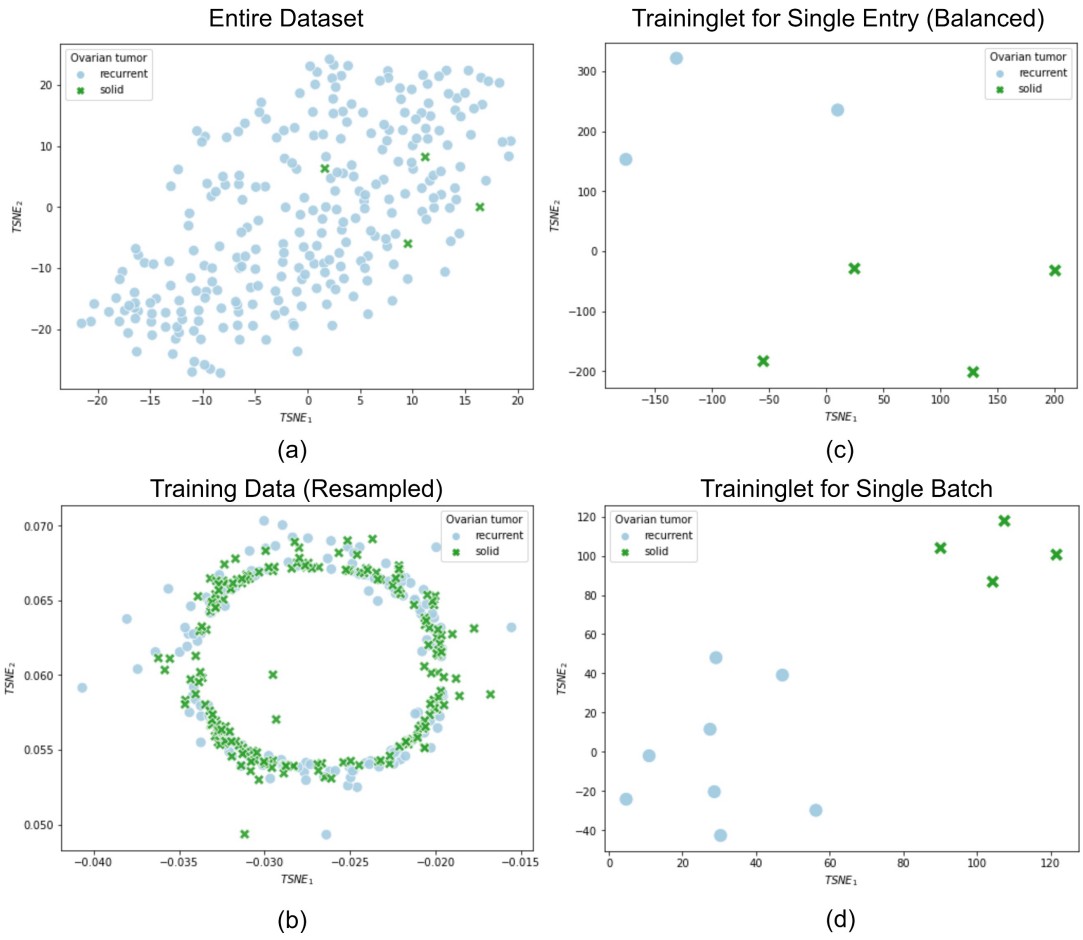

**Figure S20.** The t-SNE analysis of the imbalanced ovarian dataset (a), ROS-resampled training data (b), and traininglets identified for a single entry and batch in Naïve MIL (c-d).

# Supplemental K: High-dimensional Noisy SEC 8K vecorization data "fails" MiL

**High-dimensional noisy SEC 8K data.** The original dataset is text data: SEC 8-K filings of all companies enlisted in the S&P 500 index from 2015 to 2019 to forecast daily stock excess returns, commonly referred to as alpha values, for S&P 500 stocks using 8-K filings (Ke et al., 2019).

**Their vectorized data are high-dimensional noisy data (BERTNLP).** The corresponding BERT vectorized data consists of 768 features across 17,648 observations, and TF-IDF vectorized data contain 27,147 features across the same 17,648 observations. This representation is noisy because most features capture boilerplate or idiosyncratic wording that is only weakly and inconsistently related to the underlying return signal besides built-in noise of original text.

**The corresponding classification is an learning-hard problem (LH-P) in finance.** We further divided into three return classes: alpha $< -0.01$, alpha between $(-0.01, 0.01)$, and alpha $> 0.01$. Simplified, these are termed "buy," "hold," and "sell." With "hold" as the majority class (51.61%) and both "buy" and "sell" as minority classes (23.45% and 24.94% respectively), the dataset's inherent imbalance poses challenges, especially given its 99.7% LHI. The refined 8K data is split into 13,655 training samples and 3,993 test samples, maintaining a similar class distribution.

The LHI (learning hard index) of this Bert vectorized dataset (BERTNLP) is 99.7%, indicating its classification is a typical learning-hard problem (LH-P).

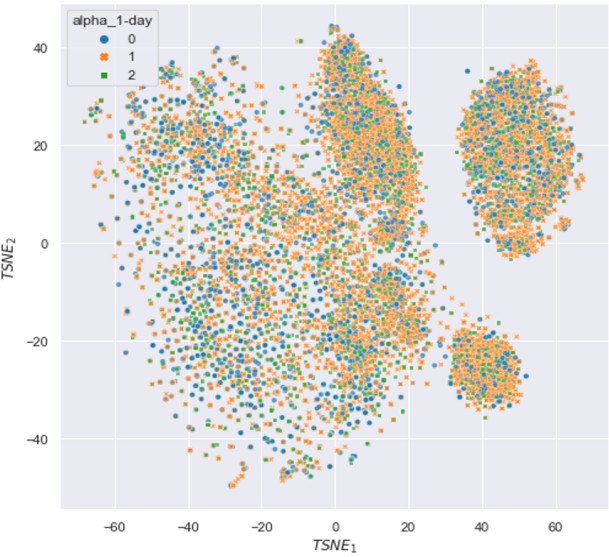

Figure 4: **t-SNE visualization of high-dimensional noisy SEC 8-K data (BERTNLP).** Two-dimensional t-SNE projection of the 768-dimensional BERT vectors for 17,648 SEC 8-K filings, coloured by 1-day ahead excess return class ($\texttt{alpha} \in \{0, 1, 2\}$). The strong overlap of the three classes in the embedding illustrates that the feature space is extremely noisy, with a LHI of 99.7%.

**MiL is only slightly better** Almost all classic ML and DL methods have failed badly on this dataset (BERT vectorized data). The whole learning procedure is almost totally hijacked by the majority class by misclassifying almost all minority samples into the majority type for all ML and ML peers.

The proposed MiL can alleviate the 'hijack' somewhat, indicating it can bring come enhancements compared to peer methods. However, it can't overcome the inferior performance caused by the high-dimesnional noisy vecctorizted data. In particular, such ligh advanatage is acheived with much high computing costs due to the high complexity of MiL.

The following Figure 5 compares the confusion matrices under classic SVM and MiL. It shows that only 77.87% of class 0 and 77.83% of class 2 samples are misclassified as the majority type under Micro-learning, but the corresponding ratios under the classic SVM model can even achieve 97.24% and 97.26%, respectively.

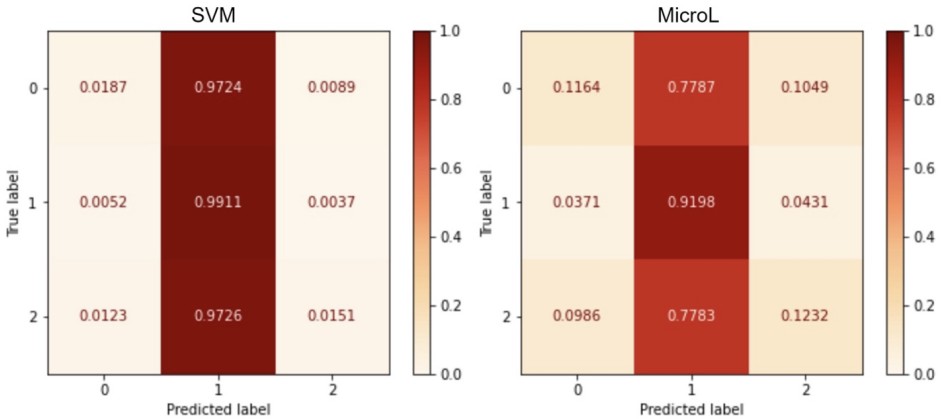

Figure 5: Normalized confusion matrices of RBF-SVM (left) and MiL (MicroL) (right) on the SEC-8K BERT vectorization dataset. Both models are dominated by the majority "hold" class, indicating that the high-dimensional noisy, imbalanced data severely hinders discrimination, although MiL partially alleviates this effect compared to RBF-SVM.

**why MiL fails on such data?** We hypothesize that high vectorization noise degrades the knowledge fusion essential for constructing effective traininglets, a problem that persists even when standard dimension-reduction–based de-noising is employed. Vectorization methods like BERT or TF-IDF can introduce significant noise from boilerplate legal text, firm-specific jargon, and other idiosyncratic wording that carries little information about the prediction task. Because standard unsupervised dimensionality reduction preserves variance rather than task-relevant structure, it cannot reliably remove these spurious components. Consequently, traininglets are built from local neighborhoods dominated by noisy similarities instead of economically meaningful ones, which prevents effective knowledge fusion.

# References

Ke, Z., Kelly, B., and Xiu, D. (2019). Predicting returns with text data. National Bureau of Economic Research Working Paper No. 26297.

# Supplemental L: Baseline Hyperparameter Details

To ensure a fair comparison between Micro-Learning (MiL) and established methods, we adhered to a rigorous hyperparameter tuning protocol for all 15 baselines. Given the small sample sizes and high dimensionality characteristic of Learning-Hard Problems (LH-Ps), all Deep Learning (DL) baselines were heavily regularized to prevent trivial overfitting.

## L.1 Experimental Setup and Budget

**Hardware:** All experiments were conducted on working stations equipped with NVIDIA A100 or RTX 4080 GPUs.

**Evaluation Strategy:** We employed the same 5-fold cross-validation (10-fold for SAVEE) used for MiL. For every fold, hyperparameters were selected via an inner validation split (nested CV).

**Computational Budget:** Each DL baseline was allowed up to 100 epochs per fold with early stopping.

## L.2 Classical Machine Learning Baselines

Classical models were implemented using package `scikit-learn`. We performed a grid search for the following hyperparameters:

- **SVM (C-SVM):** Kernel $\in$ {Linear, RBF}, $C \in \{0.1, 1, 10, 100\}$, $\gamma \in \{10^{-3}, 10^{-4}, \text{'scale'}\}$.

- **Random Forest ExtraTrees:** $n_{\text{estimators}} \in \{100, 300, 500\}$, Max Depth $\in \{10, 20, \text{None}\}$, Min Samples Split $\in \{2, 5\}$.

- **Naïve Bayes:** GaussianNB with var_smoothing $\in \{10^{-9}, 10^{-8}\}$.

- **DNN (MLP with 3 Hidden Layers):**

    - Hidden Layer Sizes: $\in \{(128, 64, 32), (256, 128, 64)\}$.
    - Activation: $\in \{\text{ReLU}, \text{Tanh}\}$.
    - L2 Penalty ($\alpha$): $\in \{10^{-4}, 10^{-3}\}$.

## L.3 Deep Learning Baselines: Architecture and Training

To ensure fairness, we did not use "vanilla" implementations; we added regularization techniques (Batch Normalization, Dropout) to every baseline to maximize their potential on small data. We note that the inclusion of Batch Normalization and Dropout in DL introduces stochasticity, which may result in some run-to-run variations compared to the fully deterministic nature of the MiL learninglet.

**Input Adaptations**

- **1D Data (Ovarian, COVID-19 tabular features):** Models used 1D Convolutional heads or Dense embeddings.

- **2D Data (IRMAS, CASIA, SAVEE spectrograms):** Models used 2D Convolutional heads.

**Common Training Parameters**

- **Optimizer:** Adam ($\beta_1 = 0.9, \beta_2 = 0.999$).

- **Loss Function:** CrossEntropyLoss (weighted for imbalanced datasets).

- **Learning Rate Schedule:**
  ReduceLROnPlateau (factor=0.5, patience=5).

- **Early Stopping:** Monitor validation loss, patience=**15** epochs.

- **Grid Search Ranges:**

  - Learning Rate: $\{10^{-3}, 5 \times 10^{-4}, 10^{-4}\}$
  - Batch Size: $\{16, 32\}$ (kept small due to data scarcity)
  - Dropout Rate: $\{0.3, 0.5\}$

**Model-Specific Architectures**

Below we detail the specific structures used. Note: $K$ denotes the number of classes.

1. **CNN (Standard):**

- Layer 1: Conv(32 filters, 3x3) $\to$ BatchNorm $\to$ ReLU $\to$ MaxPool.

- Layer 2: Conv(64 filters, 3x3) $\to$ BatchNorm $\to$ ReLU $\to$ MaxPool.

- Layer 3: Conv(128 filters, 3x3) $\to$ BatchNorm $\to$ ReLU $\to$ GlobalAveragePool.

- Head: Dense(128) $\to$ Dropout(0.5) $\to$ Dense($K$).

2. **RNN series: LSTM, GRU, Bi-LSTM, Bi-GRU:** These models utilized a feature extraction front-end (dense or shallow Conv) followed by recurrent layers.

- Input: Features extracted via a single dense layer (size 128) to reduce dimensionality.

- Recurrent Layers: 2 stacked layers. Hidden dimension $\in \{64, 128\}$.

- Bidirectional: True for Bi-LSTM/Bi-GRU (effectively doubling hidden state size).

- Head: Attention mechanism followed by Dense($K$).

3. **Hybrid model: (Conv-LSTM, Conv-GRU, Conv-BiGRU):** These models are critical for spatiotemporal data (spectrograms).

- Front-end: 2 blocks of [Conv2D $\to$ BN $\to$ ReLU $\to$ Pool].

- Reshape: Output flattened along the frequency axis to form a time-series sequence.

- Recurrent Block: 1 layer of LSTM/GRU (or Bi-directional variant). Hidden size 128.

- Head: Dense(64) $\rightarrow$ ReLU $\rightarrow$ Dense($K$).

**4. Hybrid model: Capsule Networks (CapsNet):** We adapted the standard dynamic routing architecture for inputs.

- Primary Caps: Conv2D (256 filters, 9x9, stride 1).

- Digit Caps: 8D capsules, 3 dynamic routing iterations.

- Reconstruction Loss: Added as a regularizer ($\lambda = 0.0005$) during training to stabilize feature learning on small datasets.

- *Note:* The topology was scaled down for small-sample datasets (e.g., COVID-19) to strictly control overfitting.

## L.4 Data Augmentation

We applied data augmentation (SMOTE and noise injection) specifically to the Ovarian dataset to address its extreme class imbalance (98.5% majority). No data augmentation was performed for the other datasets. For the COVID-19 dataset, despite its imbalance, the minority groups were found to be sufficiently linearly separable in the feature space, rendering synthetic augmentation unnecessary. For the balanced datasets (IRMAS, CASIA, SAVEE, CIFAR100), the raw spectral features provided a sufficient signal for training.

# Supplemental M: Performance of meta-learning on 5 benchmarks

The average performance of two mean-learning baselines: ProtoNet and MAML on the five benchmarks. Both baselines have poor performance compared to MiL, especially for the small-sample datasets such as COVID-19 and Ovarian. Beyond raw performance, ProtoNet and MAML also exhibit weaker reproducibility and explainability: their single, high-capacity global networks are sensitive to random initialization and episodic sampling, and they offer little transparent structure for tracing how individual training cases influence a given prediction, in sharp contrast to MiL's deterministic, traininglet-level decomposition.

Table S8: Average meta-learning performance (ProtoNet and MAML) across five benchmarks.

| Dataset | Model | D-index | Acc | Sen | Spe | Prec | NPV |
|---------|-------|---------|-----|-----|-----|------|-----|
| CASIA | ProtoNet | 1.7037 | **0.7583** | 0.7581 | 0.9517 | 0.7757 | 0.9526 |
| | MAML | 1.1493 | 0.3700 | 0.3693 | 0.8742 | 0.3726 | 0.8764 |
| COVID-19 | ProtoNet | 1.2347 | 0.4608 | 0.6064 | 0.6685 | 0.5383 | 0.6723 |
| | MAML | 1.4747 | **0.6172** | 0.7050 | 0.7581 | 0.6346 | 0.7564 |
| IRMAS | ProtoNet | 1.5928 | **0.6668** | 0.6575 | 0.9667 | 0.6629 | 0.9668 |
| | MAML | 1.3682 | 0.5054 | 0.4893 | 0.9504 | 0.4835 | 0.9511 |
| Ovarian | ProtoNet | 1.1152 | 0.4630 | 0.4808 | 0.4808 | 0.4972 | 0.4972 |
| | MAML | 1.5475 | **0.7778** | 0.6442 | 0.6442 | 0.5298 | 0.5298 |
| SAVEE | ProtoNet | 1.4432 | **0.5687** | 0.5538 | 0.9279 | 0.5584 | 0.9277 |
| | MAML | 0.9706 | 0.2521 | 0.2641 | 0.8753 | 0.2472 | 0.8769 |

## MiL Wins meta-learning statistically

A paired $t$-test confirms MiL significantly outperforms the best baseline in accuracy ($t(4) = 4.61$, $p \approx 0.01$) with a substantial effect size (Cohen's $d = 2.06$). Furthermore, a non-parametric sign test on the D-index demonstrates consistent improvement, with MiL surpassing the baseline across all five benchmarks (5/5, one-tailed $p = 0.032$).

## Why Meta-Learning Falters LH-Ps?

Meta-Learning methods (MAML and ProtoNet) fail to solve Learning-Hard Problems (LH-Ps) because they fundamentally lack the mechanism to reduce the intrinsic complexity of the data before training. While they adapt to new tasks via support sets, they accept the data "as is," failing to satisfy the *Latent Solvability* condition (C2) which requires active, label-aware structural knowledge fusion. By operating on the raw, high-entropy distribution (high LHI), they remain trapped in the "tangled swirl" (Fig. 1) where no global metric or initialization can generalize effectively.

Here are the three specific reasons why they fail:

- **Lack of Knowledge Fusion:** LH-Ps are defined by data that is only solvable when specific structural knowledge is fused into the training process (Def. 1). MAML and ProtoNet rely solely on "observed features" from a randomly sampled support set. Unlike MiL, they cannot perform *Traininglet Construction*—the active selection of a "knowledge-fused" subset (Theorems 1 & 2)—to mathematically lower the Learning-Hard Index (LHI) and isolate the specific manifold structure required for the query.

- **Global Priors vs. Local Sanitization:** Meta-learners attempt to learn a single global prior (an initialization for MAML or a metric space for ProtoNet) that generalizes across all tasks. However,

LH-Ps are characterized by "near-universal failure" in the global hypothesis space (Condition C1). As noted in Section 5, meta-learners rely on a "single, noise-sensitive global model," whereas MiL solves LH-Ps by determining a *local* "sweet-spot" (Prop. 1) and physically sanitizing the data to remove noise ("bad guys") specific to the query instance, which global meta-priors cannot address.

- **Vulnerability to Distribution Shift and Overfitting:** LH-Ps often involve severe small-sample or imbalanced regimes (Table 1) where deep, nested non-linearities (Eq. 1) are prone to overfitting. MAML and ProtoNet lack the theoretical guarantee of *Prop. 3*, which ensures the strict contraction of the training–test total-variation distance. While MiL minimizes the *local Rademacher complexity* (Prop. 1) by fitting simple, deterministic models (SVMs) to constrained data, meta-learners retain high-capacity deep architectures that are highly sensitive to the perturbations and noise inherent in LH-Ps.

# Supplemental N: Learning-Hard Index robustness and sensitivity analysis

## Calculate LHI w./without tSNE

To validate the robustness of the Learning-Hard Index (LHI), we compared scores derived from Direct K-Means against those using t-SNE (with 5 different perplexity parameters) projection across the five benchmarks. The following tables show there are only very tiny differences between two LHI values.

Table S9: Comparison of Learning-Hard Index (LHI) calculated via Direct K-Means versus Mean t-SNE projection. The small differences indicate the metric is robust to the projection method.

| Dataset | Direct LHI | Mean LHI (t-SNE) | LHI Difference |
| --- | --- | --- | --- |
| | | | (Mean t-SNE − Direct) |
| **CASIA** | 0.8674 | 0.8653 | **−0.0022** |
| **COVID19** | 0.7168 | 0.7776 | **+0.0608** |
| **IRMAS** | 0.9090 | 0.8926 | **−0.0164** |
| **Ovarian** | 1.0000 | 1.0000 | **0.0000** |
| **SAVEE** | 0.8013 | 0.8143 | **−0.0071** |

**No statistical difference:** A Mann-Whitney U-test confirmed that there is no statistically significant difference between the two methods ($U = 11.5, p = 0.885$). The small deviations (mean difference $\approx 0.007$) indicate that LHI captures intrinsic data difficulty regardless of the specific manifold projection used

## Calculate LHI w./without UMAP

Table S10: Sensitivity Analysis of LHI: Direct Calculation (Raw Features) vs. UMAP Projection. The small differences (non-significant via U-test) confirm LHI robustness.

| Dataset | Direct LHI | Mean UMAP LHI | Difference |
| --- | --- | --- | --- |
| | (Raw Features) | (Avg. across neighbors) | (UMAP − Direct) |
| **CASIA** | 0.8674 | 0.8338 | **−0.0336** |
| **COVID19** | 0.7168 | 0.7993 | **+0.0825** |
| **IRMAS** | 0.9090 | 0.8792 | **−0.0298** |
| **Ovarian** | 1.0000 | 1.0000 | **0.0000** |
| **SAVEE** | 0.8013 | 0.7907 | **−0.0106** |

To assess the sensitivity of the Learning-Hard Index (LHI) to dimensionality reduction, we compared LHI scores computed via Direct K-Means (raw features) versus UMAP + K-Means across the five benchmark datasets (with 5 different neighborhood parameters). A Mann-Whitney U-test revealed no statistically significant difference between the two methods ($U = 13.0, p = 1.00$). This confirms that LHI reflects the intrinsic topological difficulty of the data rather than artifacts introduced by the specific manifold learning algorithm."

**Note: LHI conservative lower bound of Ovarian** The LHI of 0.976 in Table 1 (main text) represents a conservative lower bound derived from an optimized search to detect maximal structural separation. In

contrast, the 1.00 in the robustness analysis reflects the mean behavior under standard settings, where the extreme imbalance topologically masks the minority class. The fact that the metric saturates at 1.00 typically, and only drops slightly under rigorous optimization, reinforces the dataset's intrinsic difficulty; both values far exceed the 0.80 threshold for Learning-Hard Problems."

# Supplemental O: Empirical Validation of LHI cutoff for learning-hard

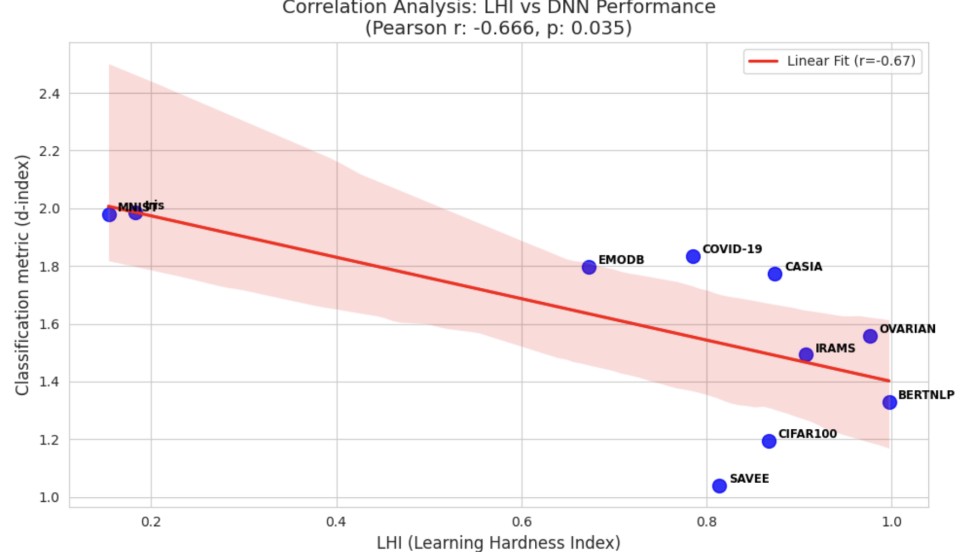

Figure 6: **Correlation Analysis: LHI vs. DNN Performance.** The scatter plot compares the Learning-Hard Index (LHI) against the classification D-index of a standard DNN with 2-hidden layers across 10 benchmark datasets. The red regression line indicates a strong inverse relationship (Pearson $r = -0.666$), and the pink shaded region denotes the 95% confidence interval. The significant $p$-value (0.035) confirms that LHI reliably predicts the degradation of standard model performance, particularly distinguishing the Learning-Hard regime (LHI $\geq 0.80$).

To empirically validate the Learning-Hard Index (LHI) and justify the selection of the $\tau = 0.80$ threshold for identifying Learning-Hard Problems (LH-Ps), we conducted a correlation study across $N = 10$ diverse datasets spanning varying modalities (image, audio, text, biological), sizes, and class imbalances.

## Experimental Setup

We evaluated the intrinsic difficulty of each dataset using the LHI formulation. As a performance benchmark, we trained a standardized Deep Neural Network (DNN) on each dataset. The benchmark model consisted of a Multi-Layer Perceptron (MLP) with 2 hidden layers (ReLU activation), trained for 100 epochs using the Adam optimizer with early stopping (patience=10).

We report the **D-index** (a composite metric of accuracy, sensitivity, and specificity, bounded $\in [0, 2]$) as the performance measure. A D-index near 2.0 indicates perfect classification, while values near 1.0 or lower indicate poor generalization. The datasets included standard "easy" baselines (Iris, MNIST), 'medium baselines (e.g., EMODB, a benchmark speech emotion recoginition dataset with 7 classes) and known complex tasks (e.g., BERTNLP (A bert vectorized dataset, see Supplemental K), Ovarian).

Table S11: LHI vs. Benchmark DNN Performance across 10 Datasets

| Dataset | Modality | LHI (Complexity) | DNN Benchmark (D-index) |
|---------|----------|------------------|-------------------------|
| Iris | Tabular | 0.1830 | 1.9860 |
| MNIST | Image | 0.1540 | 1.9800 |
| EMODB | Audio | 0.6730 | 1.7973 |
| COVID-19 | Image | 0.7850 | 1.8350 |
| SAVEE | Audio | 0.8140 | 1.0395 |
| CIFAR100 | Image | 0.8675 | 1.1954 |
| CASIA | Audio | 0.8740 | 1.7736 |
| IRAMS | Audio | 0.9070 | 1.4926 |
| OVARIAN | Bio | 0.9760 | 1.5580 |
| BERTNLP | Text | 0.9970 | 1.3309 |

## Statistical Correlation Analysis

We analyzed the relationship between the *a priori* computed LHI and the *a posteriori* DNN performance. The results demonstrate a statistically significant negative correlation, confirming that higher LHI values effectively predict lower model performance.

- **Pearson Correlation ($r$):** $-0.6663$ ($p = 0.0354$)

- **Spearman Rank Correlation ($\rho$):** $-0.6606$ ($p = 0.0376$)

Since $p < 0.05$ for both metrics, we reject the null hypothesis that LHI and model performance are unrelated. The strong negative coefficient ($r \approx -0.67$) indicates that LHI is a robust proxy for task hardness.

## 8.1   Justification of the 0.80 Threshold

The regression analysis (visualized in Figure 6) reveals two distinct regimes:

1. **The Trivial Regime ($LHI < 0.20$):** Datasets like Iris and MNIST exhibit low LHI and achieve near-perfect D-indices ($\approx 1.98$). Standard inductive biases are sufficient here.

2. **The Hard Regime ($LHI \geq 0.80$):** As LHI crosses 0.80, we observe a phase transition. Performance variance increases significantly, and the trend line drops below the robust performance zone ($D < 1.8$). While some datasets (e.g., CASIA) maintain moderate performance due to specific feature engineering, the general trend shows that standard DNNs fail to reliably solve tasks in this region without the knowledge fusion provided by MiL.

Thus, the cutoff of 0.80 is not arbitrary; it empirically demarcates the boundary where data complexity (noise, overlap, or manifold entanglements) overwhelms standard training protocols, necessitating the Micro-Learning approach.

# Supplemental P: SAM (sharpness-aware minimization) on benchmarks

SAM is applied to the 5 datasets under the same conditions. We have the following performance in the following table.

Table S2: Performance of the SAM-based classifier on LH-P datasets.

| Dataset | $d$-index | Accuracy | Sensitivity | Specificity | Precision | F1-Score |
|---------|-----------|----------|-------------|-------------|-----------|----------|
| IRAMAS | 1.5527 | 0.6324 | 0.6324 | 0.9624 | 0.6352 | 0.6309 |
| OVARIAN | 1.5580 | 0.9630 | 0.9630 | 0.0370 | 0.9273 | 0.9448 |
| SAVEE | 1.6049 | 0.6875 | 0.6595 | 0.9476 | 0.6918 | 0.6565 |
| CASIA | 1.7898 | 0.8250 | 0.8250 | 0.9650 | 0.8276 | 0.8243 |
| COVID | 1.8464 | 0.8988 | 0.8988 | 0.8897 | 0.9044 | 0.8974 |

## MiL wins SAM statistically:

MiL consistently achieves a higher $d$-index than SAM (mean 1.81 vs. 1.67). Treating the dataset-level $d$-indices as independent samples, a Mann–Whitney U test (one-sided, MiL > SAM) yields $U = 20$ (out of a maximum of 25) and a $p$-value of $p = 0.075$, indicating a clear performance trend in favour of MiL despite the small number of datasets.

MiL outperforms SAM in $d$-index on all five datasets; a non-parametric sign test on these paired scores yields a one-sided $p$-value of $p = 0.031$.

## Why SAM falters on Learning-Hard Problems (LH-Ps)?

- **Fragile reproducibility.** SAM's gains are highly sensitive to implementation choices, hyperparameters (e.g., $\rho$, learning rate, batch size), and random seeds. On small, noisy, or imbalanced data this yields large run-to-run variance, making SAM hard to reproduce as a "drop-in" improvement.

- **No label-aware structural knowledge fusion.** SAM only reshapes the loss landscape around mini-batches while still training a single global model. It does not exploit label-aware structural knowledge or construct traininglets, so it cannot systematically isolate informative local subsets or filter out conflicting samples—exactly what LH-Ps need.

- **Limited benefit in small-data / high-LHI regimes.** SAM adds computation and hyperparameters that require stable validation, which is unreliable on high-LHI datasets. In our experiments, it often saturates or even degrades performance, effectively smoothing a mis-specified global model rather than addressing the underlying data complexity.

# Supplemental Q: Pretraining for Learning-Hard Problems (LH-Ps)

## Q.1: SMOTE with noise injection with DAE pretraining + fine-tuning

We conducted augmented Unsupervised Pre-training with Supervised Fine-tuning for Ovarian data with following three steps.

1. **Manifold Augmentation:** Mitigates scarcity via adaptive oversampling (Random or $k$-adjusted SMOTE for $N \leq 5$) and Gaussian noise injection ($\mathcal{N}(0, \sigma)$), creating a robust data cloud to prevent overfitting.

2. **Representation Learning:** Pre-trains a Denoising Autoencoder (DAE) to reconstruct the underlying structure from noisy inputs, initializing encoder weights to capture robust, label-agnostic features.

3. **Discriminative Fine-tuning:** Replaces the decoder with an MLP head and fine-tunes the full network using weighted Cross-Entropy loss to optimize class separation.

**Learning Result: Complete Majority Class Collapse.** The model converged to a trivial "null" classifier, predicting the majority class for every input and failing to learn any discriminative features for the minority class.

- **The Accuracy Paradox:** While Accuracy (96.30%) appears high, it merely reflects the underlying class distribution. Crucially, the model achieved **0.00% Sensitivity** and **0.00% F1 Score**, rendering it useless for detection.

- **The D-index Illusion:** The D-index (1.558) is artificially inflated by perfect Specificity (1.0) and high Accuracy, masking the complete failure to detect minority cases.

- **Failure of Global Methods:** This result empirically validates the intractability of Learning-Hard Problems (LH-Ps) for global deep learning. Despite adaptive SMOTE and DAE pre-training, the model suffered from **manifold collapse**, treating the minority signal as noise to minimize global loss.

## More tuned pre-training methods with advanced data imbalance and high-dimensionality handling with DDPM and LASSO-feature selection

**1. DDPM-DAE pretraining.** Similarly, We implemented a pipeline integrating **Denoising Diffusion Probabilistic Models (DDPM)** for minority augmentation, followed by Denoising Autoencoder (DAE) pre-training and MLP fine-tuning. Despite this sophisticated generative approach for data imbalance handling, the model failed to capture minority patterns, yielding *0.00% Sensitivity and F1 Score*. The high Accuracy (92.59%) and D-index (1.51) are misleading artifacts of the extreme imbalance, confirming the model converged to a trivial null classifier

**2. Lasso feature selection selection + DDPM + DAE pretraining** We obtained the same results when added lasso feature selection to selected the top 300 most importrant genes before DDPM +DAE pretraining. However, the pipeline converged to a trivial "null" classifier. While Accuracy (88.89%) appears high, it merely reflects the majority class prevalence. The model achieved 0.00% Sensitivity and 0.00% F1 Score, failing to identify a single minority sample.

**Why pretraining methods with nonlinear dimension reduction (autoencoder), imbalance handling (DDPM/SMOTE+noise), feature selection (LASSO) can not solve this LH-P?**

This failure empirically validates the intractability of Learning-Hard Problems (LH-Ps) for global methods:

1. **Global Hypothesis Failure (Condition C1):** The pipeline attempts to learn a single global function $h \in \mathbb{H}$. LH-Ps are defined by the "near-universal failure" of such global models due to high intrinsic complexity (high LHI).

2. **Manifold Collapse:** Lasso and DAEs optimize global metrics (linear discriminability and MSE). In data imbalance regimes, this forces the model to prioritize the majority manifold, treating the topological minority structure as noise to be smoothed out.

3. **Missing Knowledge Fusion:** The approach lacks the query-specific operator $\varphi_\kappa$ (specifically, **Traininglet Construction**) required to satisfy **Condition C2 (Latent Solvability)**. By failing to localize the problem, the task remains in the intractable global regime.

4. **LHI Invariance:** Global dimensionality reduction cannot alter the intrinsic topological complexity. As verified in *Supplemental N*, the Learning-Hard Index (LHI) is statistically invariant across raw and projected spaces (e.g., t-SNE, UMAP), implying that the data remains in the intractable regime (LHI $\geq 0.80$) despite projection.

## Q.2: Autoencoder pretraining + DNN fine-tuning for COVID19 data

We compare the performance of autoencoder pretraining + DNN fine-tuning with MiL:

Table S3: Performance Comparison: Pretrain (Global) vs. MiL (Local) on COVID-19 Dataset

| Metric | Pretrain (Global Model) | MiL (Local) |
|---|---|---|
| Accuracy | 84.62% | **95.44%** |
| Sensitivity | 0.9000 | **0.9644** |
| Precision | 0.8910 | **0.9632** |
| F1 Score | 0.8918 | **0.9544** |
| d-index | 1.8133 | **1.9424** |

## Why Pretraining Falls Short on LH-P?

- **Global vs. Local:** The Pretrained Autoencoder learns a **single global manifold** for the entire dataset. According to your paper, LH-Ps (Learning-Hard Problems) often have complex, "tangled" decision boundaries (like the IRMAS t-SNE example) that a single global function—even a deep one— cannot resolve without overfitting.

- **Lack of Knowledge Fusion:** Pretraining is unsupervised (or supervised globally); it does not fuse **label-aware structural knowledge** specific to *each* test query.

- **The MiL Advantage:** MiL constructs a specific **traininglet** for every query, isolating a "low-capacity sweet spot" (Prop. 1 & 2) that makes the local problem trivial to solve, whereas the global model tries to solve the "hard" global problem all at once.

## Q.3: Autoencoder pretraining + DNN fine-tuning for IRAMS data

Table S4 contrasts the test performance of the global Pre-trained DAE (DNN fine-tuning) against the local Micro-Learning (MiL) framework. MiL outperforms the pre-training approach by a massive margin across all metrics.

Table S4: Test Performance Comparison on IRMAS (11 Classes)

| Metric | Pre-trained DAE + DNN | MiL | Performance Gap |
|---|---|---|---|
| D-index | 1.4622 | **1.8162** | +0.3540 |
| Accuracy | 56.90% | **84.31%** | +27.41% |
| Sensitivity | 55.53% | **83.87%** | +28.34% |
| Precision | 57.11% | **84.49%** | +27.38% |
| F1 Score | 55.94% | **84.31%** | +28.37% |

**Empirical Validation: Global Pre-training Leads to Severe Overfitting**  The experimental results on the IRMAS dataset provide a textbook example of why global deep learning architectures fail on Learning-Hard Problems (LH-Ps). As shown in the following Table, the Pre-trained Denoising Autoencoder (DAE) + DNN pipeline exhibits *a massive generalization gap*.

While the model achieves near-perfect performance on the training set (94.89% Accuracy), it collapses to 56.90% on the test set. This 38% drop confirms that the global model, forced to fit the high-complexity "tangled swirl" of the raw data (LHI $\approx$ 0.91), resorted to memorizing training noise and artifacts rather than learning robust, instrument-invariant features.

Table S5: Generalization Analysis on IRMAS: Pre-training vs MiL

| Model Architecture | Train Acc. | Test Acc. | Gen. Gap | Diagnosis |
|---|---|---|---|---|
| Global Pre-trained DAE | 94.89% | 56.90% | -37.99% | Severe Overfitting |
| **Micro-Learning (MiL)** | 88.93% | **84.31%** | **-4.62%** | **Robust Generalization** |

**Why serious overfitting?**  This overfitting validates *Proposition 1* in reverse. The global hypothesis space $\mathbb{H}$ required to fit the complex training data has an extremely high Rademacher complexity, leading to loose generalization bounds. By contrast, MiL avoids this overfitting not by regularization, but by localization: it solves a sequence of low-complexity problems (traininglets) where the local Rademacher complexity is naturally low, ensuring that high accuracy translates from training to test.

# Supplemental T: Extending Learning-Hard Problem (LH-P) to Regression: LH-R

## Learning-Hard Regression (LH-R): LH-P in regression

The LH-P framework extends naturally from classification to regression tasks with continuous targets $y \in \mathbb{R}$. Correspondingly, we extend corresponding traininglets and learninglets in this extension.

We define a *Learning-Hard Regression (LH-R)* problem as one where global models fail due to non-stationarity or high-frequency oscillation (Condition C1), yet the target function remains locally smooth within specific state-space pockets (Condition C2).

**Definition 4** (**Learning-Hard Regression (LH-R)**)**.** Consider a regression task with target $y \in \mathbb{R}$ and squared loss $L(\hat{y}, y) = (\hat{y} - y)^2$. For $h \in \mathbb{H}$ define the risk

$$R(h) = \mathbb{E}_{(x,y) \sim \mathcal{P}} \big[ (h(x) - y)^2 \big]. \tag{6}$$

The task is an *LH-R* problem with respect to $(\mathbb{H}, \mathcal{K})$ if there exist constants $0 < \tau \ll \tau^\star$ such that:

1. **(C1-Reg) Near-universal failure.**
$$\min_{h \in \mathbb{H}} R(h) \geq \tau^\star. \tag{7}$$

2. **(C2-Reg) Latent solvability.**
$$\exists \kappa \in \mathcal{K}, \ \exists h^\star \in \mathbb{H} \text{ s.t. } R\big(h^\star \circ \varphi_\kappa\big) \leq \tau. \tag{8}$$

**LH-R Interpretation:**  In the regression setting we predict a real-valued target $y \in \mathbb{R}$ using squared loss, so $R(h)$ measures the expected mean-squared error of a model $h \in \mathbb{H}$.

*Condition (C1):* even if we pick the best possible global model from a rich hypothesis class $\mathbb{H}$, its regression risk $R(h)$ is still large (at least $\tau^\star$): in other words, standard "vanilla" training fails on this task.

*Condition (C2):* there nevertheless exists some way of injecting knowledge into the input, via a preprocessing operator $\varphi_\kappa \in \mathcal{K}$, together with a model $h^\star \in \mathbb{H}$, such that the composed predictor $h^\star \circ \varphi_\kappa$ achieves very low mean-squared error (at most $\tau$). Thus *an LH-R problem is hard to any global learner, but becomes easy once the right knowledge-fused representation or local structure is used.*

## Regression Learning-Hard Index, Traininglets, and Learninglets

**Regression Learning-Hard Index** ($\text{LHI}_{\text{reg}}$)**.**  Analogous to the classification LHI, we quantify regression hardness by the ratio of local-to-global target variance. For a subset $T$, we define $\text{LHI}_{\text{reg}}(T) = \widehat{\text{Var}}(y \mid x \in T)/\widehat{\text{Var}}(y \mid x \in X)$. Values near 0 indicate a locally smooth, deterministic relationship, while values near 1 indicate local chaos indistinguishable from global noise.

**Regression Traininglets.**  For a query $x'$, we construct a regression traininglet $\mathcal{T}_{x'} \subset X$ by minimizing the pair $(\text{LHI}_{\text{reg}}(T), |T|)$ lexicographically, subject to three constraints:

- *Geometric Proximity:* $T \subseteq \mathcal{N}_k(x')$.
- *Target Smoothness (Sanitization):* We prune points $(x_i, y_i)$ violating the local Lipschitz condition, i.e., where $|y_i - \bar{y}_T|/\|x_i - \bar{x}_T\|$ exceeds a robust threshold.
- *Span Constraint:* We require $x' \in \text{conv}\{x_i \in T\}$, ensuring the query lies within the convex hull of the traininglet. This forces the local learner to perform *interpolation* rather than risky *extrapolation*.

**Regression Learninglets.** On the optimized $\mathcal{T}_{x'}$, MiL fits a Kernel Ridge Regression (KRR) or local SVR model:

$$f_{x'}(x) = \sum_{(x_i, y_i) \in \mathcal{T}_{x'}} \alpha_i K(x_i, x). \tag{9}$$

Because $\mathcal{T}_{x'}$ is constructed to have low intrinsic variance ($\text{LHI}_{\text{reg}} \ll 1$) and satisfy the convex hull property, this simple local model provides robust generalization even when the global function is intractable.

# Supplemental U: Comparing MiL with LNN (Liquid Neural Network (LNN) on benchmarks

### Liquid neural networks (LNN)

Liquid neural networks (LNN) are compact and highly expressive for modeling complex temporal dynamics with relatively few parameters. A liquid neural network models each neuron as a dynamical system with input-dependent, time-varying parameters, allowing the network's behavior to continuously adapt to changing inputs.

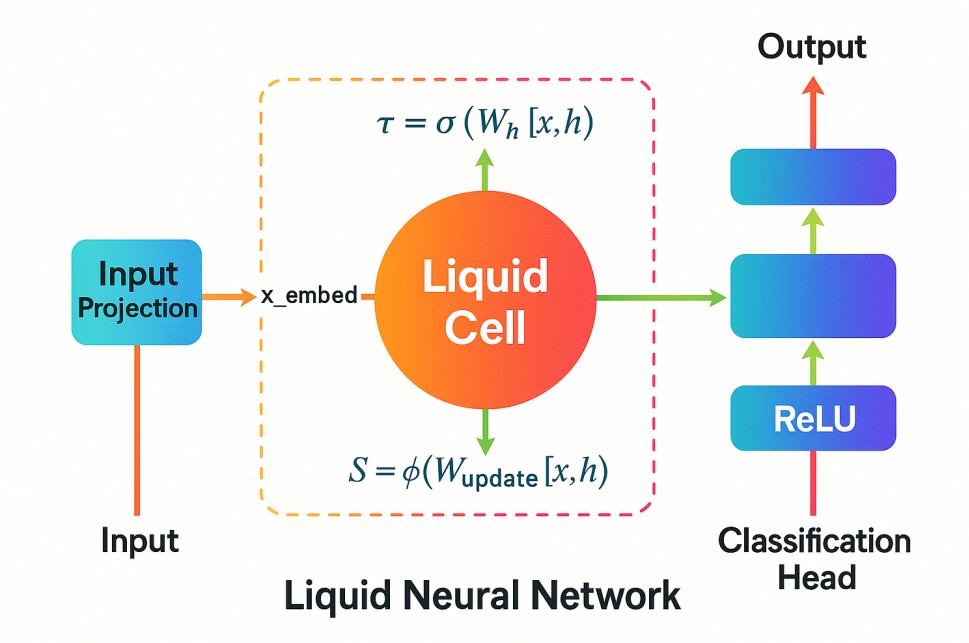

Figure 7: Overview of the Liquid Neural Network (LNN) baseline. The input features are first mapped to an embedding through an input projection layer and then processed by a Liquid Cell, whose dynamics are governed by time-varying parameters $\tau = \sigma(W_h[x, h])$ and $S = \phi(W_{\text{update}}[x, h])$. The resulting representation is passed to a ReLU-based classification head to produce the final output.

Table S6: Performance of LNN on five learning-hard benchmarks

| Dataset | D-index | Acc | Sen | Prec | F1 |
|---------|---------|-----|-----|------|-----|
| IRMAS | 1.5216 | 0.6124 | 0.6003 | 0.6036 | 0.6003 |
| CASIA | 1.7527 | 0.7958 | 0.7958 | 0.8025 | 0.7962 |
| SAVEE | 1.5888 | 0.6750 | 0.6476 | 0.6592 | 0.6448 |
| COVID19 | 1.8428 | 0.8751 | 0.9083 | 0.9123 | 0.9080 |
| Ovarian | 1.5580 | 0.9630 | 0.5000 | 0.4815 | 0.4906 |

## MiL wins LNN statistically

We compared MiL and LNN across five benchmarks (IRMAS, CASIA, SAVEE, COVID-19, Ovarian) using a common set of metrics: D-index, accuracy, sensitivity, precision, and F1. MiL achieved higher scores than LNN in all 25 dataset–metric pairs. A non-parametric paired sign test over these 25 comparisons yielded a highly significant result ($p \approx 5.9 \times 10^{-8}$), indicating that such a consistent pattern of performance improvement is extremely unlikely under the null hypothesis.

Furthermore, treating the 25 MiL and 25 LNN values as independent samples, a Mann–Whitney U-test confirmed that MiL's performance distribution is significantly superior ($U = 442$, $p \approx 0.012$, two-sided). Additionally, a focused one-sided sign test on the **D-index** metric alone ($N = 5$) showed a statistically significant improvement ($p \approx 0.031$), validating MiL's advantage in separability.

## Why LNN falters on LH-Ps?

Although the Liquid Neural Network (LNN) is a powerful dynamical baseline, it still behaves as a single global model and thus inherits the core failure modes of deep learners on LH-Ps.

First, LNN compresses each example into one evolving hidden state optimized under a global cross-entropy loss, so heterogeneous or rare regimes are averaged away; on highly imbalanced LH-Ps (e.g., Ovarian) this yields high accuracy but very low sensitivity and F1, indicating that the liquid dynamics mainly fit the majority class.

Second, LNN lacks any mechanism for *label-aware structural knowledge fusion*: it does not construct query-specific neighborhoods or traininglets, and therefore cannot move into the low-complexity "sweet-spot" regions guaranteed by our theory, making it vulnerable to noise and distribution shift when LHI is high.

Third, the strongly nonlinear continuous-time dynamics of the liquid cell, trained end-to-end with stochastic optimization, amplify small perturbations in small or noisy datasets, leading to unstable training and poor reproducibility, in contrast to MiL's deterministic RKHS learninglets on carefully sanitized traininglets.

Together, these factors explain why LNN underperforms MiL across all five benchmarks despite being a sophisticated modern architecture.

# Supplemental V: SC-let performance

Table S7: Overall performance of SC-let (traininglet 50: with 50 traininglets per query)

|  | Accuracy | Sensitivity | Specificity | Precision | F1-score | D-index |
|---|---|---|---|---|---|---|
| SC-let: Traininglet-50 | 80.24% | 80.24% | 99.80% | 80.39% | 80.23% | 1.4919 |

Table S8: Per-class classification report for SC-let (with 50 traininglets per query)

| Class | Precision | Recall | F1-score | Support |
|---|---|---|---|---|
| 0 | 0.9278 | 0.9000 | 0.9137 | 100 |
| 1 | 0.9286 | 0.9100 | 0.9192 | 100 |
| 2 | 0.6900 | 0.6900 | 0.6900 | 100 |
| 3 | 0.7172 | 0.7100 | 0.7136 | 100 |
| 4 | 0.7358 | 0.7800 | 0.7573 | 100 |
| 5 | 0.7921 | 0.8000 | 0.7960 | 100 |
| 6 | 0.8381 | 0.8800 | 0.8585 | 100 |
| 7 | 0.8085 | 0.7600 | 0.7835 | 100 |
| 8 | 0.9314 | 0.9500 | 0.9406 | 100 |
| 9 | 0.9271 | 0.8900 | 0.9082 | 100 |
| 10 | 0.6709 | 0.5300 | 0.5922 | 100 |
| 11 | 0.5052 | 0.4900 | 0.4975 | 100 |
| 12 | 0.8519 | 0.9200 | 0.8846 | 100 |
| 13 | 0.7071 | 0.7000 | 0.7035 | 100 |
| 14 | 0.8947 | 0.8500 | 0.8718 | 100 |
| 15 | 0.8788 | 0.8700 | 0.8744 | 100 |
| 16 | 0.8352 | 0.7600 | 0.7958 | 100 |
| 17 | 0.8812 | 0.8900 | 0.8856 | 100 |
| 18 | 0.8316 | 0.7900 | 0.8103 | 100 |
| 19 | 0.8298 | 0.7800 | 0.8041 | 100 |
| 20 | 0.9029 | 0.9300 | 0.9163 | 100 |
| 21 | 0.9394 | 0.9300 | 0.9347 | 100 |
| 22 | 0.8396 | 0.8900 | 0.8641 | 100 |
| 23 | 0.8763 | 0.8500 | 0.8629 | 100 |
| 24 | 0.8737 | 0.8300 | 0.8513 | 100 |
| 25 | 0.7500 | 0.7200 | 0.7347 | 100 |
| 26 | 0.8125 | 0.7800 | 0.7959 | 100 |
| 27 | 0.7030 | 0.7100 | 0.7065 | 100 |
| 28 | 0.8673 | 0.8500 | 0.8586 | 100 |
| 29 | 0.8454 | 0.8200 | 0.8325 | 100 |
| 30 | 0.7647 | 0.7800 | 0.7723 | 100 |
| 31 | 0.8989 | 0.8000 | 0.8466 | 100 |
| 32 | 0.7500 | 0.7500 | 0.7500 | 100 |
| 33 | 0.8409 | 0.7400 | 0.7872 | 100 |
| 34 | 0.8318 | 0.8900 | 0.8599 | 100 |
| 35 | 0.4639 | 0.4500 | 0.4569 | 100 |
| 36 | 0.8137 | 0.8300 | 0.8218 | 100 |
| 37 | 0.8842 | 0.8400 | 0.8615 | 100 |
| 38 | 0.7407 | 0.8000 | 0.7692 | 100 |
| 39 | 0.9020 | 0.9200 | 0.9109 | 100 |
| 40 | 0.7736 | 0.8200 | 0.7961 | 100 |
| 41 | 0.9394 | 0.9300 | 0.9347 | 100 |
| 42 | 0.8191 | 0.7700 | 0.7938 | 100 |
| 43 | 0.9255 | 0.8700 | 0.8969 | 100 |
| 44 | 0.5920 | 0.7400 | 0.6578 | 100 |
| 45 | 0.7182 | 0.7900 | 0.7524 | 100 |
| 46 | 0.6000 | 0.6900 | 0.6419 | 100 |
| 47 | 0.7059 | 0.6000 | 0.6486 | 100 |
| 48 | 0.9320 | 0.9600 | 0.9458 | 100 |
| 49 | 0.8304 | 0.9300 | 0.8774 | 100 |
| 50 | 0.6667 | 0.6400 | 0.6531 | 100 |
| 51 | 0.8632 | 0.8200 | 0.8410 | 100 |
| 52 | 0.6186 | 0.7300 | 0.6697 | 100 |
| 53 | 0.9020 | 0.9200 | 0.9109 | 100 |
| 54 | 0.8558 | 0.8900 | 0.8725 | 100 |
| 55 | 0.5842 | 0.5900 | 0.5871 | 100 |
| 56 | 0.9029 | 0.9300 | 0.9163 | 100 |
| 57 | 0.7961 | 0.8200 | 0.8079 | 100 |
| 58 | 0.9400 | 0.9400 | 0.9400 | 100 |
| 59 | 0.6634 | 0.6700 | 0.6667 | 100 |
| 60 | 0.8131 | 0.8700 | 0.8406 | 100 |
| 61 | 0.7609 | 0.7000 | 0.7292 | 100 |
| 62 | 0.7685 | 0.8300 | 0.7981 | 100 |
| 63 | 0.8667 | 0.7800 | 0.8211 | 100 |
| 64 | 0.6989 | 0.6500 | 0.6736 | 100 |
| 65 | 0.7396 | 0.7100 | 0.7245 | 100 |
| 66 | 0.9082 | 0.8900 | 0.8990 | 100 |
| 67 | 0.7957 | 0.7400 | 0.7668 | 100 |
| 68 | 0.9135 | 0.9500 | 0.9314 | 100 |
| 69 | 0.9140 | 0.8500 | 0.8808 | 100 |
| 70 | 0.8469 | 0.8300 | 0.8384 | 100 |
| 71 | 0.8652 | 0.7700 | 0.8148 | 100 |
| 72 | 0.5943 | 0.6300 | 0.6117 | 100 |
| 73 | 0.7143 | 0.7000 | 0.7071 | 100 |
| 74 | 0.5370 | 0.5800 | 0.5577 | 100 |
| 75 | 0.9286 | 0.9100 | 0.9192 | 100 |
| 76 | 0.9192 | 0.9100 | 0.9146 | 100 |
| 77 | 0.8182 | 0.8100 | 0.8141 | 100 |
| 78 | 0.6636 | 0.7300 | 0.6952 | 100 |
| 79 | 0.8019 | 0.8500 | 0.8252 | 100 |
| 80 | 0.7573 | 0.7800 | 0.7685 | 100 |
| 81 | 0.7573 | 0.7800 | 0.7685 | 100 |
| 82 | 0.9320 | 0.9600 | 0.9458 | 100 |
| 83 | 0.7959 | 0.7800 | 0.7879 | 100 |
| 84 | 0.7900 | 0.7900 | 0.7900 | 100 |
| 85 | 0.9314 | 0.9500 | 0.9406 | 100 |
| 86 | 0.8037 | 0.8600 | 0.8309 | 100 |
| 87 | 0.8393 | 0.9400 | 0.8868 | 100 |
| 88 | 0.8654 | 0.9000 | 0.8824 | 100 |
| 89 | 0.8879 | 0.9500 | 0.9179 | 100 |
| 90 | 0.7944 | 0.8500 | 0.8213 | 100 |
| 91 | 0.8866 | 0.8600 | 0.8731 | 100 |
| 92 | 0.7500 | 0.6600 | 0.7021 | 100 |
| 93 | 0.7917 | 0.7600 | 0.7755 | 100 |
| 94 | 0.9485 | 0.9200 | 0.9340 | 100 |
| 95 | 0.8021 | 0.7700 | 0.7857 | 100 |
| 96 | 0.7129 | 0.7200 | 0.7164 | 100 |
| 97 | 0.8947 | 0.8500 | 0.8718 | 100 |
| 98 | 0.6436 | 0.6500 | 0.6468 | 100 |
| 99 | 0.8265 | 0.8100 | 0.8182 | 100 |

# Supplemental X: Micro-Learning (MiL) FAQ

1. **What is a Learning-Hard Problem (LH-P)?**

   - An LH-P is a task for which almost all "vanilla" models in a broad hypothesis space perform poorly (near-universal failure), yet there exists at least one model that can perform well once appropriate label-aware structural knowledge is injected into training (latent solvability).

   - The difficulty stems from the data's intrinsic topological complexity, not from a lack of representational capacity in standard architectures.

2. **Can you simulate a baby LH-P and show how MiL can solve it?**

To rigorously validate our theoretical framework, we simulate a canonical Learning-Hard Problem (LH-P) termed the *Poisoned Spiral* (Fig. 8).

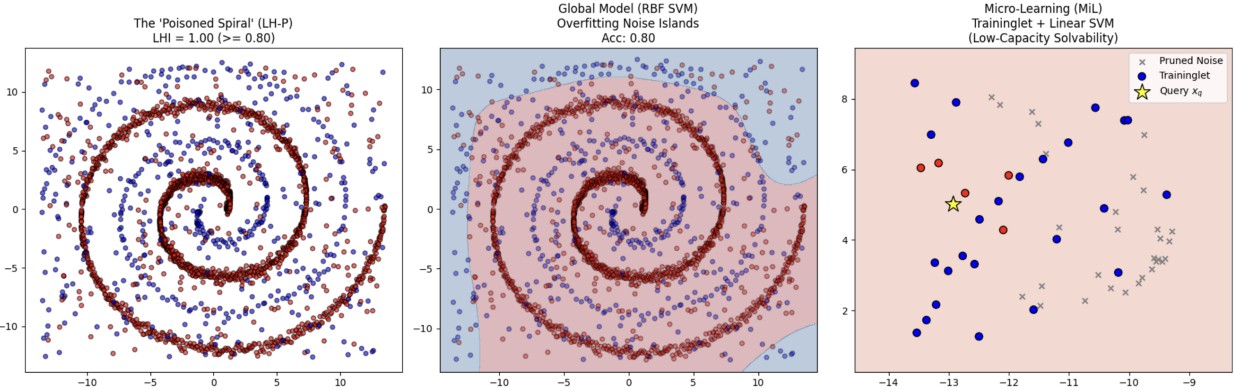

Figure 8: **Synthetic Validation of LH-P Theory: The 'Poisoned Spiral'.**

- **Data Generation (left plot)** We simulate a Learning-Hard Problem ($N = 2,250$) comprising two intertwining manifolds with heavy label noise. The dataset is dominated by the majority class (1,762/2,250 samples), resulting in a **majority class prevalence of $\approx$ 78.3%**. This dataset has a Learning-Hard Index (LHI) of 1.00.

- **Condition C1: Near-Universal Failure (mid plot):** A global high-capacity model (RBF SVM) achieves 0.80 accuracy: *the majority type hijack the whole learning so that accuracy is close to the majority type ratio.* This is a **trivial gain** over the 78% majority prevalence, proving the model failed to learn the minority structure and instead overfitted the "noise islands."

  The decision boundary (visualized in blue/red background) exhibits severe overfitting, fragmenting into disjoint "noise islands" rather than learning the continuous spiral, proving that global capacity cannot overcome intrinsic topological complexity.

- **Condition C2: Latent Solvability (right plot):** Micro-Learning (MiL) applied to query $x_q$ (star). The algorithm performs sanitization (pruning noise points shown as gray ×'s), revealing a *traininglet* that is linearly separable. This validates Proposition 1: *even when global capacity fails, a low-capacity witness exists locally.*

  Through our sanitization process, MiL successfully prunes the bridge noise (gray ×'s) that confounded the global model. On the resulting *traininglet* (blue/red circles), the decision boundary becomes linearly separable. A simple Linear SVM (low-capacity *learninglet*) solves the local task perfectly, validating Proposition 1: even when the global problem is intractable, a low-capacity witness exists locally if the data is correctly fused.

3. **Can preprocessing techniques such as imbalance handling (e.g.,SMOTE, loss weighting, etc) help to solve an LH-P with data imbalance?**

No. Because the minority manifold is topologically entangled with noise, standard methods indiscriminately amplify this noise, cementing the entanglement rather than resolving it.

- **Imbalance Handling** These methods assume that the minority class neighbors are semantically valid. In an LH-P, a minority sample's Euclidean neighbors are often noise ("poison"). *Interpolating between them (SMOTE) or increasing their weight merely amplifies the noise, exacerbating Condition C1 (Near-Universal Failure).* Thus, they are methods included in general ML/DL methods in our Condition C1.

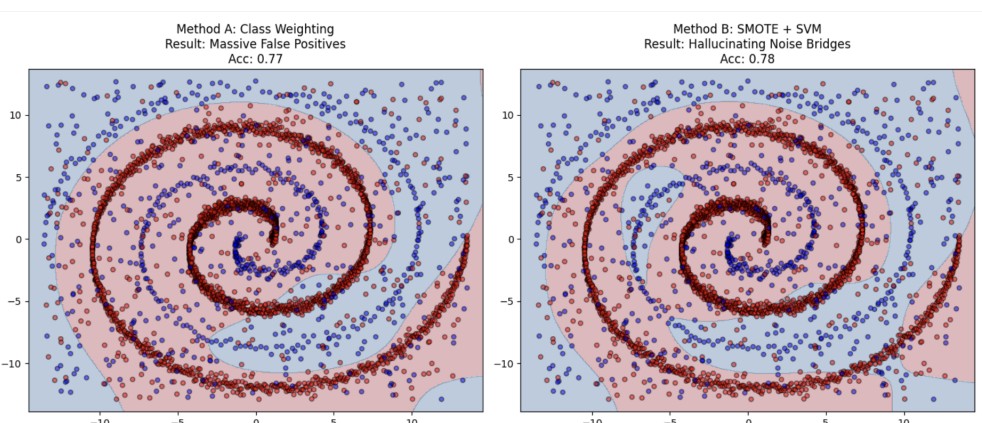

Figure 9: **Failure of Imbalance Handling Baselines on the 'Poisoned Spiral'. (Left) Class Weighting:** Heavily penalizing minority errors forces the SVM to overfit the bridge noise, creating massive "False Positive Islands" (pink regions) that engulf the majority class, degrading precision. **(Right) SMOTE:** Synthetic oversampling interpolates between minority points and nearby noise, effectively cementing the "bridge noise" into the decision boundary. Both methods fail to recover the clean spiral structure achieved by MiL (Fig. 8).

**Failure of Standard Imbalance Handling on LH-Ps.** We investigate whether standard imbalance remedies—Class Weighting and SMOTE—can resolve the Learning-Hard Problem. As shown in Figure 9, both methods fail catastrophically, albeit in different ways.

**(Left) Class Weighting:** By heavily penalizing errors on the minority class, the SVM expands the decision boundary aggressively. However, because the minority manifold is surrounded by "bridge noise," this expansion engulfs the noise, creating massive False Positive regions (pink) that swallow large portions of the majority class (blue points). The result is a degradation in overall accuracy (0.77) below the zero-skill baseline.

**(Right) SMOTE:** Synthetic Minority Over-sampling Technique (SMOTE) interpolates between minority samples. In an LH-P, minority samples are often Euclidean neighbors with noise points. SMOTE blindly generates synthetic samples along these noise bridges, effectively cementing the noise into the decision boundary. The resulting model "hallucinates" a connection between the spirals, failing to separate the manifolds and achieving a trivial accuracy of 0.78.

4. **Can generative data augmentation (e.g., Diffusion Models, GANs) solve an LH-P with data imbalance?**

No. Generative models are designed to approximate the underlying data distribution $P(X|y)$. In an LH-P, the minority class distribution $P(X|y = 1)$ is topologically entangled with noise.

We trained a Denoising Diffusion Probabilistic Model (DDPM) on the minority class to generate synthetic samples. As shown in Figure 10, this approach fails due to the **Fidelity Paradox**.

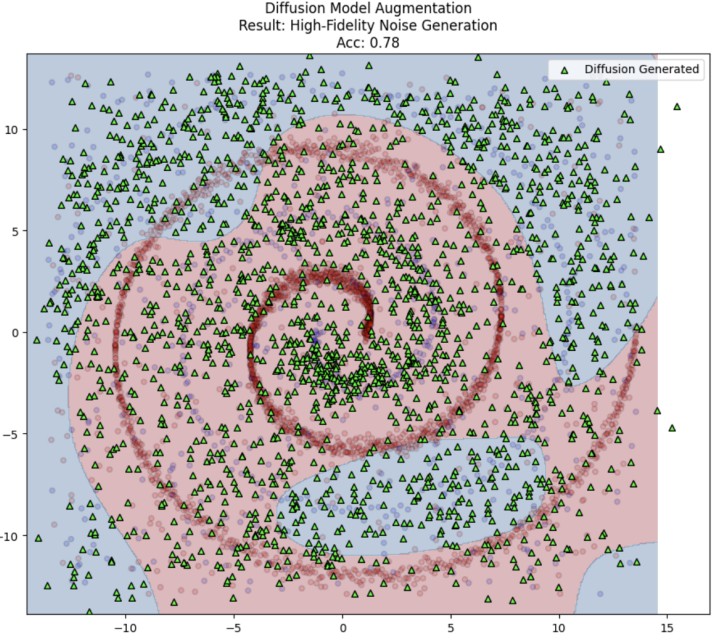

Figure 10: **Failure of Diffusion Model Augmentation.** We trained a DDPM on the minority class of the Poisoned Spiral. The generated synthetic samples (green triangles) faithfully reproduce the training distribution, which includes the "bridge noise." Consequently, the augmented dataset reinforces the class overlap, preventing the classifier from finding a clean margin and yielding near-baseline accuracy.

**Why diffusion models can't work for LH-Ps?** Diffusion models are optimized to approximate the data distribution $p_{data}(x)$. In an LH-P, the minority class distribution is a mixture of the true manifold and topological noise: $p_{minority} = (1 - \alpha)p_{signal} + \alpha p_{noise}$. Because the diffusion model is faithful to the training data, it learns to generate samples from $p_{noise}$ just as well as $p_{signal}$. It cannot distinguish "poison" from "structure" without the explicit sanitization logic provided by MiL.

**2. Empirical Evidence (Fig. 10):** The synthetic samples generated by the Diffusion model (green triangles) accurately reproduce the topology of the minority class—*including the bridge noise*.

- **Noise Reinforcement:** The model generates new points in the "bridge regions" between the spirals, effectively increasing the density of the noise.

- **Result:** The SVM trained on this augmented data achieves only 0.78 accuracy. The decision boundary remains jagged and overfitted, proving that generative oversampling cannot resolve topological entanglement.

5. **Can advanced manifold learning (e.g., t-SNE, PHATE) combined with SMOTE solve the problem?**

No. While non-linear dimensionality reduction is powerful, it fails on LH-Ps because of the **Topology Preservation Paradox**. The Topology Preservation Paradox arises because manifold learning algorithms faithfully preserve local neighborhood structures, *which inevitably means preserving the "bridge noise" that entangles distinct classes, thereby reinforcing the topological corruption rather than resolving it.*

**1. Theoretical Failure Mode:** Algorithms like t-SNE and PHATE are designed to preserve local neighborhood structures.

- **Preserving the Poison:** In an LH-P, the "bridge noise" is locally indistinguishable from the signal in the high-dimensional space. t-SNE faithfully preserves these local connections. It maps the noise points directly adjacent to (or inside) the minority clusters in the embedding space.

- **The SMOTE Trap:** When SMOTE is applied to this embedding, it interpolates between the minority points and the preserved noise neighbors. This solidifies the entanglement, creating a decision boundary that zig-zags arbitrarily around the noise rather than separating the manifolds.

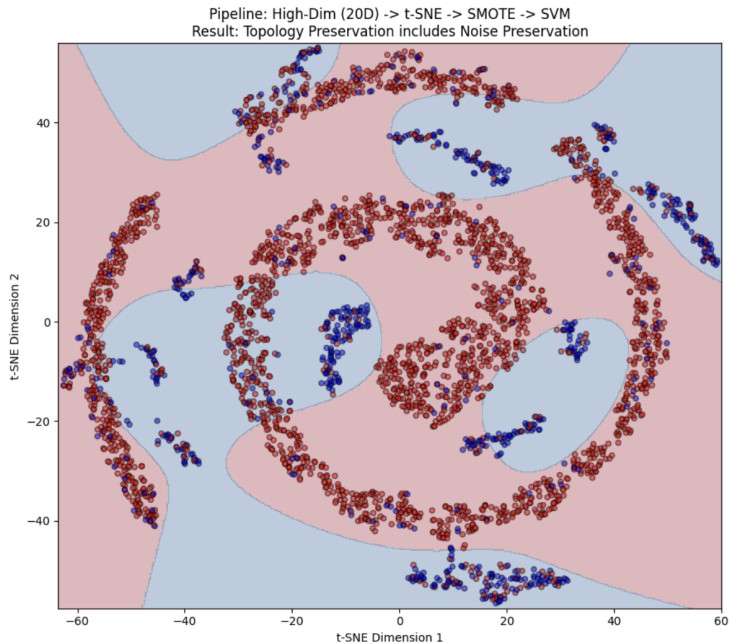

Figure 11: **Failure of t-SNE + SMOTE on High-Dimensional LH-P.** We applied t-SNE to the 20-dimensional Poisoned Spiral, followed by SMOTE and an RBF SVM. **Result:** t-SNE successfully unrolls parts of the spiral but, crucially, it preserves the "bridge noise" (mixed red/blue regions) because the noise is topologically connected to the manifolds. The SVM overfits these mixed regions, failing to find a clean margin. This confirms that manifold learning cannot sanitize data; it only visualizes the existing complexity.

6. **Can MiL outperform a Global RBF-SVM on the full 'Poisoned Spiral' dataset with LHI of** 1.0**?**

Yes. We compare the performance of a Global RBF-SVM against Micro-Learning (MiL) on the full dataset ($N = 2,250$). As shown in Figure 12, MiL demonstrates a decisive advantage in recovering the latent minority structure.

**1. Global Failure (Left Panel):** The Global RBF-SVM achieves a Total Accuracy of 0.81, *but this metric is deceptive and biased.* The model achieves this by prioritizing the majority class and "giving up" on the complex inner spiral. This is evidenced by the abysmal Sensitivity of 0.24. The decision boundary (background color) fails to penetrate the spiral arms, collapsing the minority manifold into the majority class.

**2. MiL Success (Right Panel):** Micro-Learning achieves a Total Accuracy of 0.89 and a Sensitivity of 0.74.

- **Structural Recovery:** Unlike the global model, MiL's decision boundary successfully traces the inner windings of the blue spiral.

- **The Theoretical Limit:** Note that an accuracy of $\approx 0.89$ is the theoretical maximum (Bayes Error) for this dataset, as the "poison" noise points have random labels that cannot be predicted. MiL hitting this limit while tripling the Sensitivity (from 0.24 to 0.74) confirms it has solved the topological problem.

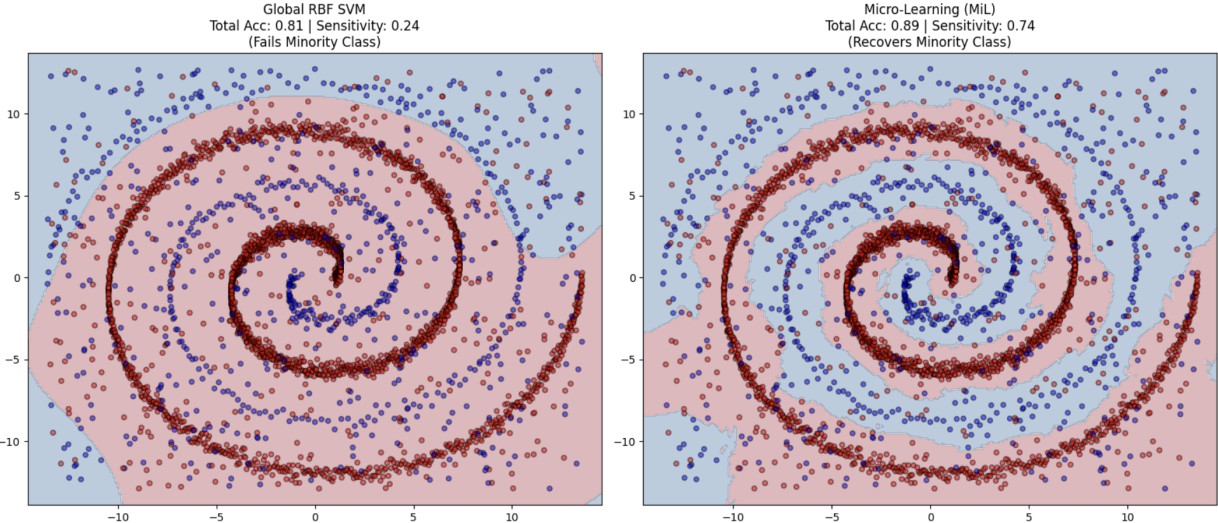

Figure 12: **MiL vs. Global SVM on the Poisoned Spiral (Whole Data). (Left)** The Global RBF SVM is overwhelmed by bridge noise. It defaults to predicting the majority class in the difficult inner regions, resulting in a failure to detect the minority spiral (Sensitivity: 0.24). **(Right)** Micro-Learning adapts locally to the manifold. It successfully recovers the continuous spiral structure, *tripling the Sensitivity to 0.74* and achieving the theoretical maximum accuracy (0.89) allowed by the label noise.

7. **What is Micro-Learning (MiL) in one sentence?**

   MiL is a framework that, for each query, builds a tiny, knowledge-fused subset of the training data (a *traininglet*) and fits an explainable, deterministic local model (a *learninglet*) on it, instead of relying on a single global black-box model.

   Mil partitions decision space as union of many explainable and deterministic local decision functions, achieving overfitting-resistant by avoiding a global deciosn function generalization.

8. **How is MiL different from standard deep learning?**

   Deep networks learn one global, highly nonlinear function over the entire dataset, which can easily overfit and is hard to interpret or reproduce. MiL instead learns many small, local models on low-complexity traininglets, which are easier to analyze, more robust to overfitting, and deterministic.

9. **What is the Learning-Hard Index (LHI)?**

   LHI is a scalar in $[0,1]$ that quantifies how misaligned the geometric structure of a dataset is with its label structure. It is defined as $\text{LHI}(X) = 1 - \text{AMI}(y, y_p)$, where AMI is the Adjusted Mutual Information between true labels $y$ and pseudo-labels $y_p$ obtained from clustering a local embedding (e.g., t-SNE/UMAP).

10. **How do you interpret LHI values?**

    Low LHI ($\lesssim 0.5$) indicates that clusters in feature space align reasonably well with labels, so standard models often work. Those falling between 0.5 to 0.8 can be solved tuned existing DL/ML models. High LHI ($\gtrsim 0.8$) indicates severe manifold entanglement: local neighborhoods mix labels heavily, signaling a Learning-Hard Problem, which should be solved by MiL.

Table S7: Learning hardness categories by LHI (where $\text{LHI} \in [0,1]$).

| Category | LHI range | Quick guidance |
|---|---|---|
| Learning-able | LHI < 0.50 | Standard pipelines usually suffice |
| Medium-learnable | LHI $\in$ [0.50, 0.80) | Use tuned existing ML/DL models |
| Learning-hard | LHI $\in$ [0.80, 1] | MiL |

11. **Why is** $0.80$ **chosen as the LH-P threshold?**

    By definition, $\text{LHI}(X) = 1 - \text{AMI}(X_r, X_p)$, so $\text{LHI}(X) \geq 0.80$ means the clustering preserves at most 20% of the neighborhood mutual information between labels and the unsupervised embedding. In this regime, local geometry is largely label-agnostic and the learner must effectively reconstruct the missing 80% of mutual information from limited labeled data, a **learning-hard** setting where standard deep networks struggle

    Empirically, across ten diverse datasets we observed a strong negative correlation between LHI and DNN performance, with a clear phase transition around LHI $\approx 0.80$. Above this threshold, performance of standard deep models degrades sharply, making specialized methods like MiL necessary.

12. **What are "quasi-LH-Ps"?**

    Quasi-LH-Ps are tasks with LHI $\in [0.75, 0.80)$: they still exhibit near-universal failure, but to a lesser degree. Strong, well-tuned global models sometimes work, though MiL typically still provides a robustness and interpretability advantage.

13. **What is a traininglet?**

    A traininglet $\mathcal{T}_{x'}$ is a small subset of the training data, tailored to a particular query $x'$, that (i) covers all labels, (ii) has low LHI (low intrinsic complexity), and (iii) is as small as possible under these constraints. It is the core object MiL uses to "localize" learning, i.e., *learn locally*

14. **What is a learninglet?**
A learninglet is the local predictor trained on a traininglet, typically a multiclass SVM or an extension an extension such as an SVM-micro-CNN-let (SC-let). Its SC-let variant couples a compact CNN-like feature extractor (e.g., ResNET) with a linear SVM head, preserving deterministic, convex optimization and support-vector-based interpretability while adding local representation-learning capacity.

It is deterministic, convex to optimize (for SVM), and its decision function is explainable in terms of support vectors in the traininglet.

15. **Why not just use $k$-NN instead of MiL?**
k-NN is a purely vote-based, non-parametric smoother: it just aggregates labels in a raw metric ball and never learns an explicit decision function or actively reduces local hypothesis complexity.

In contrast, MiL first sculpts refined, label-aware traininglets (via sanitization, rebalancing, and meta-fusion) that lower LHI and local Rademacher complexity, and then fits a local margin-maximizing classifier in an RKHS, yielding a stable decision boundary that can disentangle nearby but conflicting manifolds that k-NN simply averages over.

16. **MiL is a local SVM?**
No. MiL is much more than a local SVM: a "local SVM" would simply fit an SVM on a fixed geometric neighborhood, whereas MiL defines a full framework that performs label-aware structural knowledge fusion to construct low-complexity, query-specific traininglets with theoretical guarantees. The extension of SC-let makes MiL can handle high-dimensional large scale image data well.

The core MiL novelty and performance gains come from how MiL builds, sanitizes, and exploits these traininglets, not from the choice of SVM, which is a learning-let in MiL, itself.

17. **What does it mean that traininglet selection is NP-complete?**
The decision problem TRAININGLET-DEC: deciding whether there exists a subset of size at most $b$ with LHI at most $\ell$ and full label coverage, is NP-complete. This implies that finding the exact lexicographically optimal traininglet is NP-hard, motivating heuristic constructions like NTC and PTC.

18. **Does NP-completeness make MiL impractical?**
No. NP-completeness applies to the *optimal* traininglet selection problem, not to MiL's concrete heuristics. MiL uses efficient approximations (NTC and PTC) that are tractable for small and mid-sized datasets and empirically produce high-quality traininglets.

19. **What is Naïve Traininglet Construction (NTC)?**
NTC builds a traininglet for a query by intersecting multiple metric balls around it (e.g., Euclidean + correlation) so that retained points are consistently close across several geometric views. If some labels are missing, it rebalances by adding the nearest sample of each missing label.

20. **When is NTC sufficient?**
NTC works well when the dataset is relatively large, clean, and not extremely imbalanced, and when local geometry is already reasonably informative (e.g., IRMAS, CIFAR-100 feature space). For smaller, noisier, or highly imbalanced LH-Ps, PTC is preferable.

21. **What is Precision Traininglet Construction (PTC)?**
PTC is a four-stage pipeline that constructs high-quality traininglets for each query: (1) probing learning to choose good neighborhood and batch parameters, (2) training sanitization to remove "bad" samples and their neighbors, (3) meta-traininglet fusion combining four complementary local sets, and (4) precision pruning to remove residual noisy or adversarial points.

22. **How does PTC use "probing learning"?**

Probing learning runs Naïve-MiL over multiple random splits and a grid of $(k, z)$ to maximize a multi-class D-index. It keeps only non-dominated $(k^\star, z^\star)$ pairs, giving a data-driven, label-aware calibration of neighborhood size and batch size.

23. **What is training sanitization?**

Training sanitization uses Naive-MiL with $(k^\star, z^\star)$ to label each training point, then partitions them into "good guys" (correctly predicted) and "bad guys" (misclassified). It removes bad points and their local neighborhoods, reducing noise and LHI while preserving full label coverage via a minority-class safeguard.

24. **What are meta-traininglets in PTC?**

For each query, PTC builds four meta-traininglets: (1) a local ball via NTC on cleaned data, (2) a 1-hop transfer set from nearest "good guys", (3) a 2-hop transfer set giving broader manifold context, and (4) a random anchor traininglet to plug residual topology gaps. Their union is then pruned into the final traininglet.

25. **What is "label-aware structural knowledge fusion" in MiL?**

It is the process of constructing traininglets using both label information (which points are likely informative vs. noisy) and structural cues (geometric/semantic proximity), rather than blindly using all neighbors. This fusion is done via NTC or PTC according to different data. Such knowledge fusion allows MiL to reach the *low-capacity "sweet-spot" regions* guaranteed by the theory developed in prop 2 in the paper.

26. **How does MiL reduce overfitting compared to deep networks?**

MiL does this by *'learning locally.'* MiL avoids fitting one high-capacity model to a globally tangled dataset. Instead, it repeatedly fits low-capacity local models on small traininglets with reduced LHI and local Rademacher complexity, leading to tighter generalization bounds and smaller train–test gaps.

27. **How is local Rademacher complexity used in the theory?**

Proposition 1 guarantees that, in any hypothesis class, there exists a model $f^\star$ whose local neighborhood has minimal Rademacher complexity. MiL is designed to operate in such low-capacity neighborhoods by operating on carefully chosen traininglets, thereby achieving the tightest generalization bounds available.

28. **What is Proposition 2 (traininglet sufficiency) intuitively saying?**

It states that, for any test point, there exists some subset of the training data (a traininglet) such that a model trained only on that subset is *more likely* to match the Bayes-optimal prediction than any model trained on the full dataset. This formally justifies MiL's local-subset strategy.

29. **How does MiL handle distribution shift?**

PTC systematically removes misaligned, noisy regions and constructs query-specific traininglets that better match the local test distribution. Proposition 3 shows that, in total variation distance, the distribution induced by MiL's traininglets is strictly closer to the test distribution than the original training distribution.

30. **Why are MiL predictions reproducible?**

MiL's learninglets are based on deterministic optimization (e.g., convex SVM) over deterministic traininglets. There is no random initialization or non-convex training in the core decision rule, so repeated runs with the same data and hyperparameters produce identical predictions.

31. **How is MiL explainable?**
In an SVM-based learninglet, the decision for a query can be written as a weighted sum of kernel similarities between the query and a small set of support vectors in the traininglet. This gives a direct, instance-level explanation: a prediction can be traced to a handful of labeled examples that "pulled" the decision.

32. **What is an SVM-micro-CNN-let (SC-let)?**
An SC-let combines a compact CNN backbone (e.g., a small ResNet) to learn a feature representation, with a linear SVM head in the learned feature space. Distance computations for NTC/PTC are done in this semantic feature space, and the SVM head preserves margin-based explainability and determinism.

33. **Can MiL work on image data?**
Yes. For image tasks like CIFAR-100, MiL uses SC-lets: a CNN like model (e.g., small ResNET) maps images into a feature space where NTC builds traininglets and an SVM head makes predictions. Using small traininglets (e.g., size 50), MiL reaches $\approx 80\%$ accuracy on CIFAR-100, while largely retaining interpretability and reproducibility.

34. **How does MiL compare to standard CNNs, RNNs or variants empirically?**
Across five benchmarks (IRMAS, CASIA, SAVEE, COVID-19, Ovarian), MiL consistently outperforms 11 deep learning baselines (CNNs, LSTMs, GRUs, Bi-RNNs, Conv-RNNs, CapsNet) on accuracy, F1, and D-index, with particularly large gains on small-sample and imbalanced datasets.

35. **How does MiL compare to meta-learning methods like MAML and ProtoNet?**
MiL significantly outperforms both MAML and ProtoNet on all five benchmarks, especially on small-sample tasks like COVID-19 and Ovarian where meta-learners struggle. A paired statistical analysis shows MiL's accuracy and D-index are consistently higher with large effect sizes.

36. **How does MiL compare to liquid neural networks (LNN)?**
MiL consistently outperforms liquid neural networks (LNN) on all five benchmarks, with this advantage confirmed by a paired sign test ($p \approx 5.9 \times 10^{-8}$) and a Mann–Whitney U-test ($U = 442$, $p \approx 0.012$).

A focused one-sided sign test on the D-index alone ($N = 5$, $p \approx 0.031$) further shows that MiL yields more separable decision boundaries than LNN on learning-hard benchmarks.

37. **Why liquid neural networks (LNN) falters on LH-Ps?**
LNN remains a single global dynamical model that averages over rare or heterogeneous regimes on imbalanced LH-Ps and lacks any label-aware structural knowledge fusion or query-specific traininglets to reach the low-complexity "sweet-spot" regions MiL targets.

Its strongly nonlinear continuous-time dynamics, trained with stochastic optimization, also make it sensitive to noise and initialization, leading to less stable and less reproducible behavior than MiL.

38. **Why do meta-learning methods underperform on LH-Ps?**
Meta-learners learn global priors or metrics and still operate on the full, high-LHI distribution without systematically reducing local complexity. They do not construct query-specific traininglets and thus cannot exploit the low-capacity sweet spot guaranteed by the MiL theory.

39. **How does MiL compare to SAM (sharpness-aware minimization)?**
SAM tries to smooth the loss landscape of a global model but does not change the data or fuse label-aware structural knowledge. Experiments show MiL typically yields higher D-index and accuracy, while SAM remains sensitive to hyperparameters and can still overfit in small, noisy regimes.

40. **Did you compare MiL with pretraining + fine-tuning pipelines?**
Yes. We evaluated pipelines using autoencoders, DDPMs, SMOTE, and LASSO-based feature selection

as preprocessing, followed by fine-tuning a DNN. On LH-Ps like Ovarian and SEC 8-K, these pipelines often collapsed to trivial majority-class predictors or underperformed MiL by a large margin.

41. **Is LHI robust to the choice of embedding (t-SNE vs UMAP vs raw)?**
Yes. We compared LHIs computed on raw features, t-SNE, and UMAP embeddings across benchmarks and found only small numerical differences with no statistically significant shifts. This suggests LHI captures intrinsic task difficulty rather than artifacts of a particular embedding method.

42. **Does class imbalance alone define an LH-P?**
No. Class imbalance contributes to difficulty but is not sufficient. Some imbalanced tasks have low LHI and are easy once reweighted, while some balanced tasks (e.g., IRMAS) have extremely high LHI due to manifold entanglement. LHI measures intrinsic geometry-label misalignment, not just label counts.

43. **When should I *not* use MiL?**
MiL is not necessary when LHI is low ($< 0.75$) and standard models already achieve strong, stable performance. It is also less suitable for extremely large datasets where the $O(Mn^2p)$ offline cost of PTC cannot be amortized or approximated.

44. **What are MiL's main limitations?**
The main limitations are (1) the offline complexity of PTC, which is quadratic in $n$ and linear in feature dimension $p$, and (2) degraded effectiveness on very high-dimensional, noisy vectorized text, where constructing meaningful traininglets is harder even after de-noising.

45. **How does MiL perform on the SEC 8-K text dataset?**
On a 17k-sample BERT-vectorized SEC 8-K dataset with LHI $\approx 0.997$, MiL only modestly improves over an RBF-SVM baseline, while incurring much higher computational cost. Both methods are dominated by the majority "hold" class, demonstrating a genuine failure mode of MiL on extremely noisy vectorized text.

46. **Can MiL handle regression tasks?**
Yes. The framework can be extended to Learning-Hard Regression (LH-R), where the target is continuous and difficulty stems from local non-stationarity. In this setting, MiL constructs low-variance regression traininglets and fits local regressors (e.g., kernel ridge or SVR) to exploit local smoothness.

47. **Does MiL help with adversarial robustness?**
MiL's query-specific traininglet construction makes it harder to craft universal adversarial perturbations targeting a fixed global model. An adversary would need to simultaneously manipulate traininglet selection and the local SVM decision, which is combinatorially more complex than attacking a single differentiable global network.

48. **How scalable is MiL to large datasets?**
Exact PTC has $\mathcal{O}(Mn^2p)$ offline complexity, making it most practical for small-to-mid sized datasets ($n$ up to roughly $10^4$–$5 \times 10^4$ in our experiments). For larger datasets, we rely on NTC, SC-lets, and potential approximate nearest-neighbor search as future work, at the cost of some determinism.

49. **Can approximate nearest-neighbor methods (e.g., FAISS, Annoy) be used?**
Yes, they are a natural way to accelerate MiL for large $n$. In this initial work we did not use them to preserve strict reproducibility, but integrating them is a clear future direction, where trade-offs between scalability and determinism will need to be quantified.

50. **How are baselines tuned to ensure fair comparison?**
All classical and deep baselines are tuned via nested grid search within the same cross-validation folds used for MiL. Hyperparameter ranges and architectural details are documented in the supplemental

material, including regularization, learning rates, batch sizes, and augmentation choices where applicable.

51. **How does MiL compare to large pretrained or foundation models?**
Foundation models excel in general-purpose domains with abundant pretraining data. Our LH-Ps lie in specialized, small, noisy, and often imbalanced scientific domains where such models tend to overfit or transfer poorly. MiL explicitly targets this regime with deterministic, local learning and interpretable decision rules.

52. **Is MiL code and data available?**
Yes. An anonymized implementation and all benchmark datasets used in the experiments are available at the link provided in the paper (for the anonymous review phase), with plans for a fully documented open-source release after the review process.

53. **What is the main takeaway for practitioners?**
When LHI indicates a task is learning-hard and standard models underperform or behave unreliably, MiL offers an alternative: build small, query-specific traininglets, fit simple deterministic learninglets, and trade global complexity for a sequence of locally trivial problems with strong generalization and interpretability guarantees.