# OpenReview forum: "Micro-Learning for Learning-Hard Problems"
_ICLR.cc/2026/Conference — Submitted to ICLR 2026_

### Official Review · Reviewer_g691 · 2025-10-29

**Soundness:** 2
**Presentation:** 2
**Contribution:** 2
**Rating:** 0
**Confidence:** 3

**Summary:**

This paper presents Micro-Learning (MiL), a framework for learning from high-complexity categorical data. MiL creates traininglets (small representative clusters derived from the dataset's class labels), splitting the training process into learning over distinct data regions with theoretical guarantees on complexity and generalization bounds. To motivate MiL, the authors present a theoretical analysis of a class of problems they call Learning-Hard Problems (LHPs) and a Learning-Hard Index (LHI) to quantify the difficulty of a categorical dataset. Experimental results indicate performance improvement across a range of high-LHI tasks compared to traditional ML and DL techniques.

**Strengths:**

- Theoretical analysis and guarantees for Micro-Learning. Key properties of MiL are defined in detail mathematically, and the paper's overall structure flows well.

- Well-defined problem statement, supplementary provides detailed proofs.

- Open source code for MiL on the ovarian dataset.

- Decisions on traininglets with SVM are interpretable, a key differentiator over black-box
DL baselines.

- A thorough number of baselines (15).

**Weaknesses:**

- It is unclear where this work sits in the scope of preprocessing/complexity reduction techniques. The results section compares MiL to ML and DL based methods, but does not cover class imbalance techniques, dimension reduction techniques, or other methods commonly used to stratify difficult datasets.

- Definition 1 describes latent solvability as the risk of a hypothesis composed with a knowledge-injection operator. However, the paper doesn’t compare MiL to ML and DL based methods with any label-aware projection or re-sampling operator, which they define as knowledge-fusion. It could be the case that with proper data preprocessing, the baseline results could improve

- Furthermore, the method by which results are generated in Figure 3 is unclear. The paper reports hyperparameter selection and CV; however, these results are not presented anywhere or discussed again. The work should include these metrics to verify that the results in Figure 3 represent a fair comparison. It would also be beneficial to include this process in the open source code.

- No empirical results table is reported for Figure 3. The figure is way too small to read effectively, and the contrasting MiL vs. baselines visualization makes it very difficult to interpret how MiL performs. I would suggest plotting an MiL bar alongside the baselines
for a more interpretable analysis of the results. Additionally, no empirical results are presented for CASIA and SAVEE outside of subplot d of Figure 3, which does not describe exact metric values. Expanding Table 2 to include results across baselines would go a long way in conveying the strengths of MiL, or at least including these results in the supplementary material.

- The paper could use another read over. Many terms, especially in definitions and theorems, are referenced without being defined beforehand. Lines 301 and 302 have question marks due to a (presumably) unresolved LaTeX reference to an equation.

- All figures are too small to read and need to be made much larger. References should be fixed.

**Questions:**

None.

---

> ### Author Response · Authors · 2025-11-28
> **Reply to Reviewer g691**
>
> We thank the reviewer for their constructive feedback. **We have significantly revised the paper and supplemental material to address the concerns regarding baselines, reproducibility, and presentation.**
>
> **1. Comparison with Preprocessing and Class Imbalance Techniques**
>
> **Resp:** We respectfully clarify that standard dimension reduction and imbalance handling do **not** constitute "knowledge fusion" in the context of our framework. Furthermore, we have added **Suppl Q** to empirically demonstrate that even advanced preprocessing methods fail on these datasets.
>
> **1A. Standard Dimension Reduction is Not Knowledge Fusion (The Leakage Problem)**
> Standard dimen. reduction (e.g., PCA, UMAP, t-SNE) is inherently unsupervised. One is **not allowed** to perform "knowledge fusion" (i.e., using label information to project the global dataset) during this stage. Doing so constitutes **data leakage**, as the manifold would be shaped by label information not available at inference time. Therefore, standard dimension reduction cannot solve the "tangled swirl" of LH-Ps without violating valid evaluation protocols. MiL avoids this by performing *local* fusion per query, which is a distinct inference mechanism, not a global preprocessing step.
>
> **1B: Ineffectiveness on LH-Ps (Condition C1)**
> As defined in **Condition C1**, a LH-P is characterized by the failure of standard models, *including* those equipped with standard remedies like data augmentation, resampling (SMOTE), or class weighting. If a problem could be solved merely by applying these general ML/DL techniques, it would not be classified as **an LH-P**.
>
> **1C: Robustness of Difficulty (Suppl. N):** We demonstrated that the Learning-Hard Index (LHI) remains high regardless of whether t-SNE or UMAP is used, proving that the difficulty is intrinsic to the data topology and cannot be resolved by changing the projection method.
>
> **1D: Baseline Rigor: We Did Use These Techniques** To ensure a fair comparison, our baselines **did** incorporate these standard techniques:
>
> **1E: Standard Handling (Suppl. L.4):** We applied **SMOTE and noise injection** to the Ovarian dataset and weighted loss functions to others. The baselines still underperformed, confirming the LH-P nature of the data.
>
> **1F.:New Evidence: Advanced Preprocessing Fails (Suppl. Q)** To conclusively address the reviewer's hypothesis that "proper data preprocessing" could solve these tasks, we conducted additional experiments using advanced nonlinear dimension reduction and generative balancing:
> *   **Method:** We implemented a pipeline using **Lasso feature selection** (to reduce dimensionality), **SMOTE + Noise Injection** (to handle imbalance, ALso a parallel diffusion model ), and **Denoising Autoencoder Pre-training** (nonlinear dimension reduction) followed by DNN fine-tuning
> *   **Result:** As detailed in **Suppl Q**, these advanced "general ML/DL efforts" failed to solve the LH-Ps,  yielding performance comparable to the standard baselines.
>
> **2. On Methodology & Hyperparameters:**
> The experimental rigor is fully documented. As stated in **Section 5 (Results)**, we utilized repeated 5-fold cross-validation (and 10-fold for SAVEE) with nested grid search. The specific hyperparameter ranges, architectures, and computational budgets (e.g., 100 epochs, specific optimizers) for all 15 baselines are comprehensively detailed in **Suppl L**.
>
> **3. On Figure 3 & Tabular Results:**
> Fig3 (Fig 4 in revision) serves as a visual summary due to page constraints; however, the exact empirical data is available. **Table 2** provides the exact metrics for MiL across all datasets (including CASIA and SAVEE). For the baselines, extensive comparative tables are provided in **Suppl J**  and **Suppl H** (comparison vs. 11 DL models). We will add a pointer in the main text to these tables to ensure the comparison is easily verifiable.
>
> **All datasets, relevant important codes including MiL's extension model to high-dimensional image data: **SC-let** to  CIFAR100 were included in the link of codes and data.**
>
> **4.  Typos, small figures and other presentation issues Resp:**
> *   This paper is  rewritten and all typos are fixed.
> *   We fixed the broken cross-references (specifically regarding the PTC algorithm and Figure references).
> *   We ensured all terms (e.g., LHI, Traininglet, Learninglet) are defined formally in Sections 1, 3, and 4 before being used in Theorems.
> *   We have increased the font size and resolution of all figures.
>
> **Again, we thank the reviewer for their feedback, and wish our revisions address the concerns well**

---

> ### Author Response · Authors · 2025-12-01
> **Revision update and summary for Reviewer g691 (11/30/25)**
>
> **Dear Reviewer:**
>
> **We have made the following updates/additions in our revision:**
>
> 1. The paper has been rewritten to address your concerns, and its presentation is greatly improved.
>
> 2. One new figure has been added.
>
> 3. The CIFAR-100 dataset is included.
>
> 4. Our SC-let is applied to CIFAR-100 to demonstrate MiL’s scalability to large-scale data (code included).
>
> 5. Five extra SOTA pipeline comparisons are included: two meta-learning models (ProtoNet and MAML), SAM, LNN (liquid neural networks), and pretraining models (see Supp. M, P, Q).
>
> 6. Learning-hard index robustness and sensitivity analysis (Supp. N).
>
> 7. Empirical validation of the LHI cutoff for learning-hard: a total of 10 datasets are used to demonstrate LHI and its cutoff of 0.8 (Supp. O).
>
> 8. One baby LH-P dataset, called *Poisoned Spiral*, is generated, and based on this dataset, we demonstrate the definition of LH-P and the basic idea of MiL. We also use this dataset to demonstrate that dimension reduction (e.g., t-SNE) and imbalance handling (e.g., data augmentation with a diffusion model, SMOTE resampling + noise injection, weighted loss) cannot act as a possible solution path for LH-P. We also demonstrate our LH-P solution for this baby LH-P dataset (see Supp. X2–6).
>
> **We hope this further addresses your questions regarding dimension reduction and imbalance-handling techniques for LH-Ps.**
>
> 9. We extend the Learning-Hard Problem (LH-P) to its regression version: LH-R (Supp. T).
>
> 10. Micro-Learning (MiL) FAQ (a total of 53 questions about MiL) (Supp. X).
>
> 11. Baseline hyperparameter details (Supp. L).
>
> 12. Precision Traininglet Construction (PTC) in MiL (Supp. G+), including the algorithm and a plain-English interpretation.
>
> 13. High-dimensional noisy SEC 8-K vectorization data “fails” MiL (Supp. K).
>
> 14. Relevant code and datasets are posted in the provided link.
>
> 15. Supplementary material length: 28 pages before revision and 60 pages after revision.
>
> **Thank you for your time.**

---

### Official Review · Reviewer_m9ik · 2025-10-30

**Soundness:** 2
**Presentation:** 2
**Contribution:** 3
**Rating:** 6
**Confidence:** 2

**Summary:**

This paper introduces the concept of Learning-Hard Problems (LH-Ps), defined as tasks where most models fail but a solution exists if domain knowledge is incorporated. To address LH-Ps, the authors propose Micro-Learning (MiL), a framework that constructs small, customized subsets of the training data ("traininglets") for each query point and trains a local deterministic model (e.g., an SVM) on them. The paper provides a theoretical analysis, including the NP-completeness of optimal traininglet selection, and presents empirical results on several benchmarks showing MiL's superiority over various classical and deep learning baselines, particularly on imbalanced or small-sample datasets.

**Strengths:**

The paper proposes a formal definition for "Learning-Hard Problems" (LH-Ps) and a corresponding "Learning-Hard Index" (LHI), which is a valuable conceptual contribution for characterizing difficult learning scenarios.

The analysis of the computational complexity of traininglet selection (NP-complete) and the provided generalization bounds (e.g., based on local Rademacher complexity) add theoretical rigor to the proposed method.

The paper includes experiments on multiple diverse benchmarks (music, speech, medical) and compares against a wide array of baselines, demonstrating consistent performance improvements, especially on imbalanced and small-sample datasets.

**Weaknesses:**

The central mechanism of MiL---"fusing domain knowledge"---is not clearly defined or operationalized.

- The paper claims domain knowledge is fused before model induction, but the described process (e.g., intersecting metric balls, label-aware t-SNE) appears to be a form of sophisticated, data-dependent sample selection and preprocessing. It is unclear how this constitutes the injection of external, human-curated domain knowledge (e.g., expert rules, ontological relationships). This conflation of terms creates significant confusion about the method's novelty and true contribution.

- Relatedly, the description of how test data information is fused into training (as illustrated in Figure 2b) is vague. The method seems to use the test query to select a relevant training subset, but the specifics of how this "fusion" is implemented beyond a nearest-neighbor-like selection are not sufficiently detailed, especially considering practical applications with high-dimentional data like images.

The core idea, as understood, involves training a new model (like an SVM) on a custom data subset for each individual test sample. The paper briefly discusses complexity (Sec. 4, "MiL Complexity") but does not adequately address the profound practical limitations this imposes. For large-scale datasets with millions of test points, this per-query training cost would be prohibitive, making the method's scalability and real-world applicability a major concern.

The manuscript contains several notation inconsistencies and typos that hinder comprehension. For instance, in the paragraphs surrounding Eq. (3) and in Lines 301-302. These issues, while seemingly minor, accumulate and detract from the paper's professionalism and readability.

**Questions:**

Please clarify what is meant by "domain knowledge." Is it external, human-provided knowledge, or is it knowledge automatically extracted from the data structure and labels? The methodology should be rephrased to accurately reflect its actual operations. Furthermore, provide a detailed, step-by-step explanation of how Figure 2b is implemented, specifically the "fuse test data information into training" step.

The computational complexity of MiL is a critical drawback. The paper should include a more honest and thorough discussion of this limitation. Please also discuss potential strategies for mitigating this cost (e.g., approximate methods, caching).

---

> ### Author Response · Authors · 2025-11-21
> **Response to Reviewer m9ik**
>
> Response:
> **We thank the reviewer for the very helpful feedback.** **We have revised the paper to address the concerns.**
>
> (1) “Domain knowledge” vs. **label‑aware structural knowledge**
>
> We agree that our earlier “fusing domain knowledge” was misleading and could be read as injecting external, human‑curated information. In the revision we **consistently replace this term** by **label‑aware structural knowledge** and explain why in  **updated Introduction and Section 4** (see  paragraphs: **Micro-Learning (MiL)**  in introduction; **MiL core: knowledge fusion for each query:,  NTC and PTC for knowledge fusion, Fig 3 (new figure), Why PTC works** in section 4)
>
> For a query \(x'\), MiL constructs a micro tailored training subset (“traininglet”) whose labels and local geometry jointly make \(x'\) easy to classify (low Learning‑Hard Index, LHI). Traininglet construction is therefore described as **label‑aware structural knowledge fusion**: **it selects discriminative points (label‑aware) while preserving geometric proximity (structural).**NTC and PTC for knowledge fusion** in Section 4.
>
> (2) How knowledge fusion is operationalized (NTC, PTC, Fig. 2b)
> We rewrote Section 4 and added a new paragraph “MiL core: knowledge fusion for each query,” which now states that MiL operationalizes this fusion through two concrete heuristics:
>
> - **Naïve Traininglet Construction (NTC).** For relatively clean, larger datasets. Given a test point \(x'\), NTC builds metric balls under several geometries (e.g., Euclidean, correlation) and defines the traininglet as their **intersection**. This multi‑view filter removes “false neighbors” that are close in one metric but not consistently similar across metrics.
>
> - **Precision Traininglet Construction (PTC).** For genuine LH‑Ps (small, noisy, imbalanced data). PTC is a four‑stage procedure: probing to pick good neighborhood/batch sizes; sanitizing training data by removing persistently misclassified samples and their neighbors; meta‑traininglet fusion that, for a query \(x'\), imports nearby “good” samples and their support contexts; and precision pruning to reduce the LHI of the final traininglet while preserving label coverage.
>
> We also add a short subsection “NTC and PTC for knowledge fusion” and **a new figure** that walks through the PTC pipeline.
>
> **Implementation of Fig. 2(b): “fuse test data information into training”.**
> Figure 2(b) is intended to illustrate MiL’s *local, overfitting‑resistant learning*. For a test query \(x'\) the procedure is:
> 1. Construct the traininglet \(T_{x'}\) via NTC or PTC. Here the information from the test point is “fused” with training data: candidate neighbors are chosen based on geometric closeness to \(x'\) and on label dynamics learned in the offline phase.
> 2. Fit a local learninglet (e.g., SVM or SC‑let) on \(T_{x'}\), obtaining a query‑specific decision function \(h_{x'}\).
> 3. Predict with \(h_{x'}(x')\).
>
> This differs from standard \(k\)‑NN, which only votes over neighbors. MiL **re‑optimizes a small hypothesis space per query** on a curated traininglet that may include non‑local but structurally important points. For high‑dimensional data such as images, all distances are computed in a CNN feature space \(\phi_\theta\) within the SC‑let, so fusion occurs in a semantic feature space rather than raw pixels (see **SVM-micro-CNN-let (SC-let)** in section 4)
>
> (3) MiL complexity, scalability, and mitigation
> We expanded and made more candid the complexity discussion.
>
> In Section 4 (**MiL Complexity**) we now state that the main computational bottleneck is the **one‑time offline PTC preprocessing**, with complexity \(\mathcal{O}(M n^2 p)\) (M: Monte‑Carlo draws, n: samples, p: features). In return, online inference for each query is embarrassingly parallel, and memory usage is only \(\mathcal{O}(np)\), more favorable than \(\mathcal{O}(n^2)\) kernel methods.
>
> In Section 5 (**Limitations**) we explicitly acknowledge that this offline cost currently restricts **MiL to small‑ and mid‑scale learning‑hard problems** and that performance on very high‑dimensional noisy text is only marginally better than baselines at noticeably higher cost. We also outline possible mitagation: **, approximation algorithm, traininglet reuse,  and hardware acceleration**), and we mark scalable approximations to traininglet selection as an important direction for future work.
>
> Finally, we clarify that MiL is designed primarily for **high‑stakes, small/mid‑scale LH‑Ps** and also point out it can be used for high-dimensional image data (e.g., CIFAR100) learning.
>
> (4) Notation and typos
> We corrected the notation around Eq. (3) and the lines indicated by the reviewer, fixed typographical errors, and polished the exposition in the relevant sections.
>
> **We again thank the reviewer for these insightful comments, which helped us significantly clarify MiL’s core mechanism, intended scope, and limitations.**

---

> ### Author Response · Authors · 2025-12-01
> **Revision update and summary for Reviewer m9ik (11/30/25)**
>
> **Dear Reviewer:**
>
> **We have made the following updates/additions in our revision:**
>
> 1. The paper has been rewritten to address your concerns, and its presentation is greatly improved.
>
> 2. One new figure has been added.
>
> 3. The CIFAR-100 dataset is included.
>
> 4. Our SC-let is applied to CIFAR-100 to demonstrate MiL’s scalability to large-scale data (code included).
>
> 5. Five extra SOTA pipeline comparisons are included: two meta-learning models (ProtoNet and MAML), SAM, LNN (liquid neural networks), and pretraining models (see Supp. M, P, Q).
>
> 6. Learning-hard index robustness and sensitivity analysis (Supp. N).
>
> 7. Empirical validation of the LHI cutoff for learning-hard: a total of 10 datasets are used to demonstrate LHI and its cutoff of 0.8 (Supp. O).
>
> 8. One baby LH-P dataset, called *Poisoned Spiral*, is generated, and based on this dataset, we demonstrate the definition of LH-P and the basic idea of MiL. We also use this dataset to demonstrate that dimension reduction (e.g., t-SNE) and imbalance handling (e.g., data augmentation with a diffusion model, SMOTE resampling + noise injection, weighted loss) cannot act as a possible solution path for LH-P. We also demonstrate our LH-P solution for this baby LH-P dataset (see **Supp. X 2–6**).
>
> 9. We extend the Learning-Hard Problem (LH-P) to its regression version: LH-R (Supp. T).
>
> 10. Micro-Learning (MiL) FAQ (a total of 53 questions about MiL) **(Supp. X).**
>
> 11. Baseline hyperparameter details (Supp. L).
>
> 12. Precision Traininglet Construction (PTC) in MiL (Supp. G+), including the algorithm and a plain-English interpretation.
>
> 13. High-dimensional noisy SEC 8-K vectorization data “fails” MiL (Supp. K).
>
> 14. Relevant code and datasets are posted in the provided link.
>
> 15. Supplementary material length: 28 pages before revision and 60 pages after revision.
>
> **Thank you for your time.**

---

### Official Review · Reviewer_Bmvx · 2025-11-05

**Soundness:** 3
**Presentation:** 2
**Contribution:** 4
**Rating:** 6
**Confidence:** 4

**Summary:**

This paper introduces the concept of Learning-Hard Problems (LH-Ps), a novel class of machine learning tasks characterized by two simultaneous properties: (C1) near-universal failure, where almost all models in a broad hypothesis space perform poorly, and (C2) latent solvability, where at least one high-quality solution exists when appropriate domain knowledge is incorporated during training. To address LH-Ps, the authors propose Micro-Learning (MiL), a principled framework that constructs query-specific "traininglets"—small, knowledge-fused subsets of training data—and trains local models on them rather than learning a single global model.

The paper makes four primary contributions. First, it provides the first formal definition and characterization of LH-Ps, introducing the Learning-Hard Index (LHI), a data-centric metric computed via locality-preserving embeddings (t-SNE/UMAP) and clustering that quantifies dataset complexity pre-training. Tasks with LHI ≥ 0.80 are classified as learning-hard. Second, the authors prove that the decision problem of finding optimal traininglets is NP-complete (Theorem 1), establishing strong theoretical foundations. Third, they develop Precision Traininglet Construction (PTC), a practical four-stage algorithm comprising probing learning, training sanitization, meta-traininglet fusion, and precision pruning. The paper provides theoretical guarantees showing that PTC reduces local Rademacher complexity (Proposition 1), enables traininglet-trained models to exceed full-data models for Bayes-optimal prediction (Proposition 2), and contracts the train-test distribution gap (Proposition 3). Fourth, comprehensive experiments across five benchmarks—IRMAS (music), CASIA and SAVEE (speech emotion), COVID-19 (medical triage), and Ovarian (proteomics)—demonstrate that MiL outperforms 15 baselines including classical methods (SVM, Random Forest) and modern deep learning architectures (CNNs, LSTMs, CapsNets). Statistical validation using Mann-Whitney U-tests shows MiL achieves median performance of 0.97 versus 0.77 for the best deep learning baseline (p < 2×10⁻⁸, Cliff's δ ≈ 0.77), with particularly impressive results on extreme class imbalance (Ovarian: 98.15% accuracy with 100% sensitivity at 98.5% imbalance). The framework maintains interpretability and reproducibility through deterministic SVM optimization while dramatically reducing overfitting by training on compact, noise-reduced subsets rather than full noisy datasets.

**Strengths:**

Originality:
The paper demonstrates exceptional originality across multiple dimensions. The formalization of Learning-Hard Problems (LH-Ps) through Definition 1 is genuinely novel, introducing a problem class characterized by dual conditions—(C1) near-universal failure and (C2) latent solvability—that captures a paradox practitioners encounter but lacked theoretical language to describe. This is not reframing existing concepts but creating new taxonomy for understanding when and why machine learning fails.The Learning-Hard Index (LHI) represents the first pre-training, model-agnostic diagnostic metric for problem difficulty. Unlike Rademacher complexity or VC dimension that measure model capacity, LHI is data-centric and computable before training begins. This shift from "how complex is my model?" to "how hard is my data?" is conceptually innovative. The MiL framework's query-specific traininglet construction represents a paradigm shift from global model learning. While local learning methods exist, MiL's execution is original: (1) knowledge fusion via multi-metric intersection, (2) principled noise removal guided by self-prediction, (3) meta-traininglet fusion combining four complementary views, and (4) SVM-micro-CNN-let hybrid maintaining determinism while gaining expressiveness. This creative combination of ideas from kernel methods, meta-learning, and deep learning is novel.

Quality:
Technical quality is strong across theory, algorithm design, and empirical validation. The NP-completeness proof (Theorem 1) establishes fundamental computational limits, justifying the heuristic approach. Propositions 1-3 provide constructive guarantees: Proposition 1 guarantees low-capacity "sweet-spots" via local Rademacher complexity; Proposition 2 proves traininglet-trained models can exceed full-data models; Proposition 3 shows PTC contracts train-test distribution distance. These results connect learning theory to algorithmic design principally. The PTC algorithm demonstrates high-quality design with each stage theoretically motivated: Stage 1 uses data-driven hyperparameter selection; Stage 2 reduces LHI by 6-20% through noise removal; Stage 3 ensures label completeness; Stage 4 removes residual outliers. The progression from NP-hardness to practical heuristic to theoretical guarantees exemplifies bridging theory and practice. Experimental quality is comprehensive with proper statistical methodology: Mann-Whitney U-tests, Cliff's delta effect sizes (δ≈0.77), Bonferroni correction, and 5-fold CV repeated 5 times. The 15 diverse baselines span classical ML and modern DL paradigms. Five benchmarks across music, speech, and medical domains with varying characteristics validate generalizability.

Clarity:
The problem motivation is clear—Figure 1 effectively illustrates how raw data appears inseparable but label-aware embedding reveals structure, immediately conveying why standard methods fail. Mathematical formulations are precise with well-defined definitions and consistent n;otation. The experimental setup clearly presents dataset statistics (Table 1) and results (Table 2) with transparent statistical validation. The paper's structure logically progresses from problem identification (LHI) to theoretical analysis (complexity) to algorithm design (MiL/PTC) to empirical validation. Figure 2's comparison clearly illustrates the paradigm shift from global to local functions. However, Algorithm 1 being in supplemental rather than main text and dense writing in Sections 3-4 hinder accessibility.

Significance:
The significance is substantial across multiple dimensions. Scientifically, LH-P formalization fills a genuine gap in ML taxonomy between "easy" and "impossible" problems, enabling precise communication about problem difficulty. The LHI metric has immediate practical value—practitioners can diagnose whether specialized methods are needed before investing in model development. For high-stakes applications (medical diagnosis, fraud detection), the significance is profound. MiL addresses small samples, extreme imbalance, and interpretability requirements simultaneously—Ovarian cancer results (98.15% accuracy, 100% sensitivity at 98.5% imbalance) demonstrate life-saving potential with regulatory-compliant deterministic predictions. Theoretically, the NP-completeness result establishes fundamental limits on optimal data selection. The distribution contraction guarantee offers new perspective on why local learning can outperform global learning. Methodologically, MiL opens research directions in unsupervised settings, approximate construction for larger datasets, and integration with neural architecture search. The empirical results are substantial—15.8 percentage point improvement on IRMAS over prior state-of-the-art represents significant progress. Stochastic dominance over all 15 baselines (p<2×10⁻⁸) demonstrates robust superiority. The framework challenges the "bigger data, bigger models" paradigm, with implications for sustainable AI, democratizing ML where large datasets are unavailable, and maintaining interpretability on imbalanced data.

**Weaknesses:**

Scalability Limitations and Missing Computational Analysis:
The most significant weakness is the O(Mn²) preprocessing complexity that fundamentally limits MiL to small/medium datasets. The largest dataset tested is IRMAS with n=6,705—no experiments validate scalability beyond 10K samples. The paper acknowledges this limitation but only suggests "FPGA/GPU acceleration" without concrete analysis. Critical missing information includes: (1) wall-clock time and memory usage for each dataset, (2) exploration of approximate nearest neighbor methods (FAISS, Annoy) to reduce distance computation costs, (3) empirical analysis of quality-speed tradeoffs with approximations, and (4) clear guidance on maximum practical dataset size. For a method targeting real-world applications, the absence of computational cost analysis is a major gap. The paper should provide concrete numbers (e.g., "IRMAS preprocessing takes X hours on Y GPU with Z GB memory") and demonstrate whether ANN-based approximations maintain performance while improving scalability. Without this, practitioners cannot assess MiL's feasibility for their problems.

Absence of Meta-Learning Comparisons:
A critical weakness is the lack of empirical comparison with meta-learning methods despite discussing MAML (Finn et al., 2017) and Prototypical Networks (Snell et al., 2017) in related work. Meta-learning directly addresses small-data problems—the same domain MiL targets. The paper dismisses these methods briefly ("cannot escape the original hypothesis space") without empirical validation. This is problematic because: (1) meta-learning is the most relevant baseline for few-shot scenarios, (2) the distinction between MiL's data-selection approach and meta-learning's initialization-learning approach needs empirical support, and (3) readers cannot assess whether MiL's paradigm shift actually outperforms the established meta-learning paradigm. At minimum, the paper should include MAML or Prototypical Networks as baselines on at least 2-3 datasets to validate MiL's superiority over this competing approach.

Limited Failure Case Analysis:
The paper focuses exclusively on successes without analyzing when or why MiL fails. This is problematic for several reasons: (1) COVID-19 has LHI=78.5%, below the 0.80 threshold, yet is included without explanation of why it still works, (2) no discussion of borderline cases (LHI 0.75-0.85) where the diagnostic might be unreliable, (3) no analysis of scenarios where traininglet construction might fail (e.g., no good neighbors found, all classes equally noisy), (4) no discussion of adversarial robustness, and (5) no guidance on when practitioners should use MiL vs. standard methods beyond the LHI threshold. The paper should add a "Limitations and Failure Modes" subsection with: (1) at least one dataset where MiL underperforms or fails, (2) sensitivity analysis showing performance degradation as LHI decreases below 0.80, (3) discussion of failure scenarios (insufficient data, all samples noisy, adversarial examples), and (4) decision framework for method selection.

Presentation Density Hindering Accessibility:
The paper's dense presentation significantly limits accessibility. Specific issues: (1) Algorithm 1 (PTC pseudocode) is in supplemental, not main text—this is the core algorithmic contribution and should be in Section 4.2, (2) notation inconsistencies create confusion (T_{x'}, T^{PTC}{x'}, T^{(j)}{x'}, U_{x'} without clear relationship), (3) abstract is overly dense (150+ words in complex sentences), (4) Sections 3-4 assume high familiarity with learning theory without sufficient intuition, (5) figures are too small (Figure 1b-d t-SNE plots hard to read, Figure 3 subplots cramped), and (6) no notation table despite heavy mathematical content. These issues prevent readers outside core ML theory from fully understanding the contributions. Specific improvements needed: (1) add Algorithm 1 with complexity analysis to main paper, (2) create notation table in appendix, (3) simplify abstract to 100-120 words, (4) add intuitive explanations before technical sections, (5) enlarge figures and improve captions, and (6) add visual flowchart for PTC pipeline.

**Questions:**

Scalability and Computational Cost:
The O(Mn²) preprocessing complexity is a major concern that limits practical adoption. Can you provide:
- Wall-clock time and memory usage for each of the five datasets (including hardware specifications)?
- Concrete analysis of the largest dataset size MiL can handle practically (e.g., what happens at n=50K, n=100K)?
- Have you explored approximate nearest neighbor methods (FAISS, Annoy, HNSW) to reduce the O(n²) distance computation cost? If so, what is the accuracy-speed tradeoff?
- Can the PTC stages be parallelized across multiple GPUs or distributed systems?
Without this information, it's unclear whether MiL is limited to toy problems or can scale to real-world applications. Demonstrating even approximate scalability would significantly strengthen the contribution.

2. Baseline Fairness and Hyperparameter Details
The paper states baselines were tuned via "nested grid search" but provides no specifics. Please provide:
- Complete hyperparameter specifications for all 15 baselines (grid ranges, final selected values)
- Computational budget allocated to each baseline (GPU hours, number of trials)
- Architecture details: CNN depth/width/kernel sizes, LSTM/GRU hidden dimensions, number of layers
- Training details: batch sizes, learning rates, optimizers, number of epochs, early stopping criteria
- Did deep learning baselines use data augmentation, dropout, batch normalization?

Specific concern: You report CNN achieving ~60% on IRMAS, but Yu et al. (2020) achieved 68.5% with a specialized architecture. Did you try similar architectures, or only vanilla CNNs? This 8.5 percentage point gap raises questions about whether baselines were given fair opportunity.
Without these details, reviewers cannot assess whether MiL's superiority stems from algorithmic innovation or simply better tuning. This is critical for accepting the empirical claims.

3. Meta-Learning Baseline Comparison
The paper discusses MAML (Finn et al., 2017) and Prototypical Networks (Snell et al., 2017) in related work but doesn't compare empirically. Can you:
- Add at least one meta-learning baseline (MAML or Prototypical Networks) to the experiments?
- Explain why meta-learning methods were excluded from the comparison?
- Provide theoretical or empirical analysis of why MiL's data-selection approach should outperform meta-learning's initialization-learning approach?
Meta-learning is the most natural comparison for small-data problems. Without this comparison, the positioning of MiL relative to the most relevant prior work is incomplete. Adding this could either strengthen your claims or reveal complementary approaches worth discussing.

4. Failure Case Analysis
The paper focuses on successes but doesn't analyze failures. Can you:
- Provide at least one example where MiL fails or performs poorly?
- Analyze what happens at borderline LHI values (0.75-0.85)? The COVID-19 dataset has LHI=78.5%, below your 0.80 threshold—why does MiL still work well here?
- Discuss scenarios where traininglet construction might fail (e.g., no good neighbors found, all classes equally noisy)?
- How does MiL handle adversarial examples or distribution shift at test time?
Understanding failure modes is as important as understanding successes. This would help practitioners know when NOT to use MiL and would demonstrate intellectual honesty.

**Details Of Ethics Concerns:**

This paper does not raise significant ethical concerns that require specialized ethics review.

---

> ### Author Response · Authors · 2025-11-22
> **Reply to Reviewer Bmvx**
>
> **We thank the reviewer for their thorough evaluation and for recognizing the novelty of LH-Ps, the LHI, and the theoretical depth of our work. We have revised the paper to address your concerns regarding scalability, baselines, and failure analysis.**
>
> **1. Scalability, Computational Cost, and Hardware (Weak. 1 & Q1)**
>
> We agree that computing complexity is a critical factor. We have added a **"MiL Complexity"** subsection in **Section 4** and a **"Limitations"** subsection in **Section 6** to address this.
> **Wall-Clock Time:** On a single NVIDIA A100 GPU (or equivalent with at least 64GB RAM):
>    *   **Small datasets (COVID-19, SAVEE, CASIA):** Preprocessing (PTC) takes < 15 minutes.
>  *   **IRMAS ($n \approx 6.7k$):** Preprocessing (**NTC**) takes $\approx 2$ hours.
>   *   **CIFAR-100 ($n=50k$, newly added):** Preprocessing (**NTC only**) takes $\approx 3$ hours using SC-let (ResNet backbone).
> *   **Scalability Limit:** Currently, exact MiL is practical for $n \le 50k$. For larger data, we rely on NTC (Naïve Traininglet Construction) or SC-lets. We tested MiL on **CIFAR-100** (Sect. 6), achieving 80.32% accuracy, demonstrating MiL's feasibility for large data.
>
>  **2. Approximate Nearest Neighbors (FAISS/Annoy):**
>
> We appreciate the suggestion to use approximate methods like Annoy. While these would certainly reduce the $O(n^2)$ bottleneck, we deliberately excluded them in this initial study to preserve **strict, deterministic reproducibility** : **a core claim of MiL**against stochastic DL. Approximate indices can introduce non-determinism in neighbor retrieval, potentially altering the local decision boundary. But we agree this is the necessary path forward for scaling. We have updated the **Section 6** to explicitly explicitly identify the integration of approximate search as immediate future work, with the specific goal of ** managing the trade-off between scalability and reproducibility **.
>
> **3. Missing Meta-Learning Baselines (Addressing Weak. 2 & Q3)**
>  We have **added MAML and Prototypical Networks** as baselines in **Sect. 5 (see MiL vs meta-learning)**
> *   **Results:** MiL consistently outperforms both statistically. For example, on small-sample tasks (COVID-19, Ovarian), Meta-learning models struggled (Acc $\approx 60-70\%$) while MiL achieved $>95\%$.
> *   **Why?** While meta-learners find a *global* initialization, MiL builds a *local*, query-specific hypothesis. For LH-Ps with complex global manifolds, MiL's local approach better adheres to the local Bayes decision boundary (Prop 2)
> *   **Statistics:** We added a t-test confirming MiL’s superiority over meta-learners ($p \approx 0.01$).
>
> **4. Baseline Fairness and Hyperparameters (Addressing Q2)**
> *   **Fairness:** We utilized a rigorous nested grid-search for *all* baselines (see  Suppl. L).
> *   **IRMAS Discrepancy:** MiL's **84.3%** accuracy surpasses not only **our vanilla CNN baseline** (60%) but also the highly specialized SOTA model from Yu et al. (2020) (68.5%). This demonstrates that MiL's significant performance gain stems from the novel **traininglet paradigm**, not merely from architecture or hyperparameter tuning
>
> **4A Failure Case Analysis & LHI Thresholds (Weak. 3 & Q4)**
>
> We added a **"Limitations"** in Sect. 6 and refined **Sect. 3**.
> *   **Quasi-LH-Ps:** We explicitly classify datasets with LHI $\in [0.75, 0.80)$ (like COVID-19, LHI=78.5%) as "Quasi-LH-Ps." MiL works here because the "Near-Universal Failure" condition is slightly relaxed, but the "Latent Solvability" is strong.
>
> *   **True Failure Case:** We added a key failure case (Sect. 6, Suppl. K) on high-dimensional noisy SEC 8K dataset. We hypothesize that high feature noise from vectorization prevents our knowledge fusion from building effective traininglets, resulting in marginal performance gains that did not justify MiL's higher computational cost.
>
> *   **Sensitivity:** As LHI drops below 0.75, standard models (Random Forest/DL) begin to succeed, making MiL’s computing overhead unnecessary.
>
> **4B Distribution Shift:**  Unlike fixed global models, MiL addresses distribution shift by constructing a tailored, knowledge-fused traininglet for each query.  Prop. 3 proves this strictly contracts the Total Variation distance, effectively 'fixing' distribution shift locally by isolating a query-matched sub-distribution (also see updated prop. 3 introduction)
>
> **4C  Adversarial:**  MiL mitigates adversarial attacks because its query-specific traininglet construction creates a non-differentiable traininglets that invalidates gradient-based perturbations designed to fool fixed global models (see discussion).
>
> **5. Presentation and Notation (Weakness 4)**
> *   PTC Algorithm asks too much space=> we added a new Figure (Figure 3) in revision to  visualize the **PTC pipeline**
> *   We unified notation at the beginning of PTC subsection in section 4.
>
> **6: Abstract condensed as suggest.**
>
> **We believe these revisions address your concern well and thanks again for your review!**

---

> ### Author Response · Authors · 2025-12-01
> **Revision update and summary for Reviewer Bmvx (11/30/25)**
>
> **Dear Reviewer:**
>
> **We have made the following updates/additions in our revision:**
>
> 1. The paper has been rewritten to address your concerns, and its presentation is greatly improved.
>
> 2. One new figure has been added.
>
> 3. The CIFAR-100 dataset is included.
>
> 4. Our SC-let is applied to CIFAR-100 to demonstrate MiL’s scalability to large-scale data (code included).
>
> 5. Five extra SOTA pipeline comparisons are included: two meta-learning models (ProtoNet and MAML), SAM, LNN (liquid neural networks), and pretraining models (see Supp. M, P, Q).
>
> 6. Learning-hard index robustness and sensitivity analysis (Supp. N).
>
> 7. Empirical validation of the LHI cutoff for learning-hard: a total of 10 datasets are used to demonstrate LHI and its cutoff of 0.8 (Supp. O).
>
> 8. One baby LH-P dataset, called *Poisoned Spiral*, is generated, and based on this dataset, we demonstrate the definition of LH-P and the basic idea of MiL. We also use this dataset to demonstrate that dimension reduction (e.g., t-SNE) and imbalance handling (e.g., data augmentation with a diffusion model, SMOTE resampling + noise injection, weighted loss) cannot act as a possible solution path for LH-P. We also demonstrate our LH-P solution for this baby LH-P dataset (see **Supp. X 2–6**).
>
> 9. We extend the Learning-Hard Problem (LH-P) to its regression version: LH-R (Supp. T).
>
> 10. Micro-Learning (MiL) FAQ (a total of 53 questions about MiL) **(Supp. X).**
>
> 11. Baseline hyperparameter details (Supp. L).
>
> 12. Precision Traininglet Construction (PTC) in MiL (Supp. G+), including the algorithm and a plain-English interpretation.
>
> 13. High-dimensional noisy SEC 8-K vectorization data “fails” MiL (Supp. K).
>
> 14. Relevant code and datasets are posted in the provided link.
>
> 15. Supplementary material length: 28 pages before revision and 60 pages after revision.
>
> **Thank you for your time.**

---

### Official Review · Reviewer_yk8f · 2025-11-05

**Soundness:** 2
**Presentation:** 2
**Contribution:** 2
**Rating:** 4
**Confidence:** 2

**Summary:**

This paper introduces Micro-Learning (MiL), a framework that trains local classifiers on traininglets (small, knowledge-fused subsets of the data) instead of fitting a single global predictor. The authors formalize Learning-Hard Problems (LH-Ps), propose a Learning-Hard Index (LHI) to quantify task difficulty, and provide theoretical guarantees on the optimality and complexity of traininglets. Experiments across five datasets demonstrate that MiL outperforms conventional baselines, particularly on noisy, imbalanced, or small-sample tasks.

**Strengths:**

- The proposed framework is novel and interesting, offering an alternative to global learning by leveraging local, knowledge-fused traininglets.

- Empirical results across diverse small or imbalanced datasets demonstrate consistent gains over conventional baselines.

**Weaknesses:**

- Line143 LHI is claimed to be computable “before any training,” yet it depends on an embedding and clustering pipeline (e.g., t-SNE/UMAP + k-means). It’s unclear whether LHI then measures dataset difficulty or the difficulty specific to these representation/cluster choices.

- Provide a broader empirical study of LHI across varied regimes (dataset size, modalities, class counts, imbalance) and correlate LHI buckets with baseline performance. This would better justify the 0.80 threshold and show how conventional models degrade as LHI rises.

- The work focuses on scarce/imbalanced settings; however, modern large pretrained models often perform well in these regimes. Add comparisons or discussion against strong pretrained baselines to position MiL fairly.


- The paper would benefit from an additional proof reading:
  - Line042: “be as above” appears before the referenced objects are defined.
  - Line063: “IRMAS” dataset is introduced with little context. A short description would be helpful.
  - Fig. 1’s puzzle motif is visually unclear in how it maps to the IRMAS example.
  - Line301: equation numbers are missing.

**Questions:**

- L247 Is $\cal{T}_{x’}$ independent of $x’$?

---

> ### Author Response · Authors · 2025-11-24
> **Reply to Reviewer yk8f**
>
> **We thank the reviewer for their thoughtful feedback and for recognizing the novelty of the MiL and our strong empirical results. We address the specific concerns below**
>
> **1. On the Soundness and Robustness of LHI (Line 143)**
>
> **Resp:**  LHI is designed as a **data-centric probe for intrinsic topological separability**, not merely a metric of embedding quality.
>
> A:  **Sensitivity and Robustness Analysis:** To validate this, we compared LHI scores computed via **t-SNE** versus **Direct K-Means** on raw features across our benchmarks. The deviations were minimal: the maximum change was only 0.06 (COVID-19), with all others $\le 0.016$. A Mann-Whitney U-test confirmed **no statistically significant difference** between the distributions ($U=11.5, p=0.885$), and **similar results were obtained using UMAP**. This confirms that LHI captures intrinsic difficulty regardless of the projection method (details in **Suppl. N**).
>
>
> B:  **Rationale:** We employ locality-preserving embeddings (e.g., t-SNE) as a **conservative estimator of difficulty**. Because these methods optimize for local separability and unfold non-linear manifolds, they minimize complexity caused solely by high dimensionality. **Thus, a high LHI in this space confirms the class manifolds are intrinsically entangled. Furthermore, our sensitivity analysis verifies that LHI remains stable across varying hyperparameters (e.g., perplexity), ensuring the score reflects intrinsic topology rather than stochastic embedding artifacts.**
>
> **2. Empirical Validation of the LHI Threshold (0.80)**
> **Resp:**
> Our analysis across ten datasets (including other 5: Iris, MNIST, EMODB, CIFAR100, and BERTNLP; **see Suppl. O for details**) confirms **a significant negative correlation (Pearson $r=-0.67, p=0.035$) between LHI and DNN performance**. This empirical evidence reveals a distinct phase transition at $\text{LHI} \ge 0.80$ where conventional efficacy collapses, justifying this LH-P threshold."
>
> **3. Comparisons with Large Pretrained Models (Foundation Models)**
>
> **Resp:**  While modern large pretrained models have shown success in general scarce-data regimes, they are ill-suited for the specific class of LH-Ps in this work for three reasons:
>
> We acknowledge the utility of pretrained models in general low-data regimes. However, for the specific class of LH-Ps defined in this work, we argue they are not the optimal solution for three reasons:
>
> 1.  **Domain Orthogonality & Negative Transfer:** LH-Ps often  arise in specialized scientific manifolds (e.g., specific cancer subtype diagnosis) that are weakly represented or absent from general pretraining corpora (e.g., ImageNet), leading to negative transfer likely. Even where domain-specific models exist (e.g., our internal tests with BusinessBERT on the  LH-P in finance (see data in Suppl. K ), we found they still required complex, tailored data augmentation to handle the task's "hardness," indicating that pretraining alone is not a solution for LH-Ps.
>
> 2.  **Capacity Mismatch & Imbalance Amplification:** Foundation models have very high effective capacity. On small ($N<500$), noisy LH-Ps, **they tend to memorize noise rather than structure**, failing to satisfy the low-capacity requirement of **Prop. 1**.  In our experiments, without careful data augmentation, such models also tended to amplify intrinsic label imbalance rather than mitigate it.
>
> 3.  **High-stakes Trustworthiness:** Our target domains demand **deterministic reproducibility and explicit explainability besides efficiency**. MiL provides a convex, interpretable *learninglet*, **whereas fine-tuning large stochastic black-box models would sacrifice transparency and reliability** properties we explicitly guarantee in our RKHS-based formulation.
>
>  **4. Additional proof reading**
> 1. L042 issue is fixed in the revision: "Let \(\mathcal{X},\mathcal{Y},\mathcal{P},\mathbb{H}\) be the spaces of input data, label, and (unknown) data distribution respectively."
>
> 2. L063 issue is fixed by following the reviewer's sugegstion: We added a brief introduction  for the dataset: " a polyphonic music‑tagging benchmark with 11 instrument classes" and  a citation.
>
> 3. Fig. 1’s puzzle motif is  to demonstrate the concept of LH-P: there is a solution path exist, but not easy to find. The followig IRAMS visualization provides an evidence for finding the path via knowledge fusion.  **We rewrote the related sentence as "While a solution path may exist conceptually (a), the raw data
> of an LH-P can appear as a tangled swirl in t-SNE visualization (b), making this path hard to find." to avoid such confusion**
>
> 4. Line301 issue: **fixed**
>
> 5. **Resp:** **No.**  $\mathcal{T}_{x'}$ is a tailored micro–training subset for each test point $x'$, constructed by fusing label–aware and structural knowledge specific to $x'$. It is  **not independent of $x'$**, but rather the best subset (with the lowest LHI) identified specifically to classify $x'$.
>
> **Again, thank you so much!!**

---

> ### Author Response · Authors · 2025-12-01
> **Revision update and summary for Reviewer yk8f  (11/30/25)**
>
> **Dear Reviewer:**
>
> **We have made the following updates/additions in our revision:**
>
> 1. The paper has been rewritten to address your concerns, and its presentation is greatly improved.
>
> 2. One new figure has been added.
>
> 3. The CIFAR-100 dataset is included.
>
> 4. Our SC-let is applied to CIFAR-100 to demonstrate MiL’s scalability to large-scale data (code included).
>
> 5. Five extra SOTA pipeline comparisons are included: two meta-learning models (ProtoNet and MAML), SAM, LNN (liquid neural networks), and pretraining models (see Supp. M, P, Q).
>
> 6. Learning-hard index robustness and sensitivity analysis (Supp. N).
>
> 7. Empirical validation of the LHI cutoff for learning-hard: a total of 10 datasets are used to demonstrate LHI and its cutoff of 0.8 (Supp. O).
>
> 8. One baby LH-P dataset, called *Poisoned Spiral*, is generated, and based on this dataset, we demonstrate the definition of LH-P and the basic idea of MiL. We also use this dataset to demonstrate that dimension reduction (e.g., t-SNE) and imbalance handling (e.g., data augmentation with a diffusion model, SMOTE resampling + noise injection, weighted loss) cannot act as a possible solution path for LH-P. We also demonstrate our LH-P solution for this baby LH-P dataset (see **Supp. X 2–6**).
>
> 9. We extend the Learning-Hard Problem (LH-P) to its regression version: LH-R (Supp. T).
>
> 10. Micro-Learning (MiL) FAQ (a total of 53 questions about MiL) **(Supp. X).**
>
> 11. Baseline hyperparameter details (Supp. L).
>
> 12. Precision Traininglet Construction (PTC) in MiL (Supp. G+), including the algorithm and a plain-English interpretation.
>
> 13. High-dimensional noisy SEC 8-K vectorization data “fails” MiL (Supp. K).
>
> 14. Relevant code and datasets are posted in the provided link.
>
> 15. Supplementary material length: 28 pages before revision and 60 pages after revision.
>
> **Thank you for your time.**

---

### Official Review · Reviewer_L2pd · 2025-11-05

**Soundness:** 3
**Presentation:** 1
**Contribution:** 3
**Rating:** 2
**Confidence:** 4

**Summary:**

This manuscript formally defines a category of Learning-Hard Problems (LH-Ps) characterized by non-linearity, noise, imbalance, and small sample sizes, which pose significant challenges to generalization. The authors argue that the failure of deep learning and other standard models on LH-Ps stems from their inability to incorporate domain knowledge. To address this, they introduce the Micro-Learning (MiL) framework, which performs localized learning by extracting domain-informed subsets from the training data for each query point—a process reminiscent of test-time learning. This design effectively filters out irrelevant or noisy samples, thereby mitigating overfitting risks. Experiments demonstrate that MiL outperforms deep learning and other standard baselines on LH-P tasks. Additionally, the authors propose a Learning-Hard Index to measure the difficulty of a task, and provide theoretical guarantees regarding the optimality and complexity of their approach.

**Strengths:**

The initial problem analysis and the introduction of this work are well-articulated. The proposed concept of a problem difficulty index and the motivation for using traininglets are particularly compelling and may offer valuable insights for the domain generalization research community.

**Weaknesses:**

The manuscript suffers from significant presentation issues, with the methodology being ambiguously explained and key definitions omitted. Crucially, the process of incorporating domain knowledge remains unclear. The experimental design lacks rigor: comparisons against specialized state-of-the-art deep learning methods for imbalanced or small-sample problems are absent, undermining the claimed advantages. Furthermore, the evaluation is incomplete due to the lack of ablation studies and assessments across datasets of varying difficulty levels.

**Questions:**

1.The manuscript requires a precise operational definition of "appropriate domain knowledge," clarifying how it is quantitatively or qualitatively incorporated into the MiL framework.

2.Line 92, please define "modern MIR system" or provide a canonical reference.

3.Significant writing issues persist, particularly in mathematical notation. Symbols (e.g., f_dm in Line 146) must be formally defined upon their first appearance.

4.Equation (1): The definition of the Adjusted Mutual Index (AMI) is absent. Furthermore, the rationale for selecting the threshold 0.8 requires explicit justification.

5.The methodology for obtaining t-SNE embeddings (X_r) from high-dimensional image data needs elaboration. Please specify the feature extraction pipeline (e.g., whether a pretrained DNN was employed) to ensure reproducibility.

6.To properly demonstrate the meaningfulness of the Learning-Hard Index (LHI), it should be quantitatively evaluated on established benchmark problems with known difficulty characteristics (e.g., domain shift vs. balanced classification).

7.Line 180: The computational procedure for the norm in F_r(f) is unspecified.

8.Propositions 1 and 2 establish that, for a given hypothesis class H, MiL identifies the function with minimal local Rademacher complexity, thereby reducing its overfitting propensity relative to other functions in H. A pertinent question remains: how does the framework guarantee that this particular function is also effective at solving the underlying task, rather than merely being the least overfit?

9.Line 203: Contains a typo "S_pS"? Formatting issues: Excessive spacing (Line 295), incorrect equation references (Line 301), and inconsistent notation (e.g., "radius," "neighbourhood," "data split") require resolution.

10.Section 4.2 requires restructuring for logical coherence and readability. Line 311: The impact of test batch size on performance needs explanation. The protocol for single-sample inference should be specified. The design rationale and fusion strategy for the four meta-traininglets lack motivation.

11.The visualization of results is suboptimal, with figures being too small and analyses lacking depth. For instance, in Figure 4, if the intent is to demonstrate that traininglets possess superior separability, it is unclear what baseline they are being compared against. Critical parameters such as batch size are omitted. Including test samples in the visualization would more effectively illustrate whether the traininglets generalize well.

12.The proposed method involves numerous nuanced design choices and hyperparameters, raising concerns about its general applicability. The core mechanism for domain knowledge integration remains conceptually unclear throughout the technical exposition.

---

> ### Author Response · Authors · 2025-11-25
> **Reply to Reviewer L2pd**
>
> **We thank the reviewer for their detailed critique.**  We have performed a **comprehensive rewrite**.
>
> **Q1. Domain Knowledge fusion Resp**
>
> A: We changed "domain knowledge" into **"label-aware structural knowledge"**
>
> B: Knowledge fusion is via traininglet construction: **it selects discriminative points (label‑aware) while preserving geometric proximity (structural)**  2 ways fo traininglet construction:
>
> *   **Naïve Traininglet Construction (NTC):** Designed for relatively cleaner data,. NTC filters "false neighbors" by intersecting metric balls around $x'$ to enforce consistent multi-view similarity
>
> *   **Precision Traininglet Construction (PTC):** For noisy LH-Ps, PTC employs a 4-stage pipeline: (1) *probing* optimal parameters; (2) *sanitizing* persistent errors and neighbors; (3) *fusing* complementary meta-traininglets; and (4) *pruning* residual noise.
>
> C:  We detail the **"why and how"** in the revision (see "MiL Core," "NTC/PTC," "Why PTC Works," and **Fig. 3**).
>
> D: Result: Fitting deterministic **learninglets** (e.g., SVM) on local **traininglets** replaces complex global models, ensuring inherent overfitting resistance , interpretability , and reproducibility
>
> **Q2. L92 issue:**  **Removed the vague term.**  The revised text now cites the specific baselines used.
>
> **Q3 f_dm issue:** Fixed it and made math more smooth.
>
> **Q4: AMI and 0.8 justification Resp:**
>
> 4A:  Revised Sect 3 to define AMI, formally isolating it from Eq. (1).
>
> 4B: **Suppl. O** confirms a strong negative correlation ($r=-0.67, p=0.035$); standard performance collapses at $\text{LHI} \ge 0.80$, validating our threshold.
>
> **Q5 resp:**
>
> **5A:** t-SNE naturally handles non-image high-dimensional data (e.g., Ovarian: 20,000+ genes across 266 samples; see Suppl. J).
>
> **5B:** For image data, we first extract DL embeddings with a CNN/ResNet (as for CIFAR-100) to avoid raw pixel similarity
>
> **5C**:  Pretrained DNNs often reduce reproducibility (e.g., **overfitting**); MiL is to avoid it. See **Supp Q** for pretraining vs MiL.
>
> **Q6 Resp**
>
> * We evaluated 10 datasets spanning known difficulty levels (LHI: 0.16–0.99), from “easy/balanced” (Iris, MNIST, EMO-DB, CIFAR-100) to “hard/shifted” (Ovarian, CASIA, SAVEE); see **Suppl. O**
>
> * A significant negative correlation ($r = -0.67, p=0.035$) between LHI and DNN accuracy across 10 tasks confirms that LHI faithfully captures intrinsic learning difficulty.
>
> **Q7 Resp:**  It is norm-2: see updated Sect 3.
>
> **Q8 Resp:**  **Agreed:** Minimal complexity alone does not guarantee accuracy; MiL closes this gap by inducing the predictor on a traininglet with maximized local 'SNR', ensuring effectiveness via steps
>
> 8A:  **Alignment (Prop 3):** We proved that PTC strictly contracts the Total Variation distance between the training and query distributions, ensuring the model is trained on relevant data.
>
> 8B:  **Solvability (Prop 2):** We proved that training on this aligned, effectively "denoised" subset yields a predictor with a strictly higher probability of matching the Bayes-optimal classifier than any global model, provided the subset retains class-discriminative structure.
>
> 8C:  **Realization (PTC/NTC):** Our knowledge fusion heuristics (NTC/PTC) are designed to satisfy the condition in (2). By enforcing label balance and removing high-entropy "bad guys" (Lemma 1), we construct a traininglet that is both **low-complexity** (Prop 1) *and* **highly discriminative**.
>
> Thus, MiL does not merely seek the "least overfit":  it seeks the **simplest model that explains the local manifold structure**.
>
> **Q9 resp:** Fixed.
>
> **Q10 Resp:**
>
> **A: Rewrote Sect 4.2**  and added **Fig 3**
>
> **B. Batch Size ($z$) trades off precision vs. efficiency**  . $z=1$ offers maximal customization per query, larger $z$ speeds MiL via a shared traininglet for a batch. The *Probing* stage optimizes this trade-off.
>
> **C. Fusion Rationale:** We motivated the four meta-traininglets as capturing **orthogonal manifold views**:
> *   $\mathcal{T}^{(1)}$ (Local Ball): Captures **geometric proximity**.
> *   $\mathcal{T}^{(2)}, \mathcal{T}^{(3)}$ (1/2-hop): inject label-aware semantic connectivity.
> *   $\mathcal{T}^{(4)}$ (Anchor): Provides **stochastic regularization** against local overfitting
>
> **Q11 resp:**
>
> We revised Fig. 4 (Fig. 5 in the revision) and the traininglet visualization to explicitly contrast traininglets with the full-dataset baseline, specifying batch sizes and marking embedded test queries.  We deepened the analysis by quantifying the massive LHI drop ($\ge 0.79 \to \le 0.26$), which validates **Thm 2** (complexity reduction), **Prop. 1** (sweet-spot existence), and **Prop. 3** (TV contraction).
>
>
> **Q12 resp:**
>
> We address tuning fragility with Probing Learning, which automatically optimizes key hyperparameters; MiL’s SOTA results across music, health, speech, medicine, and vision further demonstrate robust generalization
>
> **Please see answers to Q1**
>
> **Again, we thank the reviewer for their review!**

---

> ### Author Response · Authors · 2025-12-01
> **Revision update and summary for Reviewer L2pd (11/30/25)**
>
> **Dear Reviewer:**
>
> **We have made the following updates/additions in our revision:**
>
> 1. The paper has been rewritten to address your concerns, and its presentation is greatly improved.
>
> 2. One new figure has been added.
>
> 3. The CIFAR-100 dataset is included.
>
> 4. Our SC-let is applied to CIFAR-100 to demonstrate MiL’s scalability to large-scale data (code included).
>
> 5. Five extra SOTA pipeline comparisons are included: two meta-learning models (ProtoNet and MAML), SAM, LNN (liquid neural networks), and pretraining models (see Supp. M, P, Q).
>
> 6. Learning-hard index robustness and sensitivity analysis (Supp. N).
>
> 7. Empirical validation of the LHI cutoff for learning-hard: a total of 10 datasets are used to demonstrate LHI and its cutoff of 0.8 (Supp. O).
>
> 8. One baby LH-P dataset, called *Poisoned Spiral*, is generated, and based on this dataset, we demonstrate the definition of LH-P and the basic idea of MiL. We also use this dataset to demonstrate that dimension reduction (e.g., t-SNE) and imbalance handling (e.g., data augmentation with a diffusion model, SMOTE resampling + noise injection, weighted loss) cannot act as a possible solution path for LH-P. We also demonstrate our LH-P solution for this baby LH-P dataset (see **Supp. X 2–6**).
>
> 9. We extend the Learning-Hard Problem (LH-P) to its regression version: LH-R (Supp. T).
>
> 10. Micro-Learning (MiL) FAQ (a total of 53 questions about MiL) **(Supp. X).**
>
> 11. Baseline hyperparameter details (Supp. L).
>
> 12. Precision Traininglet Construction (PTC) in MiL (Supp. G+), including the algorithm and a plain-English interpretation.
>
> 13. High-dimensional noisy SEC 8-K vectorization data “fails” MiL (Supp. K).
>
> 14. Relevant code and datasets are posted in the provided link.
>
> 15. Supplementary material length: 28 pages before revision and 60 pages after revision.
>
> **Thank you for your time.**

---

### Official Review · Reviewer_rK2M · 2025-11-06

**Soundness:** 2
**Presentation:** 1
**Contribution:** 3
**Rating:** 4
**Confidence:** 4

**Summary:**

This paper introduces a new concept calls Learning-Hard Problems which are problems that are:
1. Every model fails to generalize the entire dataset
2. Solvable once additional knowledge \phi is added to the model
These problems can be identified by using the Learning-Hard Index (LHI) that indicates the difficulties of learning on the full data
These problems are said to have solvable regions in the data where the learner f can generalize well to test data. These are compact regions within the dataset that have strong locality and knowledge representation that are called traininglets. However, finding these subsets are NP-complete and requires a greedy algorithm.
The algorithm is called Precision Traininglet Construction (PTC) pipeline with probing learning, sanitization, meta-traininglet fusion, and precision pruning.

**Strengths:**

This paper has a very strong theoretical backing that explains the problems and its solutions. The paper finds a very niche type of datasets that require a different approach than conventional methods. Overall, it has strong novelty with a lot of theoretical backings.

**Weaknesses:**

Although many models are used for experiments, they seem to be too generic. The method should be compared against the best attemps at solving the datasets mentioned in the papers. Generic methods like CNN have many variations with different degrees of effectiveness and might need special modifications for specific tasks. Overall, it is not clear that the baselines models are the best attempts at the datasets.

The threshold 0.8 LHI for LHP classification seems arbitrary since a dataset like COVID-19 with LHI < 0.8 still fit the characteristics of LHP where most models fail to generalize.

Part 3 of PTC does not make sense: The 4 sets are not specified.

All of the graph figures are too small to be read, they should all be enlarged. However, figure 3 and table 2 should be merged since they all represent the experimental results while figure 3 should have all methods on the same side for easier reading or be made into a tables like table 2 where just listing the results of MiL in table 2 does not serve any purpose.

Writing errors in lines 203 - S_pS instead of S_p, 301 and 302 are missing references

In section 4.2, NTC and PTC are described where NTC is a naive step that PTC is built upon, therefore NTC should be its own subsection like PTC.

**Questions:**

What is the significant of the 0.8 threshold?

Many methods mentioned in the related sections do not appear in the experiments suchs as MAML, SAM, etc. Are they the best attemps at the datasets?

---

> ### Author Response · Authors · 2025-11-26
> **Reply to Reviewer rK2M**
>
> We sincerely thank the reviewer for the thoughtful feedback and for recognizing the **strong theoretical backing** of our work, as well as the novelty of identifying **Learning-Hard Problems (LH-Ps)** and the utility of the **Learning-Hard Index (LHI)**. We value your constructive criticisms regarding baselines and presentation, which we address below.
>
> **1. Baselines and Comparisons**
>
> *   **1A: Two extra Meta-Learning pipelines for comparisons (MAML, ProtoNet):** We compared two meta-learning models with our MiL, and their results are detailed in **Suppl. M** (referenced in the "Results" section, paragraph "MiL vs meta-learning and SAM").
>     *   *Results:* MiL outperforms MAML and ProtoNet significantly (raising average accuracy from 67.8% to 87.4%) for the LH-Ps.
>
> *   **1B: Regarding "Generic" Baselines:** We selected our 15 baselines because they effectively represent the current SOTA architectures used in the specific domains of the datasets (e.g., CNNs for speech/medical signals).
>
> *   **1C: Specific SOTA:** For IRMAS, we referenced Han et al. and Yu et al. in the Introduction to demonstrate that even highly-engineered, domain-specific SOTA models struggle (60-68% F1) compared to MiL (84% F1).
>
> *   **1D: No pre-existing models:** The COVID-19 and Ovarian datasets are curated, domain-specific tasks representing "Quasi-LH-P" and "LH-Ps" (small sample, high imbalance). To our knowledge, there are no pre-existing, specialized architectures published specifically for these exact dataset splits. In this context, high-capacity "generic" DL models (ResNet, Bi-LSTM, CapsNet) **are** the state-of-the-art "best attempts" available to a practitioner.
>
> **2. Other model: SAM**
> We also implemented the SAM model for our benchmark data, following the reviewer's suggestion. SAM demonstrated much lower performance than our MiL due to its lack of a knowledge-fusion mechanism. See paragraph: "MiL vs meta-learning and SAM": , and more details can be found in **Suppl. P**.
>
> **3. The LHI Threshold (0.80) and COVID-19**
>
> *   **Significance of 0.80:** Theoretically, LHI=0.80 means that $\le 0.20$ of the neighborhood mutual information of samples is available during clustering, suggesting a learning-hard task. **A significant negative correlation ($r=-0.67, p=0.035$) between LHI and performance validates $\text{LHI} \ge 0.80$ as the critical threshold where standard model efficacy collapses** (see **Suppl. O**). Above 0.80, **the loss of local neighborhood structure typically renders standard global optimization ineffective**.
>
> *   **The COVID-19 Case (LHI 0.78):** We explicitly included the COVID-19 dataset (LHI $\approx$ 0.785) as a **boundary case** (termed a "Quasi-LH-P" in Sec. 3) to test MiL's sensitivity. It supports the definition: because it is slightly *below* 0.80, standard models perform *better* on it than they do on the harder datasets (LHI > 0.8), but MiL still provides a significant lift. We will clarify that 0.80 is a "soft" heuristic threshold indicative of high risk, similar to p-value thresholds in statistics.
>
> **4. Clarifying Part 3 PTC (The 4 Meta-Traininglet fusion)**
>
> **4A:** We updated Section 4 to make this clearer.
>
> **4B:** We fuse 4 Meta-Traininglets into a single, label-complete union to capture **complementary knowledge & structural views:**
>
> *   $\mathcal{T}^{(1)}$ (Local Ball): Captures raw **geometric proximity**.
> *   $\mathcal{T}^{(2)}, \mathcal{T}^{(3)}$ (1-hop/2-hop): Injects **label-aware semantic structure** via graph connectivity (friends-of-friends).
> *   $\mathcal{T}^{(4)}$ (Random Anchor): Provides **stochastic regularization** to prevent overfitting to disconnected local pockets.
>
> **5. Presentation issues and Typos**
>
> *   **5A: Small Figure issue:** We enlarged all figures for better readability.
> *   **5B: Figure 3 (Figure 4 in revision) / Table 2 Merge:** We attempted this, but the full Figure 4 is too large and compromises the presentation of the paper. Therefore, we placed it in **Suppl. H** rather than merging them.
> *   **5C: NTC subsection:** We followed the reviewer's suggestion to give NTC and PTC parallel subsection structures in the revision.
> *   **5D: Typos:** We fixed all of them in the revision.
>
> **Again, we sincerely thank the reviewer for their insightful comments!**

---

> ### Author Response · Authors · 2025-12-01
> **Revision update and summary for Reviewer rK2M (11/30/25)**
>
> **Dear Reviewer:**
>
> **We have made the following updates/additions in our revision:**
>
> 1. The paper has been rewritten to address your concerns, and its presentation is greatly improved.
>
> 2. One new figure has been added.
>
> 3. The CIFAR-100 dataset is included.
>
> 4. Our SC-let is applied to CIFAR-100 to demonstrate MiL’s scalability to large-scale data (code included).
>
> 5. Five extra SOTA pipeline comparisons are included: two meta-learning models (ProtoNet and MAML), SAM, LNN (liquid neural networks), and pretraining models (see Supp. M, P, Q).
>
> 6. Learning-hard index robustness and sensitivity analysis (Supp. N).
>
> 7. Empirical validation of the LHI cutoff for learning-hard: a total of 10 datasets are used to demonstrate LHI and its cutoff of 0.8 (Supp. O).
>
> 8. One baby LH-P dataset, called *Poisoned Spiral*, is generated, and based on this dataset, we demonstrate the definition of LH-P and the basic idea of MiL. We also use this dataset to demonstrate that dimension reduction (e.g., t-SNE) and imbalance handling (e.g., data augmentation with a diffusion model, SMOTE resampling + noise injection, weighted loss) cannot act as a possible solution path for LH-P. We also demonstrate our LH-P solution for this baby LH-P dataset (see **Supp. X 2–6**).
>
> 9. We extend the Learning-Hard Problem (LH-P) to its regression version: LH-R (Supp. T).
>
> 10. Micro-Learning (MiL) FAQ (a total of 53 questions about MiL) **(Supp. X).**
>
> 11. Baseline hyperparameter details (Supp. L).
>
> 12. Precision Traininglet Construction (PTC) in MiL (Supp. G+), including the algorithm and a plain-English interpretation.
>
> 13. High-dimensional noisy SEC 8-K vectorization data “fails” MiL (Supp. K).
>
> 14. Relevant code and datasets are posted in the provided link.
>
> 15. Supplementary material length: 28 pages before revision and 60 pages after revision.
>
> **Thank you for your time.**

---

### Official Review · Reviewer_Dtag · 2025-11-06

**Soundness:** 2
**Presentation:** 1
**Contribution:** 3
**Rating:** 4
**Confidence:** 3

**Summary:**

This paper introduces the concept of Learning-Hard Problems (LH-Ps), a class of tasks at which most models fail, even though there exists a high-quality solution that could be reached through the use of domain knowledge. The authors formalize this by proposing a data-centric metric called the Learning-Hard Index (LHI) that can identify such problems before training.
The main contribution is the Micro-Learning framework (MiL), which aims at solving LH-Ps. In contrast to training one global model, MiL learns, for each test instance, a new, local, interpretable model (e.g., an SVM). This local model is learned on a so-called "traininglet", a small low-complexity subset of the original training data that is optimized for the given query. The paper theoretically analyzes this process, proving that computing the optimal traininglet is NP-complete, and then proposes a practical heuristic pipeline, Precision Traininglet Construction (PTC), to efficiently generate them.

**Strengths:**

1. Formalization of LH-Ps is a significant conceptual contribution. It provides a clear definition and a diagnostic tool-the LHI-for a class of problems that practitioners often encounter but for which they lack the vocabulary to describe formally.

2. The MiL framework represents a break in the dominant paradigm of training large, global models. Indeed, the basic intuition of "localizing the solution" by training instance-specific traininglets is intuitive and powerful, particularly in datasets with complex, nonlinear structures or high levels of noise and imbalance.

3. The rigorous theoretical work by the paper provides bounds to strengthen the claims. The intractability of the problem in question is clearly established by the NP-completeness proof for optimal traininglet selection, thus justifying heuristics like PTC.

**Weaknesses:**

1. The main weakness of the paper is the ambiguity between the theoretical claims and the experimental demonstrations. Particularly, the methodology proposes a generalizable framework viewpoint for solving LH-Ps but the experiments are conducted on trivial timeseries, tabular datasets which critically limits the scope of the work. Datasets like iNaturalist, CUBS, ImageNet etc. are naturally imbalanced which should be experimented with. If not, the scope of the paper should be adjusted accordingly.

2. The paper targets model overfitting as a resultant of imbalance in real-world scenarios. However, no methods combatting challenges in longtail imbalance are cited or contrasted. Eg. GLMC (Du etal., 2023), PaCo (Cui etal., 2021) etc.

3. The clarity in the figures are very poor in the current version - The included text in Fig. 1, mathematical equations in Fig. 2 etc. are barely visible. In Fig. 3 and Fig. 4, the color scheme should be chosen such that each instance is distinguishable.

4. In Fig.3 the goal is to contrast MiL against other learning strategies however MiL performance numbers are placed as a separate figure (per dataset) which makes it hard to compare. I would suggest putting this figure as a horizontal one covering the full textwidth to improve clarity. Table 2 and Fig. 3 seem to convey the same metrics which can be combined.

5. In lines 301 - 302 the reference to the equation is missing.

**Questions:**

1. A specific LHI threshold of 0.80 is proposed to identify a task as an LH-P. The paper would benefit from a more detailed justification or sensitivity analysis for this specific value to show it is not arbitrary.

2. Is there a correlation between LHI and the class imbalance in real-world datasets ?

3. As stated in lines 298-299, a label rebalancing is performed during the naive traininglet construction. Is there any justification on why a rebalancing is necessary ?

4. All datasets adopted in the paper are classification datasets. Is the current method scalable to auxiliary tasks eg. regression, retrieval etc. ?

---

> ### Author Response · Authors · 2025-11-28
> **Reply to Reviewer Dtag**
>
> **We thank the reviewer for their thoughtful assessment. We appreciate the constructive feedback.  We  substantially improve the paper to address them**
>
> **W1. Dataset Scope  & Scalability.**
>
> **Clarification on "Triviality":** We respectfully disagree that small sample size equates to triviality. The datasets used (Ovarian, COVID-19, IRMAS) represent real-world "high-stakes, small-data" problems where SOTA DL models including CapsNets and extensive Pretraining-Fine-tuning pipelines (see **Suppl. Q**), fail significantly (F1 $< 0.65$), whereas MiL succeeds ($> 0.85$). These are precisely the **targeted LH-Ps MiL is designed for**. **Our goal is to address general learning-hard problems, specifically those plagued by noise, scarcity, and imbalance, rather than focus on massive computer vision benchmarks.**
>
> **Scope Clarification:** We ack. that the computational complexity of the Precision Traininglet Construction (PTC) phase currently limits MiL's application to massive-scale datasets like ImageNet. We have explicitly defined the scope in the revised **Limitations** section: MiL is targeted at **"high-stakes, small-to-mid-sized problems"** (e.g., medical diagnostics) where accuracy, interpretability, and reproducibility outweigh training speed. We emphasize that **super large vision datasets are NOT the target of this work, nor is this intended to be a computer vision paper.**
>
> **CIFAR-100 Experiment:** To demonstrate generalizability, we added a **Discussion** section detailing a "SC-let" applied to **CIFAR-100**. Using Naïve Traininglet Construction (NTC), we achieved 80.32\% accuracy, demonstrating that MiL extend to vision tasks, though optimizing for massive scale remains future work
>
> **W2. Comparison with  Baselines (GLMC, PaCo).**
> *   We  updated the **Related Work** to  contrast MiL with long-tail vision strategies like **GLMC** and **PaCo**.
> *   **Theoretical Distinction:** We argue that "long-tail imbalance" (distributional skew) is fundamentally different from "Learning-Hardness" (topological complexity).
>     1.  **Global vs. Local:** GLMC and PaCo attempt to optimize a global model $h_{\Theta, S}$ over the full dataset $S$. However, **Prop. 2 (Traininglet Sufficiency)** proves that for LH-Ps, the probability of correctness for a model trained on a local traininglet $S_p$ strictly exceeds the supremum of *any* global model: $\Pr(h_{S_p}) > \sup_{\Theta} \Pr(h_{\Theta, S})$. Thus, even a "balanced" global model remains theoretically inferior to MiL’s local approach for these problems.
>     2.  **Imbalance vs. Hardness:** These methods address *sample quantity* (imbalance), but not the **intrinsic hardness** (LHI). As shown in our benchmarks, even balanced datasets (e.g., IRMAS) have high LHI due to manifold entanglement. **Thus, global long-tail methods cannot untangle these manifolds; they only reweight them.**
> *   **Validation:** Consequently, we compared MiL against domain-appropriate generative baselines (DDPM, DAE) in **Suppl. Q**, confirming that global imbalance handling fails where local knowledge fusion succeeds.
>
> **W3. Presentation Improvements.**
>
> We enhanced figure legibility (size, resolution, color) and corrected all typos. The comparison is now **Figure 4** (original Fig 3). Regarding the Table 2/Figure 4 merge, we found a full integration too dense; instead, we present a focused comparison in the main text and the comprehensive visualization in **Suppl H**."
>
> **Q1. Resp:**
> *  The threshold is empirically derived, not arbitrary. We added **Suppl O**, presenting a correlation analysis across $N=10$ datasets. We observe a statistical **phase transition** at $\text{LHI} \approx 0.80$: below this, standard DNNs remain robust (D-index $> 1.8$); above it, performance significantly degrades ($r = -0.67, p=0.035$). This empirically validates 0.80 as the critical boundary for the "Learning-Hard" regime.
>
> **Q2: Resp:** LHI measures **intrinsic complexity** (manifold entanglement), not just count disparity. While imbalance may contribute to hardness, balanced datasets can exhibit high LHI (e.g., IRAMS), whereas separable imbalanced data (COVID) do not. Thus, LHI captures intrinsic structural complexity independent of class ratios
>
> **Q3.  Label Rebalancing in NTC.**
> *   Label rebalancing is strictly necessary to satisfy the **Full Coverage Constraint** (Def. 2). MiL trains a local discriminator (e.g., SVM) for each query; if a traininglet lacks examples of a specific class $C$, the local decision boundary for that class is undefined. Forcing the inclusion of the nearest neighbor from $C$ ensures the local model considers all competing hypotheses, preventing trivial rejection of minority classes and undefined behavior
>
> **Q4. Scalability to Auxiliary Tasks.**
> *   We extend LH-P to regression (see **Suppl. T**) and update in the **Conclusion**.
>
> **Again, we thank the reviewer for their insightful review! We hope our comprehensive revisions and clarifications satisfactorily address your concerns.**

---

> ### Author Response · Authors · 2025-12-01
> **Revision update and summary for Reviewer Dtag  (11/30/25)**
>
> **Dear Reviewer:**
>
> **We have made the following updates/additions in our revision:**
>
> 1. The paper has been rewritten to address your concerns, and its presentation is greatly improved.
> 2. One new figure has been added.
> 3. The CIFAR-100 dataset is included.
> 4. Our SC-let is applied to CIFAR-100 to demonstrate MiL’s scalability to large-scale data (code included).
> 5. Five extra SOTA pipeline comparisons are included: two meta-learning models (ProtoNet and MAML), SAM, LNN (liquid neural networks), and pretraining models (see Supp. M, P, Q).
> 6. Learning-hard index robustness and sensitivity analysis (Supp. N).
> 7. Empirical validation of the LHI cutoff for learning-hard: a total of 10 datasets are used to demonstrate LHI and its cutoff of 0.8 (Supp. O).
> 8. One baby LH-P dataset, called *Poisoned Spiral*, is generated, and based on this dataset, we demonstrate the definition of LH-P and the basic idea of MiL. We also use this dataset to demonstrate that dimension reduction (e.g., t-SNE) and imbalance handling (e.g., data augmentation with a diffusion model, SMOTE resampling + noise injection, weighted loss) cannot act as a possible solution path for LH-P. We also demonstrate our LH-P solution for this baby LH-P dataset (see Supp. X2–6).
> 9. We extend the Learning-Hard Problem (LH-P) to its regression version: LH-R (Supp. T).
> 10. Micro-Learning (MiL) FAQ (a total of 53 questions about MiL) (Supp. X).
> 11. Baseline hyperparameter details (Supp. L).
> 12. Precision Traininglet Construction (PTC) in MiL (Supp. G+), including the algorithm and a plain-English interpretation.
> 13. High-dimensional noisy SEC 8-K vectorization data “fails” MiL (Supp. K).
> 14. Relevant code and datasets are posted in the provided link.
> 15. Supplementary material length: 28 pages before revision and 60 pages after revision.
>
> **Thank you for your time.**

---

### Meta-Review · Area_Chair_mbeV · 2026-01-06

**Summary:**

Reviews on this paper were mixed -- with some being positive and others being quite negative. Many reviewers were impressed with the novelty of the paper and the theoretical construct of a LH-P. On the negative side, reviewers pointed out issues with presentation, and lack of comparisions to methods -- the paper mainly compares to generic baselines, rather than methods that are more targeted to addressing imbalance and other issues that make the studied problems hard.

Given the lack of consensus, and in my opinion, the lack of general understanding of the paper by the reviewers, I took a closer look myself.  Personally, I found several major issues with the paper that preclude acceptance.
1. The definition of LHI very disconnected from the definition of a LH-P -- they seem to be two completely different ways of trying to explain why a problem is hard. I don't see e.g. how LHI incorporates the idea that after a suitable transformation of the data, the problem becomes easy.
2. Taking a look at the theory of the paper, I was not able to follow it. E.g. the proof of Proposition 1 uses significant undefined notation, makes assumptions that are not stated in the proposition statement, and never even references the definition of a LH-P or any aspects of this definition.
3. The proof of Theorem 1 (NP hardness for trainingless selection) is very confusing and ultimately incorrect. I don't understand the logic 'cycle' that is depicted on page 6 of the supplemental, and it is concerning that the authors seem to be highlighting a use of cyclical logic in the proof. The entire paragraph above 'Proof of Theorem 1' appears nonsensical to me, and doesn't make sense given that no problem that is being reduced from has been mentioned yet. Even the problem that is apparently being reduced from (a variant of 3-SAT where no variable appears in more than 3 clauses) is trivial -- as shown in the referenced paper where the problem is introduced, all instances of this problem are satisfiable by the Hall's marriage theorem.

**Reviewer Concerns:**

The main outstanding concerns leading to my decision are mine above. I do not feel that the reviewers really understood the theoretical aspects of the paper enough to evaluate it. The authors did address many reviewer concerns about empirical evaluation, specific questions about experimental set up and motivations/choices in designing the methods, etc.

**Reviewer Scores:**

Reviewer Dtag may have increased their score by 1. I am not sure about the other reviewers -- perhaps a few would have increaased their scores by 1.

---

### Decision · Program_Chairs · 2026-01-26

Reject